# Respiratory long COVID in aged hamsters features impaired lung function post-exercise with bronchiolization and fibrosis

Long-term consequences of SARS-CoV-2 infection affect millions of people and strain public health systems. The underlying pathomechanisms remain unclear, necessitating further research in appropriate animal models. This study aimed to characterize the trajectory of lung regeneration over 112 days in the male hamster model by combining morphological, transcriptomic and functional readouts. We demonstrate that in the acute phase, SARS-CoV-2 Delta-infected, male, aged hamsters show a severe impairment of lung function at rest. In the chronic phase, similar impairments persisted up to 7 weeks post-infection but were only evident after exercise on a rodent treadmill. The male hamster model recapitulates chronic pulmonary fibrotic changes observed in many patients with respiratory long COVID, but lacks extra-pulmonary long-term lesions. We show that sub-pleural and interstitial pulmonary fibrosis as well as alveolar bronchiolization persist until 112 dpi. Interestingly, CK8[+] alveolar differentiation intermediate (ADI) cells are becoming less prominent in the alveolar proliferation areas from 28 dpi on. Instead, CK14[+] airway basal cells and SCGB1A1[+] club cells, expressing cell proliferation markers, mainly populate alveolar bronchiolization areas at later time-points. We postulate that pulmonary fibrosis and SCGB1A1[+] club cell-rich areas of alveolar bronchiolization represent potential risk factors for other diseases in long-COVID survivors.

In 2023, the WHO officially declared an end to the public health emergency caused by the severe acute respiratory syndrome coronavirus 2 (SARS-CoV-2)[1]. Acute case numbers have been on a stable downward trend since the beginning of 2023. However, as the cases of acute diseases declined, the concern shifted to sub-acute and persistent complications of the acute infection. Frequently observed symptoms indicate a dysfunction of multiple organ systems, like the respiratory tract or the nervous system, and can persist for 4 weeks or more[2-8]. Relapses and the development of new symptoms after 30 or more days after the initial infection are also reported[7]. This condition has been termed post-COVID-19 condition by the WHO which is commonly known as long-COVID or post-acute sequelae of COVID-19 (PASC). Respiratory PASC is now one of this diseases most common

phenotypes[9]. Around 12% of patients suffer from symptoms like chest pain, ageusia or anosmia, shortness of breath, dyspnea, or general fatigue[9]. PASC is reported especially in patients who suffer from severe acute disease[10]. Multiple mechanisms of pathogenesis have been proposed, including immunological dysfunction[11,12], viral persistence or latent virus reactivation[13,14], manipulation of host mitochondria[15], and, specifically in the lung, impairment of gas exchange due to dysregulated alveolar regeneration[16] and fibrosis[17]. Correlating pathomorphological changes with in vivo lung function analysis would be helpful to better understand respiratory long-COVID. In our previous work[18], we demonstrated that the Syrian golden hamster (*Mesocricetus auratus*) represents a suitable animal model to study alveolar regenerative mechanisms following SARS-CoV-2 infection in the acute to sub-acute

✉ e-mail: Wolfgang.baumgaertner@tiho-hannover.de

phase until 14 days post infection (dpi). We found that foci of peri-bronchiolar epithelial proliferation derived from airway progenitors and cytokeratin 8 (CK8)[+] alveolar differentiation intermediate (ADI) cells persist until 14 dpi. These foci were accompanied by the onset of lung fibrosis, indicating a dysregulated regeneration. In the current work, we sought to expand these findings and determine whether this process is associated with functional impairment and whether it resolves over time, and consequently to determine if the hamster model can be used for research on respiratory long-COVID. Based on what is known about long-COVID in humans and hamsters, we hypo-thesized: (i) SARS-CoV-2 infection induces long-lasting lung function impairment; (ii) akin to what is reported in humans in the chronic phase of the disease, physical exercise-induced stress would exacer-bate this lung function impairment; (iii) the underlying cause of this impairment is a dysregulated alveolar regeneration. The current longitudinal study evaluated a series of parameters over a period of 16 weeks (112 days), based on the observation that patients suffering from long-term COVID tend to have symptoms for more than 12 weeks[2]. Only male, aged (-1-year-old) hamsters were chosen for the study. The rationale behind this choice was to thoroughly characterize a respiratory long COVID-19 model to better understand the patho-genetic processes associated with this syndrome. The choice of our study design was based on the following observations (i) old age and male sex are risk factors for severe acute COVID-19[8,19–23], (ii) severe acute disease is a risk factor for respiratory long-COVID[8,21,23,24], and (iii) male hamsters show more severe disease course with more prominent lung function alterations and slower recovery, as well as more severe histological pulmonary lesions compared to females[22,25–30]. SARS-CoV-2 Delta variant was chosen since it showed the most prominent pul-monary pathological changes among the most common variants worldwide at the time[31,32].

Our goal was to provide comprehensive insights into the tra-jectory of post-acute sequelae of a respiratory SARS-CoV-2 infection in the hamster model, focusing on correlating pathomorphological findings with lung functional parameters. Serial, non-invasive lung function data was gathered using a combination of whole-body ple-thysmography (WBP) with respiratory gas analysis combined with exercise on a rodent treadmill. Findings were consecutively related to morphological, transcriptomic, and protein expression data. Our study shows that (i) in the acute phase, SARS-CoV-2-infected ham-sters show a severe impairment of lung function and reduced metabolic activity in a resting state, which resolves at 10 dpi and is associated with acute pneumonia and transcriptomic changes dominated by immune responses; (ii) in the chronic phase, hamsters fail to recover their initial body weight, show lung function altera-tions after exercise persisting up to 7 weeks after infection, a reduced running behaviour in the treadmill, and reduced metabolic activity at rest and after exercise; (iii) histological alterations of lung archi-tecture persist until 112 dpi, including sub-pleural and interstitial fibrosis as well as bronchiolization, and are associated with pro-longed presence of M2-like macrophages and pro-fibrotic tran-scriptomic changes; (iv) viral RNA persists in the lung until 112 dpi; (v) CK8[+] ADI cells do not persist beyond 28 dpi; (vi) airway progenitor cells dominate the cellular composition within the alveolar epithelial proliferation areas in the sub-acute and chronic phase, and SCGB1A1[+] club cells are the predominant cell type in bronchiolization foci; (vii) Ki67 expression is detectable in bronchiolization areas until 112 dpi, suggestive of ongoing proliferative activity.

## Results

### SARS-CoV-2 delta infection markedly affects lung function and metabolic activity during the acute phase of the disease in aged hamsters

Male, 1-year-old Syrian hamsters were infected with SARS-CoV-2 Delta variant and sacrificed at 1, 3, 6, 14, 28, 56, and 112 dpi. During the experiment, repeated lung function and metabolic measure-ments were conducted using a combination of WBP with respiratory gas analysis. Physical exercise was used to exacerbate possible latent changes in the later phase of infection. Accordingly, our study design allowed us to distinguish three phases of the disease: acute phase (infection−6 dpi), sub-acute phase (14 dpi), and chronic phase (28 dpi−112 dpi). Due to animal welfare reasons, physical exercise-induced stress was only applied to animals from 21 dpi onwards (Supplementary Fig. 1). For this reason, the clinical results of the acute and sub-acute phases will be shown separately from the chronic phase.

The acute disease presented with weight loss, upper respiratory clinical signs, and behavioural changes. In the first week after the infection, most SARS-CoV-2-infected animals (34/48) lost up to 10% of their initial body weight. Fewer animals (3/48) lost up to 15%, while one had a maximum weight loss of 17% at 6 dpi. This weight loss was transient, and the animals began to gain weight from around 7 dpi onwards (Fig. 1a). Upper respiratory signs mainly consisted of mild sneezing and nasal discharge, which subsided around day 8 (Fig. 1a). Mild behavioral changes such as reduced grooming and reduced activity were observed in 19/48 animals up until 10 dpi. (Supplementary data 1). Based solely on the weight loss and clinical scoring, we were able to monitor acute disease severity according to previously performed COVID-19 studies[31,33,34]. However, lower respiratory tract signs such as dyspnea or labored breathing were not clearly evident by clinical monitoring. Assessment of subtle lower respiratory signs and altered breathing is difficult in hamsters, particularly in aged males which have a high amount of subcutaneous fat that masks chest movement. To provide a more sensitive and precise readout of breathing parameters, WBP was performed.

WBP revealed marked mechanical lung function changes in the acute phase of the disease. Here, we focused on specific mechanical metrics such as respiratory rate (Frequency), tidal volume (TV), EF50 (mid-expiratory flow), inspiration time (Ti), and expiration time (Te). Infected hamsters showed a significantly decreased frequency and EF50 at 3 and 6 dpi compared to mock-infected animals. TV and Te were significantly higher compared to mock-infected animals at 3 dpi (Fig. 1b). No significant differences were noted in Ti.

WBP measurement was coupled with a respiratory gas analyzer to assess metabolic function. vO2 (O2 uptake), vCO2 (CO2 production), MR (metabolic rate) and RQ (respiratory quotient) were measured. vO2 and, vCO2 were significantly reduced at 3 and 6 dpi in SARS-CoV-2-infected hamster compared to mock-infected animals, whereas MR was significantly reduced only at 3 dpi. No significant changes were observed in RQ. Similar to mechanical WBP changes, these differences between SARS-CoV-2 and mock-infected animals were not observed at 10 or 14 dpi (Fig. 1b).

To summarize, the acute disease induced by SARS-CoV-2 Delta variant infection was characterized by weight loss, nasal discharge, and reduced activity. While no signs of lower respiratory tract dis-tress were detected by clinical monitoring, significant lung function changes were detected by WBP. The data indicate slower and par-tially deeper breathing in SARS-CoV-2-infected hamsters, with pro-longed expiration and reduced expiratory airflow. This breathing pattern is typically observed in conditions with airflow limitation due to obstruction of airways[35,36]. Moreover, respiratory gas analysis revealed a decreased metabolic activity. These changes were marked, transient, and disappeared -10 dpi. However, based on studies in long-COVID-patients[37], we hypothesized that an underlying altera-tion in lung function and metabolism in the chronic phase of the disease could be exacerbated in a situation of physical stress. To substantiate our hypothesis, we evaluated lung function at later time points before and after inducing physical stress, i.e., running on a rodent treadmill.

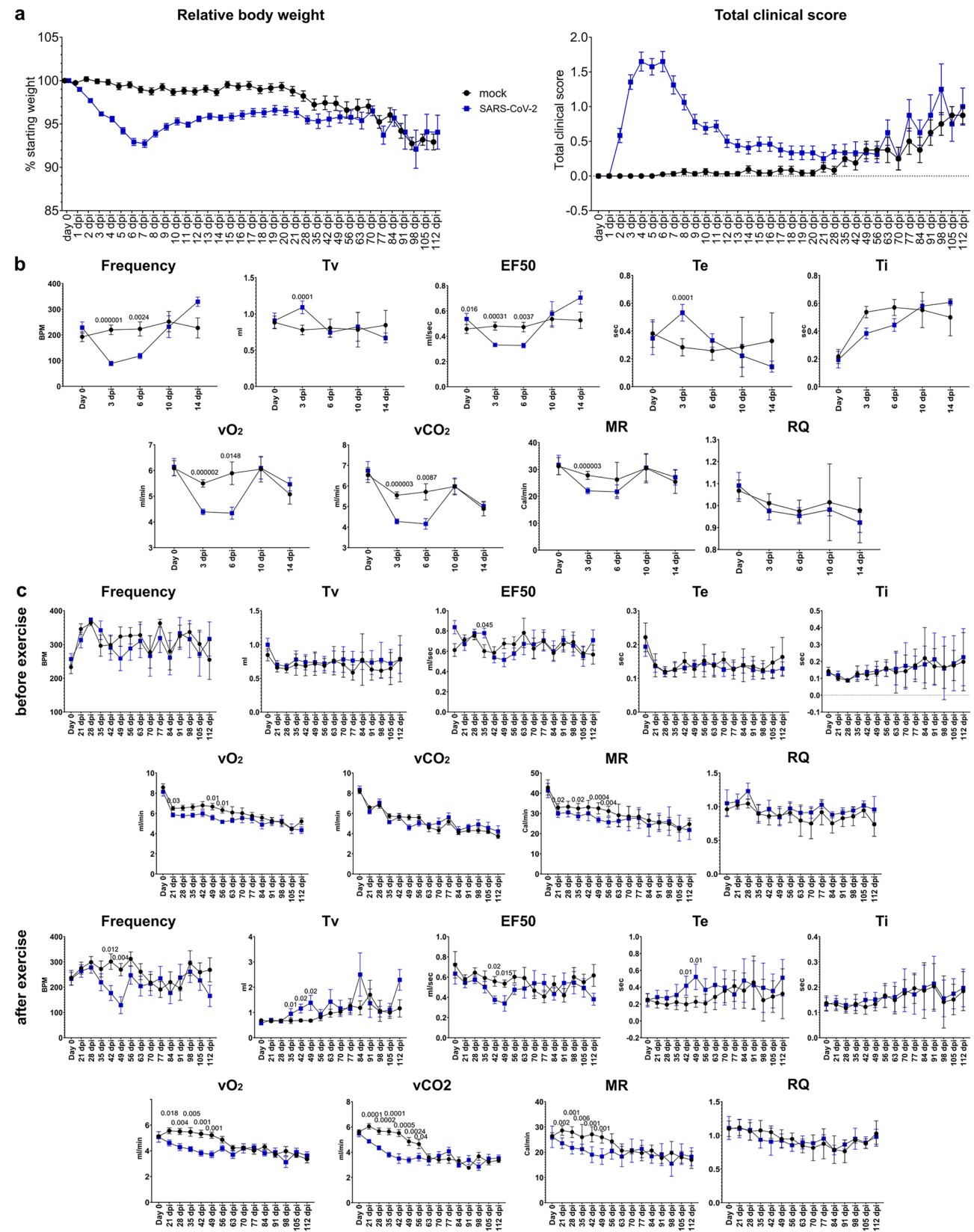

## Lung function and metabolic alterations are detectable after physical exercise on a rodent treadmill in aged hamsters recovering from SARS-CoV-2 Delta infection

After the resolution of acute SARS-CoV-2 infection, hamsters did not show any respiratory signs at the daily clinical evaluation. However, SARS-CoV-2-infected hamsters did not recover their initial body weight, which remained stable between 12 and 63 dpi. A gradual weight loss was observed in mock- and SARS-CoV-2-infected animals, starting from 63 dpi (Fig. 1a). One animal died unexpectedly due to atrial thrombosis. The weight loss at the later time-points and atrial thrombosis were interpreted as age-related or spontaneous background lesions and considered unrelated to the infection[38–40]. In line

**Fig. 1 | SARS-CoV-2 infection causes lung function and metabolic changes that are exacerbated by exercise in aged hamsters. a** Left panel: relative body weight (% starting body weight) of SARS-CoV-2 and mock-infected hamsters over time. Right panel: total clinical score (including weight loss, respiratory, and behavioral signs) of SARS-CoV-2 and mock-infected hamsters over time. Data was collected daily until 21 days post infection (dpi) and weekly from 21 to 112 dpi. **b** Results of whole-body plethysmography (WBP) measurements from 0 to 14 dpi without exercise prior to the measurement. First row: frequency (f) expressed in breaths per minute (BPM), tidal volume (TV), mid-tidal expiratory flow (EF50), expiration time (Te), and inspiration time (Ti). Second row: volume of $O_2$ uptake (v$O_2$), $CO_2$ production (v$CO_2$), metabolic rate (MR), and respiratory quotient (RQ). **c** Results of WBP measurements from 21 to 112 dpi. Upper two rows: measurements taken before exercise. Lower two rows: measurements taken immediately after exercise on a rodent treadmill (10 minutes with accelerating speed starting at 10 m/min and ending at 15 m/min at a 5° upward slope). Graphs show mean and standard error of the mean (SEM). Data from WBP measurements were tested by a Wilcoxon test. A $p$ value of ≤0.05 was chosen as the cutoff for statistical significance. **a, b** $N = 56$ (0–1 dpi), 48 (2–3 dpi), 40 (4–6 dpi), 32 (7–14 dpi), 24 (15–28 dpi), 16 (29–56 dpi), or 8 (57–112 dpi) animals/group. **c** $N = 18$ (0–28 dpi), 12 (29–56 dpi) or 6 (57–112 dpi) animals/group. Source data are provided as a Source Data file.

with the lack of clinical signs related to infection, no differences in lung function between the two groups were detected by WBP in a resting state. However, v$O_2$ and MR were slightly but significantly lower in SARS-CoV-2-infected hamsters between 21 and 56 dpi, indicating lasting effects on metabolism. No differences were observed in v$CO_2$ and RQ.

Long-COVID patients often suffer from reduced pulmonary function, which can be exacerbated by physical exercise[37,41]. To reproduce this condition experimentally, we used a combination of WBP and exercise on a rodent treadmill in the chronic phase of the disease. From 21 dpi onwards, the animals underwent mild physical exercise once a week, consisting of 10 minutes of running on a treadmill, with a slight upward slope and gradually increasing speed (10 m/min to 15 m/min). The intensity of training was well tolerated by the animals, with none of the animals displaying signs of exhaustion. The running behavior was scored with a scoring system that considered the animals' disposition to run constantly and the frequency of breaks. Infected hamsters showed a slight decrease in the group mean score from 21 to 56 dpi compared to the mock-infected group. This result could point towards a higher reluctance to perform movements or prolonged activity which could be an indicator of exercise intolerance reported in humans with respiratory long-COVID (Supplementary Fig. 1).

Interestingly, after mild exercise, SARS-CoV-2-infected animals showed a lower breathing frequency compared to the mock-infected group until around 7 weeks after infection. A mild significant increase of TV, a decrease of EF50, and an increase of Te was observed in SARS-CoV-2-infected animals compared to controls around similar time-points (Fig. 1c). No Ti changes were observed. v$O_2$, v$CO_2$, and MR showed marked and significant differences from 21 dpi to 49 or 56 dpi. Differences in mechanical and metabolic values became less prominent around 56 dpi (Fig. 1c) and were no longer detectable thereafter. Of note, the lack of differences in metabolic values after 56 dpi was not caused by a recovery of SARS-CoV-2-infected animals, which showed lower values as compared to pre-infection until the end of the experiment, but rather caused by the gradual decrease of vO2, vCO2 and MR in mock-infected animals starting at 63 dpi (Fig. 1c). This decrease was closely associated with a gradual body weight loss starting at the same time in these animals and was considered related to aging.

In summary, these results demonstrate that SARS-CoV-2-infected hamsters in the chronic phase i) do not recover their pre-infection body weight, ii) show lung function alterations after exercise persisting up to 7 weeks after infection iii) reduced running behavior in the treadmill, and iv) reduced metabolic activity values at rest, that become more pronounced after exercise. These findings are in line with changes in lung function and metabolic parameters observed in long-COVID patients after physical exercise[37,41–43], which provides further evidence that the hamster is a suitable model for studying long-term pulmonary effects of SARS-CoV-2 infection. Our next aim was to characterize the underlying pathomorphological and transcriptomic changes associated with these findings.

## RNA-sequencing analysis reveals distinct transcriptomic signatures in acute, sub-acute, and chronic SARS-CoV-2 Delta infection in aged hamsters

In order to gain insight into transcriptomic changes associated with sub-acute and especially long-term functional sequelae of SARS-CoV-2 infection, we performed bulk RNA-sequencing (RNAseq) of lung tissues at 1, 3, 6, 14, 28, 56, and 112 dpi. Principal component analysis (PCA) of gene expression levels distinctly separated mock-infected and SARS-CoV-2-infected hamsters. The most pronounced response to infection was observed at 6 dpi. Moreover, SARS-CoV-2-infected animals in the acute phase (1, 3, and 6 dpi) of the disease clearly separated from animals in the sub-acute (14 dpi) and chronic stages (28, 56, and 112 dpi; Fig. 2a).

Pairwise comparison between mock- and SARS-CoV-2-infected animals revealed the presence of a variable number of up- and down-regulated differentially expressed genes (DEGs) at all time points. Transcriptomic changes indicated a strong response to infection in the acute phase, with 75, 2020, and 3007 DEGs at 1, 3, and 6 dpi, respectively. This response was lower in the sub-acute phase (566 DEGs at 14 dpi) and further decreased throughout the chronic phase (587, 56, and 22 DEGs at 28 dpi, 56 dpi, and 112 dpi, respectively).

Notably, 6 dpi was the time-point with the highest number of DEGs with 1340 upregulated and 1667 downregulated (Fig. 2b, c; Supplementary data 2). Pathway enrichment analysis showed that at 6 dpi, most of the upregulated genes were involved in immune cell proliferation, activation, regulation, and signaling (Fig. 2d). Most of the down-regulated genes play a role in angiogenesis, extracellular matrix organization, and regulation of nervous system development (Fig. 2e). It should be noted that RNAseq analysis performed in this study was made primarily to validate and support our morphological findings, which will be elaborated in detail in the subsequent sections. For this reason, an in-depth analysis of all DEGs and pathways at all time points was not performed. However, the full dataset is accessible to the research community for any additional analyses (see data availability statement).

For the sub-acute and chronic phases of the disease, we focused on DEGs that were unique to each time-point. 504, 534, 22, and 18 DEGs at 14, 28, 56, and 112 dpi, respectively (Fig. 3 a).

Remarkably, among the DEGs unique to 14 dpi, several genes encoding collagen were identified, including: *Col3a1, Col11a1, Col17a1, Col1a2, Col1a1*. In addition, there were genes encoding for cytokeratin 14 (*Krt14* gene), mainly expressed by airway basal cells[44], as well as *Scgb3a2, Scgb3a1* genes mainly expressed by club cells[45]. Also, among the 28 dpi unique DEGs were genes encoding for collagen production (i.e., *Col6a3, Col6a5, Col4a2, Col4a3, Col4a1*). Similarly, at 56 dpi, among the unique DEGs were genes involved in extracellular matrix remodeling/deposition (i.e.*Col9a1, Col2a1, Adam12*)[46] as well as receptors for GABA-A (*Gabrp*) and ephrin (*Ephb1*). Most importantly, *Sox9*, which is reported to play multiple roles in the lung epithelium, balancing proliferation and differentiation and regulating the extracellular matrix[47], was a gene that was differentially expressed exclusively at 56 dpi. Among the 112 dpi unique DEGs were genes like *Mmp9* and *Mmp25*,

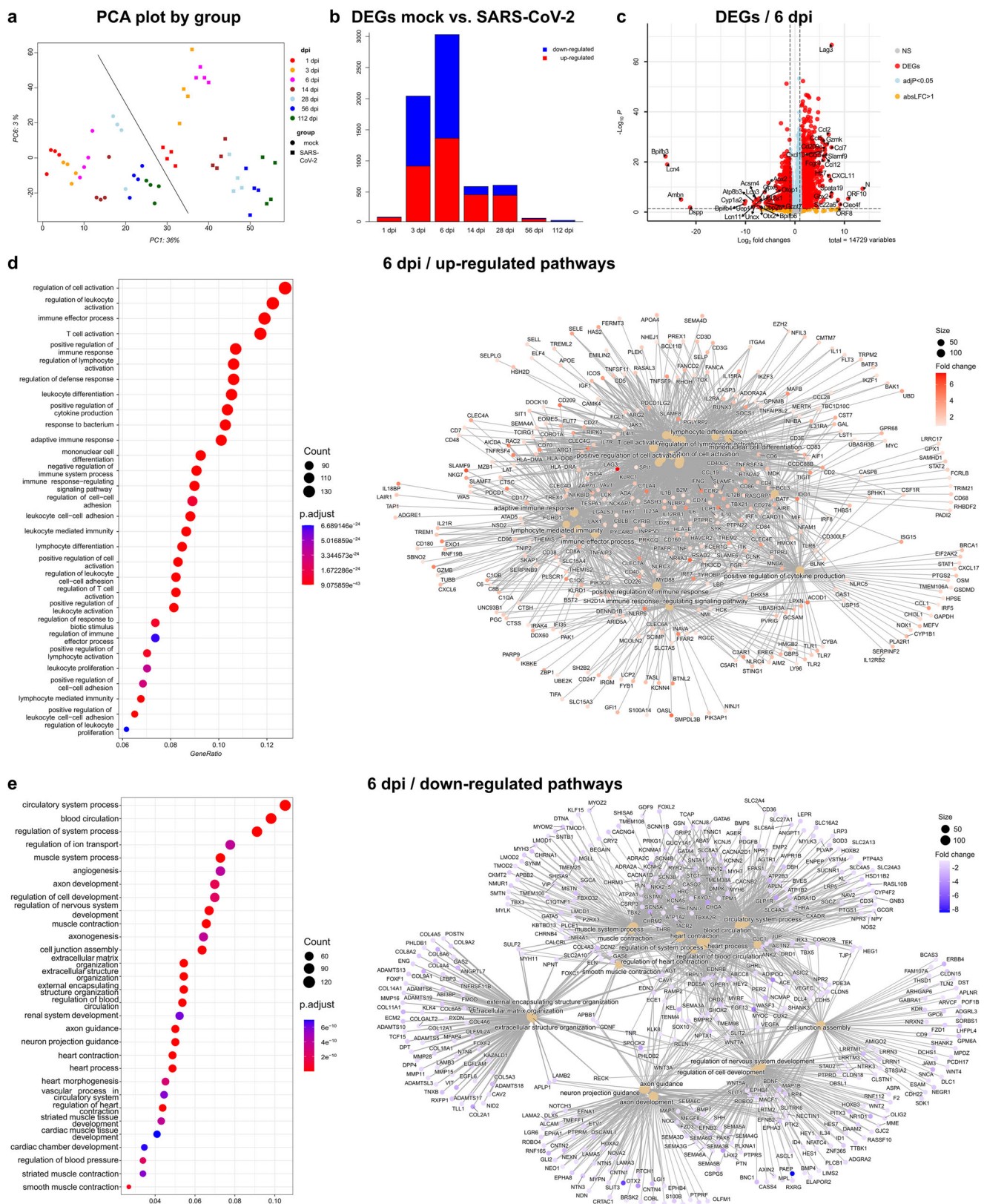

**a** PCA plot by group

**b** DEGs mock vs. SARS-CoV-2

**c** DEGs / 6 dpi

**d** 6 dpi / up-regulated pathways

**e** 6 dpi / down-regulated pathways

which are involved in extracellular matrix remodeling/deposition[48]. Other interesting genes upregulated at 112 dpi were *S100a8 and S100a9*. These genes encode for proteins involved in the pathogenesis of cystic fibrosis and COVID-19[49,50]. A full list of up- and down-regulated DEGs at the given time points is available in the supplementary material (Supplementary data 3).

Pathway enrichment analysis of the DEGs unique to 14 dpi showed that some of the genes were involved either in the modulation of chemical synaptic transmission or in the regulation of trans-synaptic signaling (Fig. 3b). At 28 dpi, some unique DEGs belonged to small GTPase-mediated signal transduction, cell-cell adhesion via plasma-membrane adhesion molecules, and homophilic cell adhesion via

**Fig. 2 | RNA-sequencing analysis reveals the strongest host response at 6 dpi in SARS-CoV-2-infected aged hamsters.** RNAseq analysis in lungs of SARS-CoV-2-infected hamsters at 1, 3, 6, 14, 28, 56, and 112 days post infection (dpi). **a** Principle Component Analysis (PCA) showing PC1 and PC6, which were correlated with infection. Mock- and SARS-CoV-2-infected animals are represented by dots and squares, respectively. Different colors represent different dpi. **b** Bar plot of the number of differentially expressed genes (DEGs) in SARS-CoV-2- versus mock-infected animal per dpi. Upregulated DEGs are shown in red and down-regulated in blue. **c** Volcano plot showing the DEGs at 6 dpi. $y$ axis: $-\log 10$ multiple testing adjusted $p$ values, $x$ axis: $\log_2$-fold change. DEGs (absolute $\log_2$-fold change >1 and adjusted $p$ value < 0.05) are colored red and the top 20 up- and down-regulated (by

$\log_2$-fold change) DEGs are labeled. Blue: genes with adjusted $p$ value < 0.05. Yellow: genes with absolute $\log_2$-fold change >1. **d, e** Cluster profiler of EnrichGO pathway analysis for up- and downregulated DEGs from the contrasts of SARS-CoV-2- versus mock-infected controls at 6 dpi. Left panel: Dot plots of up- (in **e**) and down- (in **d**) regulated pathways. The $y$ axis shows the enriched pathways and the $x$-axis shows the GeneRatio (gene-count/total number of genes). Dot size indicates the number of genes and color indicates the $p$ value. Right panel: gene-concept network depicting the linkages of genes and pathways of the genes involved in the up- (in **e**) and down- (in **d**) regulated pathways at 6 dpi. The dot size of the labelled pathways indicates the size of the gene count belonging to that pathway, and the colour indicates the fold change.

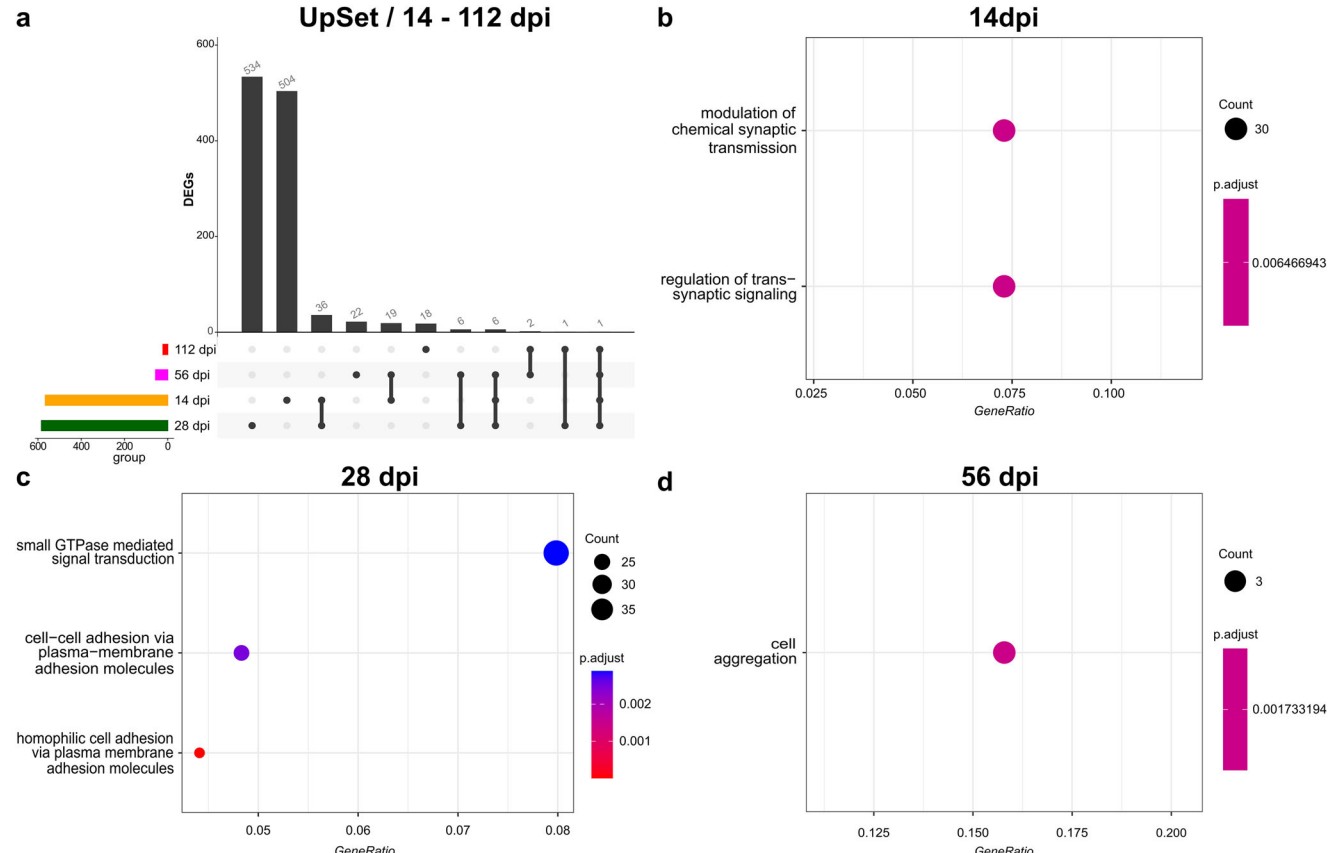

**Fig. 3 | Bulk RNA-sequencing analysis reveals the presence of distinct DEGs in the sub-acute and chronic phases of the disease. a** UpSet plot of unique differentially expressed genes (DEGs) and overlapping DEGs at different days post infection (dpi) from the sub-acute and chronic phases (14, 28, 56, 112 dpi). The total number of DEGs associated with each dpi is depicted in the colored histogram to the left. The scheme below the boxplots depicts unique or overlapping numbers of DGEs at a given dpi. Connected dots represent DEGs shared by two or more dpi. The number of unique or overlapping DEGs is shown on the top of the bars. **b**–**d** Cluster profiler of EnrichGO pathway analysis for unique DEGs at 14, 28, 56 dpi shown in the Upset plot. The $y$ axis shows the enriched pathways, and the $x$ axis shows the GeneRatio (gene-count/total number of genes). Dot size indicates the number of genes and color change indicates the $p$ value.

plasma-membrane adhesion molecules (Fig. 3c). In contrast, only three 56 dpi unique DEGs belonged to cell aggregation (Fig. 3d).

Notably, 14 dpi shared 36 DEGs and 19 DEGs with 28 dpi and 56 dpi, respectively. All three-time points together shared 6 DEGs. Similarly, 28 dpi and 56 dpi shared 6 DEGs. On the other hand, 112 dpi only shared 2 DEGs with 56 dpi, 1 DEG with 28 dpi, and only 1 DEG with 14, 28, and 56 dpi (Fig. 3a). In particular, among the 36 shared DEGs between 14 dpi and 28 dpi were multiple genes involved in neuronal growth, plasticity and signaling, such as *Slc17a7*[51], *Tagln3*[52], *Plppr1*[51], *Th*[53], *Cam2kb*[54], and *Grin1*[55]. Additionally, there were genes involved in extracellular matrix remodeling/deposition like *Adamts4* and *Lox* or in cell migration like *Map7d2*[56]. Among the 19 shared DEGs between 14 dpi and 56 dpi was a gene involved in the negative regulation of Notch signaling (*Dlk2*)[57], a gene encoding for a protein associated with collagen (*Emilin3*)[58], a gene that promotes the cell

cycle (*Wee1*)[59] and most importantly *Krt 15*, which is reported to be also expressed by airway basal cells[44]. Among the 6 DEGs shared by 14 dpi, 28 dpi, and 56 dpi there was a gene encoding for a member of the cytochrome P450 superfamily (*Cyp2e1*)[60] and a gene involved in the regulation of interferon-stimulated gene expression (*Arntl*)[61,62]. Interestingly, DEGs shared by 28 dpi and 56 dpi were *Fat2*, involved in cell adhesion and cell proliferation[63], and *Col9a3*, encoding for type IX collagen[64].

Overall, transcriptome analysis reveals the presence of distinct sets of DEGs across all three phases of SARS-CoV-2 infection in hamsters. The strength of the host response in terms of DEGs confirmed the three phases of the disease (acute, sub-acute, and chronic). In the acute phase, transcriptomic changes were dominated by immune cell proliferation, activation, regulation, and signaling, while in the sub-acute and chronic phase, modulation of chemical

synaptic transmission, extracellular matrix remodeling/deposition, cell migration, and cell aggregation were more prominent. Our transcriptome data also supported many of our pathomorphological findings and their interpretations elaborated below in the following sections.

### SARS-CoV-2 Delta infection induces transient inflammation and chronic lesions with persistence of viral RNA in the lungs of aged hamsters

Infection was confirmed by immunohistochemical staining and quantification of SARS-CoV-2 nucleoprotein (NP) antigen in the lung. As described previously[18], viral antigen was present in the alveolar and airway epithelial cells as well as in alveolar macrophages (Supplementary Fig. 1). Positive cells were detected as early as 1 dpi with the highest number of positive cells at 3 dpi. At 14 dpi, no viral protein was detectable (Supplementary Fig. 1). Similar results were obtained by the immunohistochemical staining and quantification of SARS-CoV-2 spike protein (SP, (Supplementary Fig. 1). Mock-infected animals showed no viral protein at any time-point. Additionally, RNAseq revealed no SARS-CoV-2 spike (S) and nucleoprotein (N) transcripts at any time-point in the mock-infected animals. Both SARS-CoV-2 *S* and *N* gene transcripts peaked at 1 dpi, were marginally present until 6 dpi, and were not detected in samples from the following time-points in SARS-CoV-2-infected hamsters (Supplementary Fig. 1). There is growing evidence that either replication competent virus or viral products can persist in the lung of COVID-19 patients and SARS-CoV-2 animals beyond the seeming recovery from disease[65,66]. Moreover, it has been reported in hamsters that pulmonary viral persistence is associated with chronic weight loss[67]. Since we observed that SARS-CoV-2-infected hamsters did not recover their initial body weight, we additionally performed PCR for viral RNA-dependent RNA polymerase (RdRp) and subgenomic viral RNA (Esub) in the lung of mock- and SARS-CoV-2-infected animals from 28 dpi onwards. Interestingly, RdRp RNA was detected in all animals at 28 dpi, 6/7 animals at 56 dpi, and 6/8 animals at 112 dpi. Esub RNA was present in 6/8 animals at 28 dpi, but not detected at later time-points (Supplementary Fig. 1). Interestingly, the animals with detectable viral RdRp RNA in the lung during the chronic phase were the ones with the most severe weight loss and/or the ones that showed the least recovery of initial body weight (Supplementary Data 1).

One of the most prominent pathomorphological features of SARS-CoV-2 infection was marked, but transient, broncho-interstitial pneumonia as described previously[18,31]. Infection-associated histological lesions were present from 1 dpi on, and severity peaked at 6 dpi. From 28 dpi onwards the histopathological lung lesion score was low in almost all animals but still present. The lesions in the chronic phase were mostly characterized by alveolar proliferation areas and sub-pleural fibrosis, while inflammation was minimal (Fig. 4a, b; Supplementary Data 4).To quantify the extent of inflammation in both groups, Iba-1-, MPO+-, CD3+-, and Pax5+-cells were quantified in immunolabelled whole left lung lobe sections. Significantly higher numbers of Iba-1+ histiocytic cells were detected in SARS-CoV-2-infected hamsters from 1 dpi to 28 dpi, with the highest numbers observed at 6 dpi. MPO+ and CD3+ cell numbers were significantly higher at 6 dpi. On the other hand, significantly higher numbers of Pax5+ cells were found from 6 to 28 dpi (Supplementary Fig. 2). Transcriptome data analysis revealed an increased expression of *Cd8a*, *Cd3d*, *Cd3e*, *Cd4*, *Alf1* (Iba-1) genes in SARS-CoV-2-infected hamsters at 6 dpi. In addition, CybersortX deconvolution showed M1-like macrophage signature peaking at 6 dpi confirming this time-point as the most active from an inflammatory point of view (Supplementary Fig. 2).

While inflammatory infiltrates were not prominent in the chronic lesions of SARS-CoV-2-infected hamsters, regenerative changes were present throughout the investigation period. It has been demonstrated

in a murine model of influenza A virus (IAV) pneumonia that an important mediator of macrophage-epithelial cross-talk for alveolar proliferation is Placenta-expressed transcript-1 (Plet-1)[68]. Quantification of Plet-1 immunolabelling in the lungs of hamsters revealed significantly higher numbers of Plet-1+ cells in SARS-CoV-2-infected compared to mock-infected hamsters at 3, 6, and 14 dpi. Interestingly, transcriptome data analysis revealed an increased expression of *Mertk* and *Plet1* at 6 dpi, while at 14 and 28 dpi only *Plet1* was upregulated (Supplementary Fig. 3).

From 6 dpi onwards, there were moderate to marked, multifocal to coalescing areas of epithelial hyperplasia with occasional atypical cells, that were particularly prominent in areas of bronchioloalveolar junctions (Fig. 4a). These alveolar epithelial proliferation foci were present with variable extent in 8/8 animals at 6 and 14 dpi, 7/8 animals at 28 dpi, 7/7 animals at 56 dpi, and 4/8 animals at 112 dpi (Fig. 4b, Supplementary Data 4). Particularly at the late time-points, these alveolar epithelial proliferation foci were present throughout the lung and were characterized by an airway-like cell morphology, consistent with bronchiolization (Fig. 4a). Another striking histopathological feature was sub-pleural fibrosis. This lesion was present with variable extent in 1/8 animals at 6 dpi, 6/8 animals at 14 dpi, 3/8 animals at 28 dpi, 1/7 animals at 56 dpi, and 2/8 animals at 112 dpi (Fig. 4c; Supplementary Data 4).

In conclusion, SARS-CoV-2 infection of aged male hamsters resulted in persistent architectural changes of the lung tissue, characterized by bronchiolization and fibrosis, that were associated with persistence of viral RNA until the end of the investigation period. In the following, we aimed to further characterize the fibrosis and alveolar epithelial proliferation foci. Due to the persistence of these lesions, we speculated that they might be associated with long-lasting functional impairment.

### SARS-CoV-2 Delta infection induces pulmonary interstitial fibrosis and a pro-fibrotic transcriptomic signature up to 112 days post infection in aged hamsters

To further analyze pulmonary fibrosis in SARS-CoV-2-infected animals, we performed quantification of Azan-positive areas in the entire alveolar interstitium and investigated features of a pro-fibrotic environment. Besides the more obvious focal areas of sub-pleural fibrosis, we speculated that more subtle increases in extracellular matrix deposition within alveolar septa could also contribute to impaired lung function. Thus, we used the Azan stain to visualize collagen and performed a digital quantification of the positive area on entire lung sections. Analysis of the total Azan+ area in the whole lung tissue revealed a statistically significant increase of collagen deposition in alveolar septa of SARS-CoV-2-infected animals compared to mock-infected animals from 14 dpi to 112 dpi (Fig. 4d, e). A pro-fibrotic environment in SARS-CoV-2-infected mice[69] and hamsters[18] seems to be associated with the presence of CD204+ M2-like macrophages. Immunohistochemistry showed a notable rise in CD204+ macrophages in the whole lung of SARS-CoV-2-infected hamsters compared to mock-infected animals from 6 to 56 dpi. In the next step, we investigated the expression of genes associated with a pro-fibrotic environment (Supplementary Table 1), which are known to be upregulated in COVID-19 patients[70,71]. This analysis revealed that genes involved in extracellular matrix remodeling (*Mmp12, Mmp14, Tgfb1*, Supplementary Fig. 4) were upregulated at early time points (3 dpi and 6 dpi), while mostly collagen-encoding genes (*Col5a1*, *Col1a1*, *Col3a1*, Supplementary Fig. 4) were upregulated at later stages of the disease (Fig. 4g), suggestive of a long-lasting increase of collagen production.

After showing that SARS-CoV-2 Delta variant-infected hamsters exhibited interstitial pulmonary fibrosis starting from 14 dpi and that this was supported by a pro-fibrotic transcriptome environment, we aimed to examine alveolar epithelial proliferation foci, with a specific focus on ADI cells as well as airway basal cells and club cells.

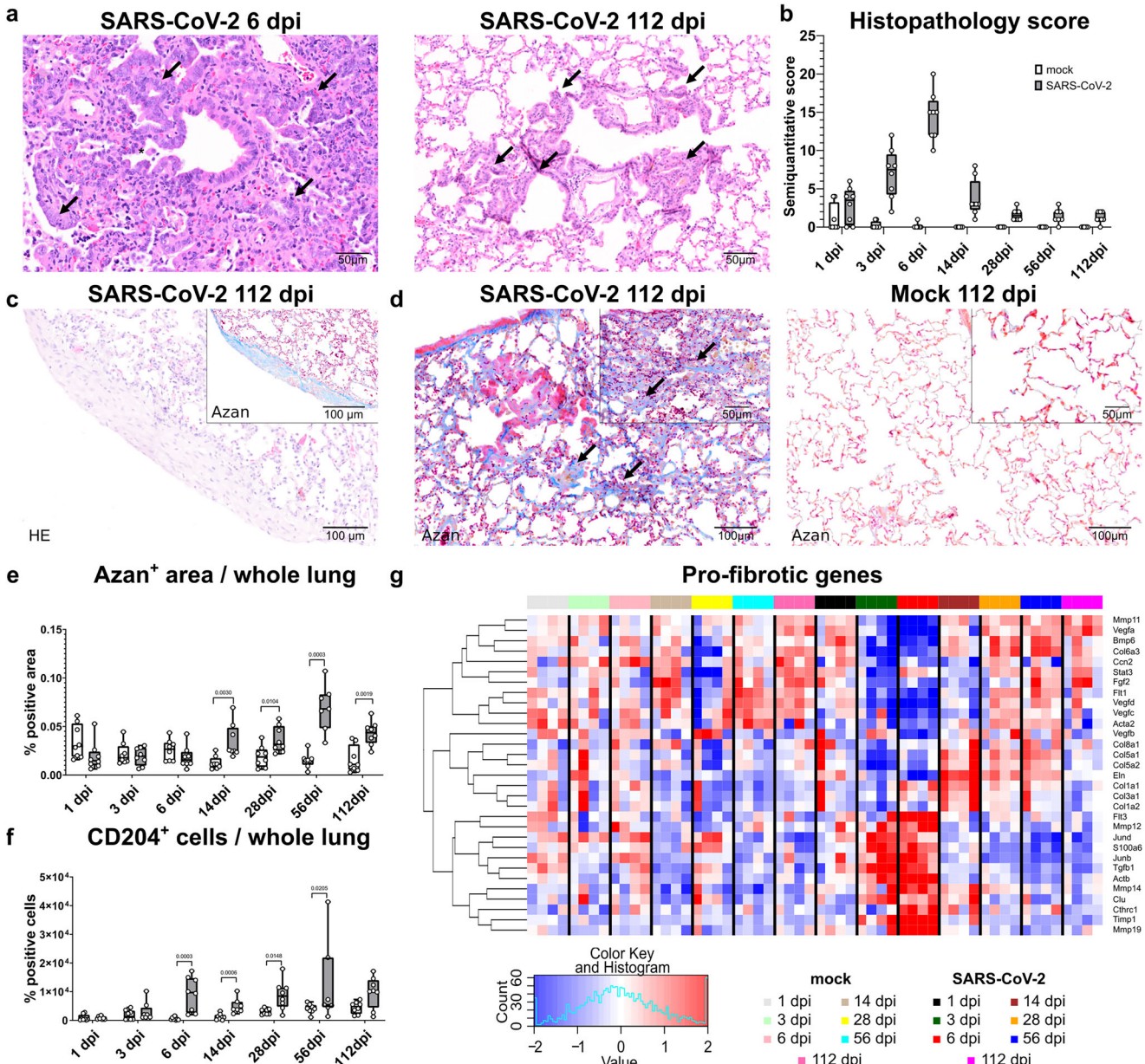

**Fig. 4 | SARS-CoV-2 infection causes persistent, sub-pleural, and interstitial fibrosis in aged hamsters. a** Representative images showing epithelial proliferation in the left lung lobe of SARS-CoV-2-infected hamster at 6 days post infection (dpi, left) and 112 dpi (right) on hematoxylin and eosin (HE) stained sections. At 6 dpi, there is alveolar consolidation and inflammation, with prominent epithelial proliferation around a central bronchiole (arrows), with a cuboidal, airway-like morphology, typical of bronchiolization. At 112 dpi, inflammation is absent, but areas of alveolar bronchiolization persist throughout the lung (arrows). **b** Semi-quantitative scoring of histopathological lung lesions in mock- and SARS-CoV-2-infected hamsters. **c, d** Representative images showing sub-pleural and interstitial fibrosis in the left lung lobe of a SARS-CoV-2-infected hamster at 112 dpi.
**c** Representative image of well-demarcated area of sub-pleural fibrosis on HE stained section, characterized by aggregates of mesenchymal cells and pale eosinophilic, fibrillary, extracellular matrix. Inset: same area stained with azan, demonstrating the presence of mature collagen fibers (blue staining).

**d** Representative images of azan-stained sections showing an increased presence of mature collagen fibers (blue staining) in the thickened alveolar septa (arrows) of SARS-CoV-2-infected animals compared to mock-infected ones. **e** Digital quantification of azan-positive areas in relation to the whole lung tissue in mock- and SARS-CoV-2-infected hamsters. **f** Quantification of CD204+ M2-like macrophages in total lung area of mock- and SARS-CoV-2-infected hamsters. **g** Heatmap of normalized expression values for genes associated with a pro-fibrotic environment (Supplementary Table 1) at each dpi in mock- and SARS-CoV-2-infected hamsters. Expression values are scaled by row. Red indicates higher and blue lower relative expression levels. **b, e, f** Data are shown as box and whisker plots. The bounds of the box plot indicate the 25th and 75th percentiles, the bar indicates medians, and the whiskers indicate minima and maxima. Dots indicate individual values. Data from **e, f** was tested by two-tailed Mann–Whitney U test. A p value of ≤0.05 was chosen as the cutoff for statistical significance. N = 8 animals/group (**b, e, f**) or 4 animals/group (**g**). Source data is provided as a Source Data file.

## After 28 dpi the presence of ADI cells in lungs of SARS-CoV-2 Delta-infected aged hamsters is markedly decreased

In our previous work, we demonstrated that alveolar regeneration in SARS-CoV-2-infected hamsters involves persistence of ADI cells until 14 dpi, which is believed to play a role in the pathogenesis of protracted recovery and lung fibrosis[18]. In the following, we focused on the detection of ADI cells in the acute, sub-acute, and chronic phases of the disease.

CK8+ ADI cells were identified by immunohistochemistry. The apical cytoplasm of luminal cells within bronchi, bronchioles, and

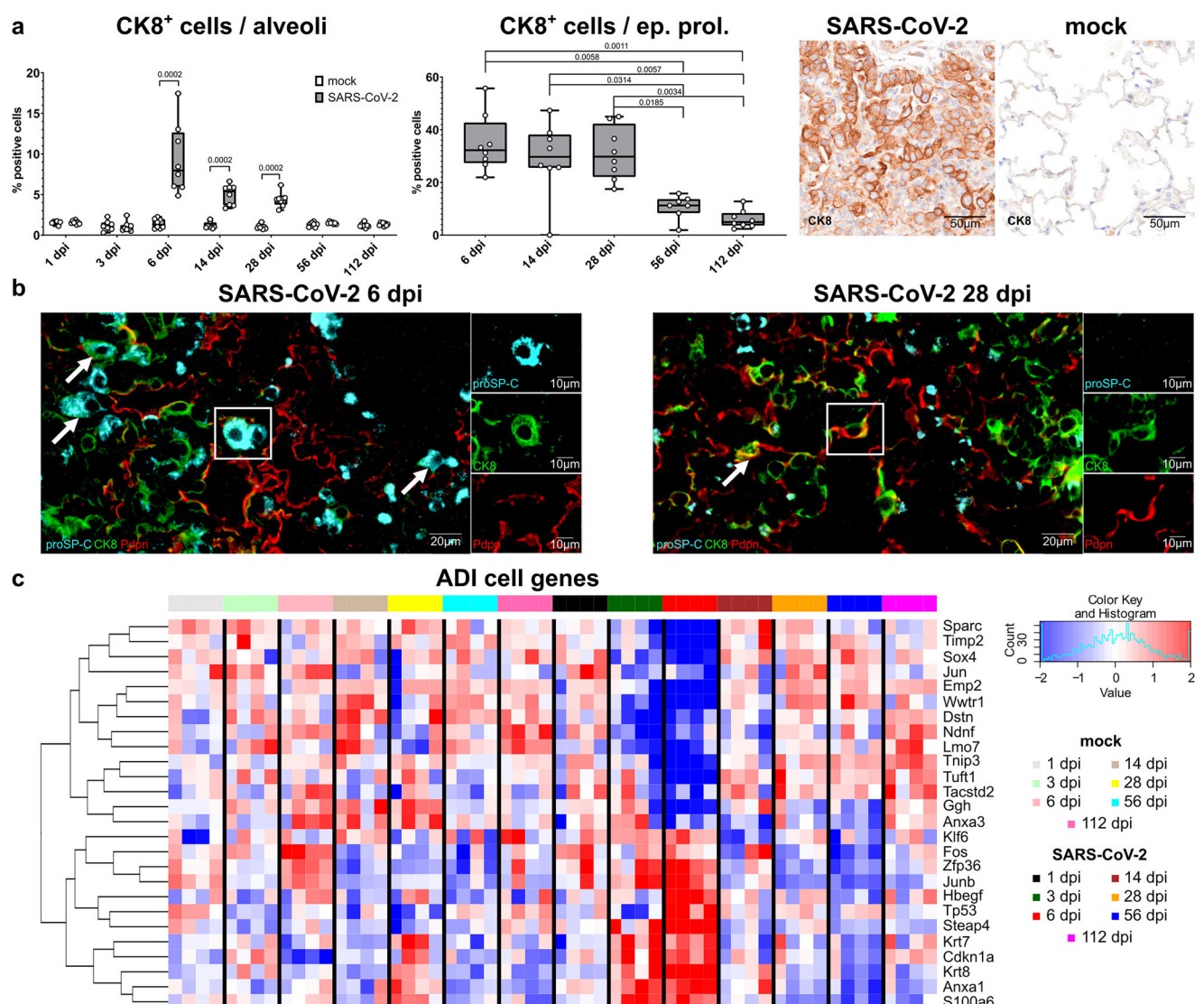

**Fig. 5 | Alveolar differentiation intermediate (ADI) cells in SARS-CoV-2-infected hamsters. a** Quantification of cytokeratin 8 (CK8)⁺ alveolar differentiation intermediate (ADI) cells within the whole alveolar space and within alveolar epithelial proliferation (ep. prol.) foci. On the right, representative pictures of immunolabeling (brown signal) in the alveoli of SARS-CoV-2- and mock-infected hamsters at 6 days post infection (dpi). **b** Representative image of triple immunofluorescence in an alveolar proliferation focus at 6 (left panel) and 28 (right panel) dpi. Cells are labeled with CK8 (green), pro-Surfactant protein C (proSP-C, light blue), and Podoplanin (Pdpn, red). In the left picture at 6 dpi there are either round proSP-C⁺CK8⁻PDPN⁻ or proSP-C⁺CK8⁺PDPN⁻ cells (arrows and inlet) in the alveolar proliferation focus. In the right picture at 28 dpi there are moderate numbers of elongated proSP-C⁻CK8⁺PDPN⁺ (arrow and inlet) in the alveolar proliferation focus. Scattered proSP-C⁻CK8⁺PDPN⁻ and numerous proSP-C⁻CK8⁻PDPN⁺ cells are seen at

both time points. The staining was performed in four animals/time points with the same results. **c** Heatmap of normalized expression values for genes associated with ADI cells (Supplementary Table 1) at each dpi in mock- and SARS-CoV-2-infected hamsters. Expression values are scaled by row. Red indicates higher and blue lower relative expression levels. **a** Data are shown as box and whisker plots. The bounds of the box plot indicate the 25th and 75th percentiles, the bar indicates medians, and the whiskers indicate minima and maxima. Dots indicate individual values. Statistical analysis was performed by a two-tailed Mann–Whitney $U$ test. For multiple comparisons between time points (quantification within alveolar proliferation foci), a Benjamini–Hochberg correction was applied. $P$- and $q$ values ≤ 0.05 were considered significant. $N$ = 8 animals/group (**a**) or 4 animals/group (**c**). Source data is provided as a Source Data file.

terminal bronchioles in all animals expressed CK8 ubiquitously. Rare CK8⁺ cells were observed in the alveoli of mock-infected animals, whereas CK8 was abundantly expressed within alveolar epithelial proliferation foci in SARS-CoV-2-infected animals. In the whole alveolar space, there were significantly higher percentages of CK8⁺ cells in SARS-CoV-2-infected hamsters compared to mock-infected animals from 6 dpi until 28 dpi (Fig. 5a). The highest percentages of CK8⁺ ADI in the whole alveolar spaces were detected at 6 dpi (16% of total cells). Digital quantification of immunolabeled cells exclusively in areas of alveolar epithelial proliferation revealed that CK8⁺ ADI cells peaked at 6 dpi with a maximum of 55.72% of the total cell population. By 28 dpi,

CK8⁺ ADI cells still constituted 31.10% of the total cell count in these areas on average, whereas at 56 dpi and 112 dpi average percentages dropped down to 10.68% and 6.06%, respectively (Fig. 5a).

Subsequently, to complete our investigation of ADI cells and the alveolar epithelial cell type 2 (AT2) -ADI- alveolar epithelial type 1 (AT1) trajectory, different stages of ADI cells were identified using triple-labeling immunofluorescence for proSP-C (AT2), CK8 (ADI), and PDPN (AT1) as shown in Fig. 5b. Briefly, at 3 dpi, numerous round proSP-C⁺CK8⁻PDPN⁻ cells and occasional proSP-C⁺CK8⁺PDPN⁻ cells were observed within epithelial proliferation areas, whereas proSP-C⁻CK8⁺PDPN⁺ elongated cells were very rare. At 6 dpi, epithelial

proliferation foci contained occasional proSP-C⁺CK8⁻PDPN⁻ or proSP-C⁺CK8⁺PDPN⁻ round cells and moderate to numerous proSP-C⁻CK8⁺PDPN⁻ cells with typical ADI morphology. Moreover, rare elongated proSP-C⁻CK8⁺PDPN⁺ cells with AT1 morphology were also detected. From 6 dpi to 28 dpi, the proSP-C⁻CK8⁺PDPN⁻ ADI cells became less prominent, and more of the late, elongated AT1- like proSP-C⁻CK8⁺PDPN⁺ were present.

To further support our morphological findings about ADI cells, we subsequently investigated the expression of ADI cell-related genes on the transcriptome level based on the hamster ADI cells-specific gene list we previously generated (Supplementary Table 1)[18]. The heatmap Fig. 5c shows that at 3 and 6 dpi, there is an upregulation of many ADI-specific genes, especially genes typical of ADI cells in the full state after differentiation from AT2 or club cells[72], including *S100a6, Krt8, Anxa1, TpS3*, and *Hbegf* (Fig. 5c; Supplementary Fig. 5). From 14 dpi onwards, a variable downregulation of these genes was observed. At the same late time-points, a different subset of genes, typical of the late stage of ADI cells, closing the AT2-ADI-AT1 trajectory[72] was variably upregulated over time, for instance, *Sparc, Sox4*, and *Wwtr1* (Fig. 5c; Supplementary Fig. 5).

Taken together, our data showed that ADI cells were a prominent feature of alveolar regeneration processes in SARS-CoV-2 Delta variant-infected aged hamsters during the acute and sub-acute phases of the disease. However, their presence does not appear to be so prominent in the late stage of the disease. Based on the fact that, upon severe damage, alveolar regeneration is aided by airway progenitor cells[18,71–75], we focused our next analysis on characterizing airway progenitors in the alveolar compartment.

## Multipotent airway-derived CK14+ progenitors and SCGB1A1+ club cells persist in alveolar epithelial proliferation foci up to 112 dpi in SARS-CoV-2 Delta-infected aged hamsters

In our previous work, we demonstrated that CK14⁺ basal cells are the predominant basal cell type in the airways of hamsters, followed by CK14⁺ΔNP63⁺ cells, whereas CK14⁺ ΔNP63⁺CK5⁺ cells are rare in the distal airways. We also demonstrated that following SARS-CoV-2 infection, SCGB1A1⁺cells and CK14⁺ cells are the main airway progenitors in alveolar proliferation foci that contribute to alveolar regeneration. Finally, we postulated that CK14⁺ basal cells seem to be the hamster equivalent of CK5⁺ basal cells reported in humans and mice[18]. Therefore, in the present study, we investigated the number of cells expressing CK14 and SCGB1A1 in the airways, in the whole alveolar space, as well as in the alveolar epithelial proliferation foci of bronchiolization during the acute, sub-acute, and chronic phase of the disease.

The number of CK14⁺ basal cells in the airways of SARS-CoV-2-infected animals was significantly higher, from 3 to 28 dpi, compared to the mock-infected ones. However, the number of SCGB1A1⁺ club cells did not have any significant variation in the airways upon infection (Fig. 6a, b). In the whole alveolar space, the percentage of CK14⁺ cells was significantly higher in SARS-CoV-2-infected hamsters compared to mock-infected animals from 3 dpi until 112 dpi, with a peak at 6 dpi. The percentage of SCGB1A1⁺ cells in the total alveolar space was significantly higher from 14 dpi to 112 dpi, with a steady level throughout the entire period (Fig. 6a, b). Next, we aimed to further characterize the cellular composition of the alveolar proliferation foci, consistent with persistent areas of bronchiolization, in more detail. Therefore, we quantified CK14⁺ basal cells, SCGB1A1⁺ club cells, CK5⁺ basal cells, ΔNP63⁺ basal cells, Mucin-5AC (MUC5AC) and Mucin-5B (MUC5B)⁺ secretory cells exclusively in these regions. CK14⁺ basal cells were elevated at 6 and 14 dpi, representing ~40% of the total cell count in these bronchiolization areas. This proportion decreased gradually to ~10% at 112 dpi (Fig. 6a). Notably, SCGB1A1⁺ club cell numbers increased at 14 dpi, later than any other cell type within these alveolar epithelial proliferation foci. Moreover, they persisted until 112 dpi, consistently constituting approximately over 40% of the total cell count (Fig. 6b). ΔNP63⁺ basal cells represented less than 0.6% of the total cell count at all time points (Fig. 6d). Immunolabeling

for CK5, MUC5AC, and MUC5B did not detect any cells within the alveolar epithelial proliferation at any time-point. This suggests that these cell types are not involved in the alveolar regeneration after SARS-CoV-2 infection in hamsters.

In our previous work, we postulated the multipotency of CK14⁺ basal cells by demonstrating the co-expression of CK14 and AT2 (proSP-C), ADI cell (CK8) and club cell (SCGB1A1) markers in the alveolar epithelial proliferation areas[18]. In this current work, considering the prominent presence of SCGB1A1⁺ club cells compared to the CK8⁺ ADI cells at the later time-points, we mainly focused on the club cell differentiation.

Double-labeling immunofluorescence showed that there are moderate numbers of CK14⁺ SCGB1A1⁺ cells in the epithelial proliferation areas at 14 and 28 dpi. This number of double-positive cells is constant over time with low to moderate numbers of CK14⁺ SCGB1A1⁺ cells in the epithelial proliferation areas until 112 dpi. Nevertheless, CK14⁻ SCGB1A1⁺ cells were the most prominent cells at all time points (Fig. 6c). These results are supportive of the hypothesis that SCGB1A1⁺ club cells seen in the alveolar spaces also originate from a pool of CK14⁺ basal cells even at the later time-points.

To further support our morphological findings concerning the contribution of airway progenitors to alveolar regeneration processes, we subsequently investigated the expression of club cell- and airway basal cell-related genes based on our hamster club cell-specific gene list (Supplementary Table 1)[18] and literature[70,72]. The heatmap in Fig. 6e, f shows that among club cell genes, *Gss* and *Pigr* were the only two with an upregulation at the later time-points, whereas all the other genes showed a variable upregulation at different time-points of the acute and sub-acute phase. Similarly, airway basal cell genes like *Ngfr* or *Pou2f3* were variably upregulated at the later time-points. After peaking at 6 dpi, *Krt14* showed a slight decrease but remained upregulated until the later time-points. *Krt5* expression was variably stable over time (Fig. 6e; Supplementary Fig. 6).

In summary, airway progenitors, in particular CK14⁺ basal cells and SCGB1A1⁺ club cells, dominated the cellular composition within the alveolar epithelial proliferation areas in the sub-acute and chronic phase after SARS-CoV-2 Delta infection. In particular, SCGB1A1⁺ club cells were the predominant cell type in the bronchiolization foci persisting until 112 dpi.

## CK14+ airway basal cells and SCGB1A1+ club cells express proliferation markers in the alveolar bronchiolization areas until 112 dpi

Next, we wanted to assess the expression of proliferation and cell cycle arrest markers in the alveolar epithelial proliferation areas.

Ki67⁺ cells peaked at 6 dpi, representing ~40% of the total cell count in these areas. At the following time-points, this proportion decreased to ~10% at 112 dpi (Fig. 7a). Despite representing only about 1% of the total cell count in these areas, p53⁺ cells were also most present at 6 dpi. At the subsequent time-points, only 2 animals still showed p53⁺ cells at variable percentages (Fig. 7b). A similar trend was observed with p21⁺ cells (Fig. 7c).

Afterwards we wanted to visualize which cell types were expressing proliferation markers. At 6 dpi, moderate numbers of CK8⁺Ki67⁺ ADI cells and CK14⁺Ki67⁺ airway basal cells were present in the alveolar epithelial proliferation areas, whereas neither SCGB1A1⁺Ki67⁺ nor SCGB1A1⁺Ki67⁻ club cells were observed. At 14 and 28 dpi, there were minimal to low numbers of CK8⁺Ki67⁺ ADI cells. CK14⁺Ki67⁺ airway basal cells, and SCGB1A1⁺Ki67⁺ club cells were present at low to moderate numbers. Notably, at 56 dpi and 112 dpi there were no CK8⁺Ki67⁺ ADI cells but moderate numbers of CK14⁺Ki67⁺ airway basal cells and SCGB1A1⁺Ki67⁺ club cells in persistent bronchiolization areas. (Fig. 7d).

In conclusion, CK14⁺ basal cells and SCGB1A1⁺ club cells in alveolar persistent bronchiolization foci expressed Ki67 until 112 dpi, suggesting ongoing proliferative activity long after the initial infection.

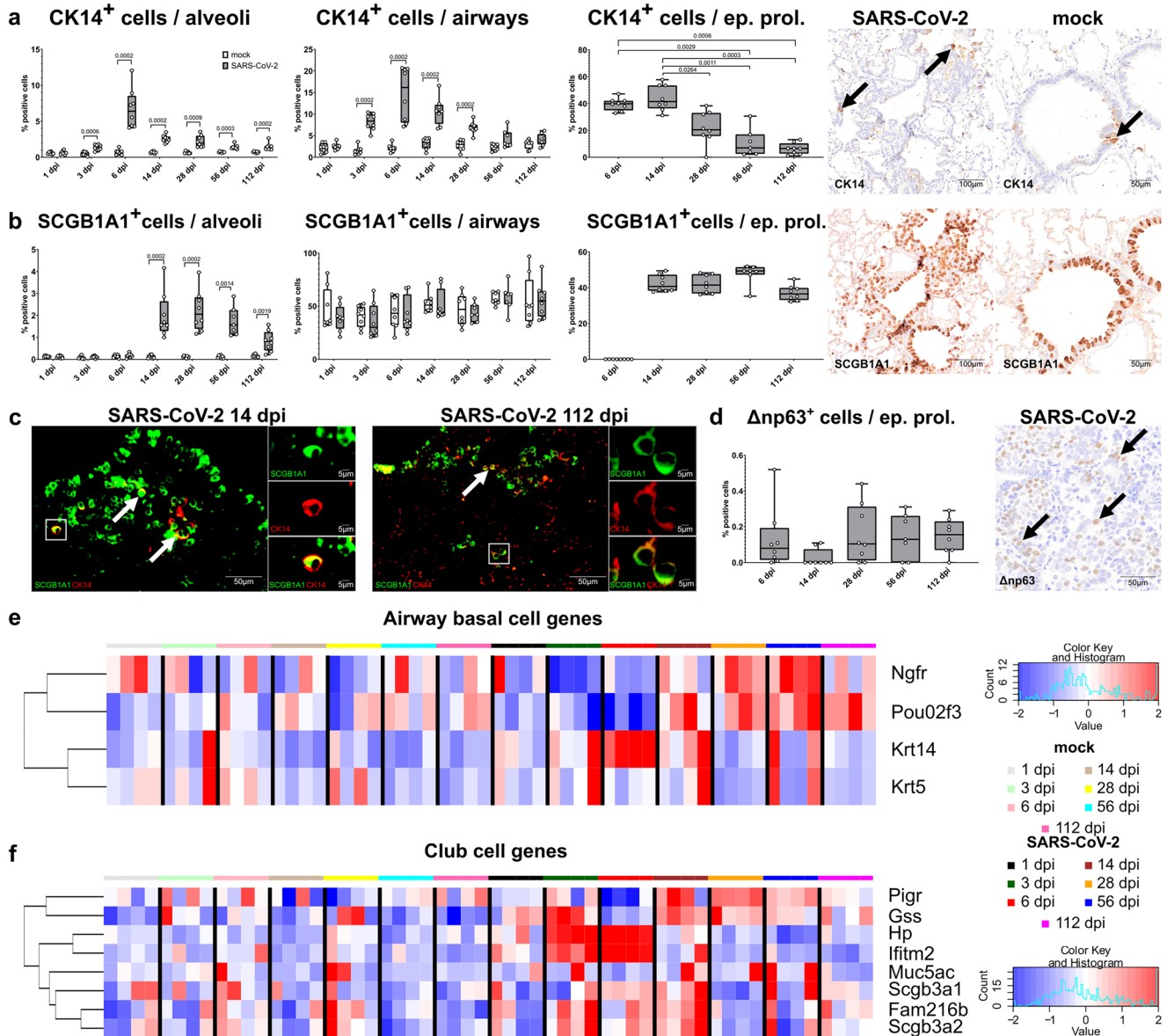

**Fig. 6 | Airway progenitor cells take part in the alveolar regeneration process after SARS-CoV-2 infection in hamsters. a, b** Quantification of cytokeratin 14 (CK14)⁺ airway basal cells (**a**) and secretoglobin 1a1 (SCGB1A1)⁺ club cells (**b**) within the whole alveolar space, airways, and within alveolar epithelial proliferation foci (ep. prol.). On the right, representative pictures of immunolabeling (brown signal) of CK14 (arrows) and SCGB1A1 in the alveoli and airways of SARS-CoV-2 and mock-infected hamsters at 112 days post infection (dpi). **c** Representative images of double immunofluorescence in an alveolar proliferation focus at 14 and 112 dpi. Cells are labeled with SCGB1A1 (green) and CK14 (red). In both pictures there are low to moderate numbers of CK14⁺SCGB1A1⁺ cells (arrows and inlets). The staining was performed in four animals/time-point with same results. **d** Quantification of ΔNP63⁺ airway basal cells within alveolar proliferation foci and representative picture of immunolabeling (brown signal, arrows) in an alveolar proliferation focus

of a SARS-CoV-2-infected hamster at 6 dpi. **e, f** Heatmap of normalized expression values for genes associated with airway basal cells and club cells (Supplementary Table 1) at each dpi in mock- and SARS-CoV-2-infected hamsters. Expression values are scaled by row. Red indicates higher and blue lower relative expression levels. **a, b, d** quantification data are shown as box and whisker plots. The bounds of the box plot indicate the 25th and 75th percentiles, the bar indicates medians, and the whiskers indicate minima and maxima. Dots indicate individual values. Statistical analysis was performed by two-tailed Mann−Whitney *U* test. For multiple comparisons between time-points (quantification within alveolar proliferation foci), a Benjamini−Hochberg correction was applied. *P*- and *q* values ≤ 0.05 were considered significant. *N* = 8 animals/group for mock and SARS-CoV-2 respectively. Source data are provided as a Source Data file.

## Transcriptome analysis of SARS-CoV-2 Delta-infected aged hamsters does not provide evidence of mitochondrial dysfunction but show activation of pathways involved in pulmonary vascular remodeling

Analysis of metabolic parameters by respiratory gas analysis revealed decreased vO₂, vCO₂, and MR, which could be suggestive of a mitochondrial dysfunction in SARS-CoV-2-infected animals. Mitochondrial dysfunction and metabolic reprogramming have been demonstrated during SARS-CoV-2 infection in humans and mouse models[76–80].

For instance, it is reported that SARS-CoV-2 inhibits mitochondrial oxidative phosphorylation (OXPHOS) to increase reactive oxygen species (ROS) production in mice[79]. To investigate metabolic reprogramming and mitochondrial dysfunction on a transcriptome level, we used a published list[78] of hallmark genes involved in metabolic pathways like β-oxidation of fatty acids, ketone metabolism, pyruvate metabolism, and tricarboxylic acid (TCA) cycle, ROS scavenging system, and all the complexes involved (complex I to V) in OXPHOS.

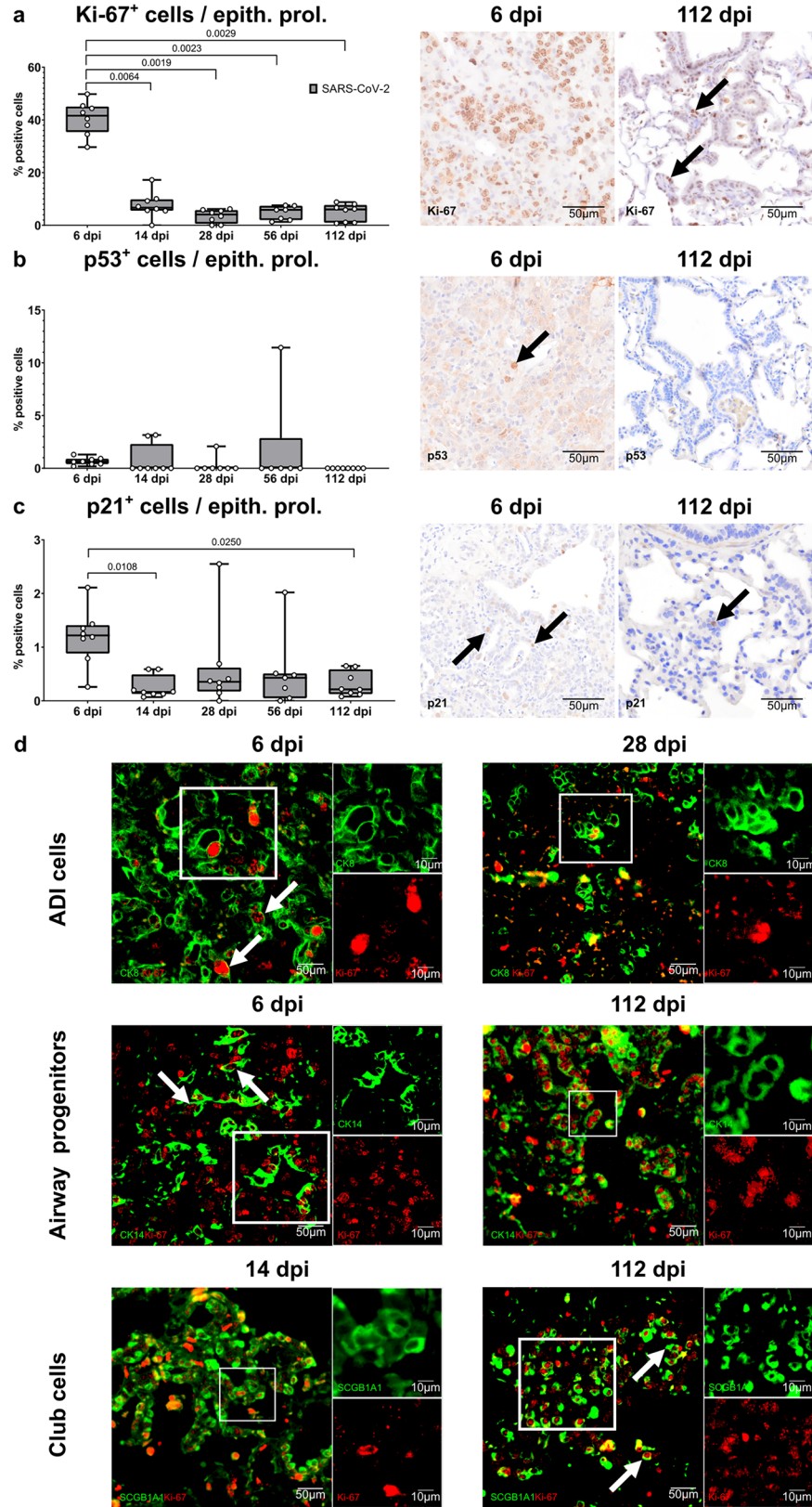

During the acute phase, SARS-CoV-2-infected hamsters showed downregulation of some genes involved in the β-oxidation of fatty acids (*Acad11, Acss1, Acsl1, Acam3,* and *Decr1),* whereas some genes involved in ketone metabolism were upregulated (*Gpd2, Bdh1, Rpia).* This metabolic switch has been reported in humans and in a murine model for COVID-19[76]. Expression of genes involved in the TCA cycle (*Fh1, Idh2,*

*Idh3g*) was upregulated in the acute disease phase. Some genes involved in pyruvate metabolism also showed upregulation (*Rpla, Pdhb, Bdh1*) at 3 and 6 dpi. However, none of these changes were present in the chronic phase of the disease (Supplementary Fig. 7; Supplementary Data 7).

SARS-CoV-2-infected hamsters showed a marked upregulation of genes involved in the ROS scavenging systems, especially at 3 and

**Fig. 7 | Immunolabeling of proliferation and senescence markers in SARS-CoV-2-infected hamsters. a–c** Quantification of Ki67 (**a**), p53 (**b**), and p21 (**c**) immunolabelled cells within alveolar epithelial proliferation foci (ep. prol.). On the right, representative pictures of immunolabeling (brown signal, arrows) in alveolar proliferation foci of SARS-CoV-2-infected hamsters at 6 and 112 days post infection (dpi). **d** Representative images of double immunofluorescence of Ki67 in combination with CK8⁺ ADI cells, CK14⁺ airway progenitor cells, or SCGB1A1⁺ club cells in alveolar proliferation foci at different time-points. In the top panel, cells are labeled with CK8 (green) and Ki67 (red). There are moderate numbers of double-labeled cells (arrows) at 6 dpi, whereas they are rare at 28 dpi. In the central panel, airway progenitor cells are labeled with CK14 (green) and Ki67 (red). There are moderate numbers of double-labeled cells (arrows) at both 6 and 112 dpi. In the bottom panel, cells are labeled with SCGB1A1 (green) and Ki67 (red). All stains were performed in four animals/time-point with same results. There are moderate numbers of double-labeled cells (arrows) at both 14 and 112 dpi. **a–c** quantification data is shown as box and whisker plots. The bounds of the box plot indicate the 25th and 75th percentiles, the bar indicates medians, and the whiskers indicate minima and maxima. Dots indicate individual values. Statistical analysis was performed by a two-tailed Mann–Whitney $U$ test. $P$- and $q$ values ≤ 0.05 were considered significant. $N = 8$ animals/group for mock and SARS-CoV-2 respectively. Source data is provided as a Source Data file.

6 dpi. Of note, genes like *Sod2*, *Gpx1*, *Gsr*, *Prdx1*, *Prdx3* were markedly upregulated. Importantly, some of these genes play a crucial role in mitochondrial ROS scavenging. However, none of these genes were differentially upregulated at the later time-points.

Similarly, SARS-CoV-2-infected hamsters showed an upregulation of most genes belonging to Complex I to Complex V of the mitochondrial OXPHOS only during the acute phase of disease (Supplementary Fig. 8; Supplementary Data 7). These results of upregulated ROS scavenging system and upregulated OXPHOS complexes-related genes are similar to what has been reported in a recent publication that investigated this same gene set in SARS-CoV-2-infected hamsters during the acute phase of the disease[78].

To investigate whether the transcriptomic changes would have a functional correlate on the systemic level, we also quantified levels of Glutathione peroxidase (GSH/GSSG) and peroxiredoxin-3 (PRDX3) in serum samples of hamsters. GSH/GSSG was performed to verify the systemic oxidative stress level, while increased PRDX3 in the serum has been recently reported in long-COVID patients and is considered a potential biomarker of mitochondrial dysfunction[80]. No differences were found between Mock- and SARS-CoV-2-infected hamsters (Supplementary Fig. 8).

In summary, transcriptomic data obtained from lung tissues indicate that mitochondrial dysfunction and metabolic alterations were only present briefly during the acute disease in hamsters, correlating with the peak of inflammation and viral replication. In contrast, we did not find evidence of mitochondrial dysfunction or metabolic derangement in the chronic disease.

The decreased vO₂, vCO₂, and MR in chronic disease could also be linked to increased dead space ventilation due to pulmonary vascular abnormalities, resulting in decreased oxygen uptake in the lung[81–83]. Pulmonary vascular remodeling in COVID-19 is characterized by an angiogenesis of the intussusceptive type, which is induced by increased expression of *Cxcl12* (stromal-derived factor-1) and *Cxcr4*. Another hallmark of this peculiar SARS-CoV-2-induced vascular remodeling is the expression of *Ccl12*, *Gdf15*, *Cd163*, *Col3a1*, which underline the unique vascular etiology of COVID-19 distinguishing it from other forms of interstitial lung disease[84–86]. For these reasons, we sought to investigate the hallmark genes of COVID-19 vascular remodeling in the hamster model. Transcriptome data analysis revealed that *Ccl12*, *Gdf15*, *Cd163* were markedly upregulated during the acute phase at 3 and 6 dpi. *Col3a1* was markedly upregulated from 14 dpi to 56 dpi. Despite a marked downregulation during the acute phase at 3 and 6 dpi, *Cxcl12* was upregulated at 28 and 56 dpi (Supplementary Fig. 9; Supplementary Data 7).

Increased neural precursor cell expressed developmentally downregulated protein 9 (Nedd9), also known as enhancer of filamentation 1 (EF1), serum levels has been recently reported in long-COVID patients and is considered a potential biomarker of vascular remodeling[83,84]. However, we found no differences in the serum levels of EF between Mock- and SARS-CoV-2-infected hamsters at any time-point (Supplementary Fig. 9). Altogether, these results are suggestive of activation of pathways driving pulmonary vascular remodeling typical of COVID-19 during the acute phase.

## Transcriptome dynamics of ADI and airway basal cell signature as well as pro-fibrotic genes are comparable between SARS-CoV-2 and IAV infection

One of the main questions when analyzing data from COVID-19 patients and animal models is whether an observed morphologic, functional, or transcriptomic change is specific to the disease or a common feature of respiratory infections. The most comparable infectious disease is IAV infection. Hamsters are naturally susceptible to IAV and are considered a good model for respiratory as well as systemic disease[87]. In one published study, with infection of young hamsters with SARS-CoV-2 and IAV with inoculation doses adapted to reach equivalent viral loads, the disease severity, histopathological lesions, inflammatory infiltrates, and transcriptomic changes in the lung were qualitatively and quantitatively comparable during acute (3, 7 dpi) and chronic disease (31 dpi). In particular, both viruses induced chronic bronchiolization lesions as observed in our study[88]. Since this study did not look into epithelial cells participating in alveolar regeneration in detail, we decided to analyze selected gene signatures as determined in our own analysis in this published RNAseq dataset.

We selected the datasets obtained at 3 and 31 dpi as representative for acute and chronic infection, respectively, and compared the relative expression of gene sets in SARS-CoV-2-infected and IAV-infected lungs. Subsequently, we compared them to the relative expression in our hamsters at 3, 6, and 56 dpi, representing the time points with the earliest, most prominent, and latest significant changes, respectively. We chose to focus on genes expressed by early-stage ADI cells (*S100a6*, *Krt8*, *Anxa1*, *Tp53*, *Hbgef*), late-stage ADI cells (*Sparc*, *Sox4*, and *Wwtr1*), club cells (*Gss* and *Pigr*), airway basal cells (*Ngfr*, *Pou2f3*, *Krt14*, and *Krt5*) and genes indicating a pro-fibrotic signature (*Mmp12*, *Mmp14*, *Tgfb1*, *Col5a1*, *Col1a1*, and *Col3a1*), which were differentially expressed in our own experiment.

In our dataset, early ADI cell genes were upregulated in the acute phase, whereas late ADI genes are variably upregulated in the chronic phase from 14 dpi onwards, as described above. In the study by Frere et al., SARS-CoV-2-infected animals showed a similar pattern of expression for ADI cell genes, whereas IAV-infected hamsters displayed a mostly homogeneous ADI cell gene expression already in the acute phase of the disease (Supplementary Fig. 10). This could indicate differences in the dynamics of ADI cell trajectories, suggesting a slight delay of ADI cell maturation in SARS-CoV-2 compared to IAV infection.

Club and airway basal cell progenitor genes were upregulated mainly in the chronic phase in our hamsters. In the study by Frere et al., SARS-CoV-2-infected animals showed only a slight upregulation of typical club and airway basal cell progenitor genes in the chronic phase of the disease, whereas IAV-infected hamsters displayed a more prominent upregulation of these genes at this phase (Supplementary Fig. 10). Of note, the authors reported that bronchiolization areas appeared more prominent in SARS-CoV-2-infected animals compared to IAV-infected ones in histology, albeit no significant difference was observed upon morphometry.

In our dataset, early pro-fibrotic environment genes involved in ECM remodeling were upregulated in the acute phase, while collagen-encoding genes were upregulated in the chronic phase of the disease,

as stated above. The study by Frere et al. showed a comparable pattern of expression in both SARS-CoV-2- and IAV-infected hamsters, but upregulation of both ECM-remodeling and collagen-encoding genes was only noted in the acute phase of the disease (Supplementary Fig. 10). This discrepancy is in line with the lack of obvious collagen deposition reported by Frere et al., as opposed to the significant interstitial collagen deposition reported in this study.

In summary, a comparison of our data with a published dataset from SARS-CoV-2 and IAV-infected hamsters showed mostly over-lapping patterns in the expression of epithelial cell genes, with comparable changes in both viral diseases and both experiments. Differences between our data and the data from Frere et al. were observed in the pro-fibrotic gene expression in the chronic phase, with lasting changes in our hamster model as opposed to the transient upregulation in the SARS-CoV-2 and IAV-infected hamsters in Frere et al.

### SARS-CoV-2 Delta infection does not cause long-term effects in the heart, kidney, liver, or spleen of aged hamsters

Although respiratory deficiencies are among the most common symptoms of long-COVID, the respiratory tract is not the only organ system that can be affected by long-lasting disturbances[89]. Especially cardiovascular impairment is considered one of the main contributors to long-COVID. Since the heart and the respiratory tract are intricately connected, we aimed to investigate the heart in greater detail too. Although it has been thoroughly studied in acute hamster models for COVID-19 already[88,90], data on the sub-acute and chronic phase is still sparse[90,91].

No SARS-CoV-2 NP immunolabeled cells were detected in the heart of any animal at any time point. Histopathological examination revealed mild to moderate, mostly sub-epicardial inflammation (11/55 SARS-CoV-2-infected animals, 5/56 mock-infected animals), mild fibrosis (3/55 SARS-CoV-2-infected animals, 1/55 mock-infected animal), focal thrombi in small blood vessels (3/55 SARS-CoV-2-infected animals, 1/56 mock-infected animals) and mild to moderate focal mineralization (2/55 SARS-CoV-2-infected animals, 2/55 mock-infected animals) (Supplementary Fig. 11; Supplementary Data 5). Since mild inflammatory infiltration, thrombosis, and calcification are known to occur as age-related, non-neoplastic lesions in hamsters[39,92], we concluded that our observations were most likely background lesions due to the advanced age of the animals. Since myocardial fibrosis has been reported in SARS-CoV-2-infected hamsters during the acute and sub-acute phase of the disease[90], we analyzed the collagen content in heart sections from 1 to 112 dpi. However, quantification of the Azan-positive areas in the myocardial interstitial spaces revealed no significant differences between SARS-CoV-2-infected hamsters and mock-infected controls (Supplementary Fig. 11). Since the observed sub-epicardial inflammatory infiltrate appeared predominantly histiocytic on histology, immunolabeling for Iba-1 was performed. Quantification of Iba-1+ cells did not display any significant difference (Supplementary Fig. 11).

In the next step, we investigated histological lesions and performed SARS-CoV-2 NP immunolabeling in the kidneys, liver, and spleen. None of the organs displayed SARS-CoV-2 NP immunolabeled cells (Supplementary Data 6). Renal lesions were overall mild and detected in both SARS-CoV-2 and mock-infected animals. They encompassed mild, multifocal tubular dilation or degeneration, mild glomerular vacuolization, and focal or multifocal chronic infarction. All of the aforementioned lesions affected equally Mock- and SARS-CoV-2-infected hamsters (Supplementary Data 6). Azan, Iba-1, MPO, and CD3 semi-quantitative scoring of positive cells did not yield any difference (Supplementary Data 6). In the liver, we mainly observed mild periportal lympho-histiocytic inflammation in both groups. No lesions were observed in the spleen at any time point (Supplementary Data 6).

In conclusion, intranasal infection of aged, male hamsters with SARS-CoV-2 Delta variant does not cause pathological changes in heart, kidneys, liver or spleen.

## Discussion

Although an end to the global health emergency due to the SARS-CoV-2 pandemic has been declared, a large number of patients still struggle with lasting consequences long after they have recovered from the acute disease[2–5]. Even if tissue samples from survivors could be acquired, establishing a clear causal connection between the initial SARS-CoV-2 infection and lesions observed at later time points can be challenging. For this reason, the use of an appropriate animal model in research is of paramount importance. In our previous work, we showed that hamsters are suitable for modeling alveolar regeneration mechanisms in the acute and sub-acute phases of SARS-CoV-2 infection[18]. Based on the presence of alveolar proliferation foci, ADI cells displaying cell cycle arrest, alveolar airway progenitors, and sub-pleural fibrosis at 2 weeks after infection, we postulated that an impaired alveolar regeneration and persistent tissue remodeling could account for prolonged respiratory symptoms and impaired lung function reported in patients with respiratory long-COVID. As a consequence, this work aimed to characterize the trajectory of lung regeneration over the period of 112 days in the hamster model by combining morphological, transcriptome analysis, and functional readouts.

The early phase of the disease was characterized morphologically by pneumonia with high numbers of inflammatory cells, and subsequent features of alveolar regeneration with high numbers of ADI cells and airway cell proliferation. Hamsters had very mild to moderate respiratory signs detectable by clinical scoring but showed a marked alteration of lung function in WBP. This shows that clinical scoring based on periodic visual inspection alone might be a sub-optimal method to assess respiratory impairment in hamsters and that more sensitive methods like WBP have to be applied to accurately assess the impact on the lower respiratory tract. The WBP alterations are in agreement with previous studies using the method to measure lung function in acute SARS-CoV-2 infection in hamsters[32,69,93]. The changes were characterized by a lower frequency and partly increased TV, with a prolonged, slower expiration phase characterized by increased Te and reduced EF50, but unchanged inspiration phase. This breathing pattern is typically observed in conditions with airflow limitation due to obstruction of airways, e.g. rodent airway hyperresponsiveness models[35,36]. During acute SARS-CoV-2 infection, this most likely corresponds to the damage to airway epithelium and obstruction of the airways by exudate, debris, and inflammatory cells. Additionally, from 6 dpi onwards, prominent hyperplasia and migration of airway progenitor cells in the terminal bronchioles is observed, as we reported in detail previously[18]. This process putatively contributes to airflow restriction due to luminal narrowing. Interestingly, there are some differences between our data and the changes reported in SARS-CoV-1 and SARS-CoV-2-infected mice[32,93,94], which show an increase of EF50 after infection. The SARS-CoV-1 study reported that this goes along with a prolonged Te and lower Rpef, indicating a rapid exhalation of the majority of the volume, but a prolonged time needed to exhale the remainder volume. This breathing pattern combines elements of both restrictive (rapid exhalation due to potentially reduced compliance) and obstructive (slow expiration with reduced flow rate in late expiration) lung disease patterns[94]. Our hamster model does not recapitulate the element indicative of restrictive disease and is rather consistent with an obstructive phenotype. Of note, the prominent epithelial proliferation in terminal bronchioles and alveolar bronchiolization that occurs in SARS-CoV-2-infected humans and hamsters[18,95] are not observed in mice[69]. This could potentially explain differences in plethysmography changes between the two species.

Another possible contributor to the reduced respiratory rate observed in the acute phase is a behavioral difference with the reduced activity of the hamsters in the plethysmograph. Recent publications evaluating the hamster model argued that the reduced respiratory rate observed after SARS-CoV-2 infection is driven by behavioral changes such as reduced exploratory behavior, grooming, or chewing, related to general malaise[96,97]. As a matter of fact, during acute disease, the SARS-CoV-2-infected hamsters in our experiment tended to sit quietly in the plethysmograph, while some mock-infected animals showed intermittent phases of activity. However, in order to reduce this bias, we used long acclimatization times, and abnormal breathing patterns indicative of sniffing or grooming were removed from the measurements.

In the chronic phase, no differences were observed in respiratory rate, TV, Te, Ti, or EF50 before exercise. This suggests that the observed changes in the acute phase are directly related to the acute damage and inflammation, which are mostly resolved within 14 days. Interestingly, after exercise, we again observed marked lung function changes. The use of a rodent treadmill was inspired by the use of similar tests in human medicine[98] and we think that its use should be taken into consideration in future animal studies modeling long-term respiratory signs induced by SARS-CoV-2. The majority of studies in PASC patients show a gradual improvement of lung function over time, although this is not always accompanied by a regression of symptoms[8,99–101]. In contrast, a recent study comprised exclusively of patients with persistent pulmonary PASC shows no improvement of clinical symptoms, diffusion impairment, and pulmonary restriction, which is suggestive of lung fibrosis[21]. However, studies utilizing lung tissue from PASC patients are rare, and therefore the association of lung function with a specific pathotype remains speculative. Therefore, one of the goals of this study was to determine whether there is an association between pathomorphological changes (bronchiolization and interstitial fibrosis) and lung function alterations. The observed changes in lung function after exercise were present until 7 weeks post infection and showed the same pattern as observed in the acute disease, indicative of an obstructive phenotype. We assume that this obstruction is caused by the space-occupying effect of persistent proliferation foci at the bronchioloalveolar junction areas, which were present in almost all animals until 56 dpi, but declined thereafter. In contrast, pulmonary interstitial fibrosis does not appear to alter lung function since changes in lung function were not indicative of a restrictive condition and since the fibrosis was stable until 112 dpi, but lung function recovered. Apparently, the relatively low degree of fibrosis does not impact breathing or can be compensated by remaining unaltered lung tissue. As for the acute phase, we assume that behavioral differences also contributed to the changes in parameters measured by plethysmography after exercise. Infected hamsters showed a slight decrease of the running score on the treadmill from 21 to 56 dpi compared to the mock-infected group, which could point towards a higher reluctance to run and a higher level of exhaustion, reminiscent of exercise intolerance reported in humans with respiratory long-COVID. The observed reduced respiratory rate in SARS-CoV-2-infected hamsters could indicate that the animals were spending more time resting.

Besides the alterations of breathing parameters, SARS-CoV-2-infected hamsters also showed decreased vO2, vCO2, and MR in the acute and chronic phase. We hypothesize that these alterations could have been caused by one or more of the following: i) reduced metabolic rate due to reduced physical activity, ii) reduced muscle mass, iii) impaired mitochondrial function, and iv) increased dead space ventilation, e.g., due to vascular remodeling. As discussed above, SARS-CoV-2-infected animals were calmer in the plethysmograph in the acute disease phase and were performing slightly worse on the treadmill and this may have contributed to the differences in metabolism between the groups. However, vO2 and MR were already lower in

SARS-CoV-2-infected hamsters before exercise, pointing towards a lower resting metabolic rate. This could be explained by reduced skeletal muscle mass resulting in lower oxygen consumption. Infected animals lost weight after infection and did not recover their initial weight until the end of the study. We assume that this is at least partly due to the lasting loss of skeletal muscle mass. Loss of fat-free mass, skeletal muscle mass, and reduced resting metabolic rate is also observed in COVID-19 patients, even those with mild disease not requiring hospitalization[42]. Furthermore, there is evidence of mitochondrial dysfunction and metabolic reprogramming during SARS-CoV-2 infection in humans and mouse models[76–80]. Our transcriptomic data indicate that mitochondrial dysfunction and metabolic alterations were only present briefly during the acute disease in hamsters, correlating with the peak of inflammation and viral replication, which is in line with the published evidence in COVID-19[78]. In contrast, we did not find evidence of mitochondrial dysfunction or metabolic derangement in the chronic disease. However, it should be mentioned that for a complete evaluation of mitochondrial dysfunction causing a metabolic change observed in this study, organs like skeletal muscles, heart, and liver should be investigated in depth. Since this metabolic alteration was not expected, these organs were not sampled to perform transcriptome analysis for the current study. Lastly, increased dead space ventilation due to pulmonary vascular abnormalities, resulting in decreased oxygen uptake in the lung, could have contributed to decreased vO2, vCO2, and MR in this experiment[81–83]. In our study, some markers of the intussusceptive vascular remodeling typical for COVID-19 were upregulated during the acute (Ccl12, Gdf15, and Cd163) or chronic phase (Col3a1, Cxcl12). These results could be suggestive of the activation of pathways driving pulmonary vascular remodeling. However, further studies, including micro-CT analysis, are needed to assess for micro- and macro-vascular changes in the lung of SARS-CoV-2-infected hamsters and to substantiate the molecular findings and interpretations.

It has been postulated that persistence of SARS-CoV-2 in the lung could be the key to explain lasting symptoms of respiratory long-COVID[102,103]. Sporadic reports of viral RNA shedding for up to 83 days in the upper respiratory tract, 59 days in the lower respiratory tract, 126 days in stools, and 60 days in serum have been identified by a meta-analysis[104]. In one study conducted in deceased patients SARS-CoV-2 Spike and NP immunohistochemistry revealed signal in bronchial cartilage chondrocytes and parabronchial glands of most patients, despite repeated negative results in nasopharyngeal swabs or bronchioalveolar lavage (BAL) for an average of 105 days[65]. In addition, at least 60 cases with conclusive evidence for persistent SARS-CoV-2 infection with ongoing virus replication have been described[102]. We detected viral Esub RNA until 28 dpi and RdRp RNA until 112 dpi, which could point to a prolonged presence of replicating virus[105]. However, the correlation between the presence of Esub RNA and active virus replication/transcription has been contested[106]. In animal models, only a few long-term studies have been performed so far, and there is limited and partly inconsistent data on virus persistence. While most studies fail to demonstrate infectious virus, viral RNA, and/or viral protein in the lung beyond acute disease[26,88], others have reported minimal amounts of infectious virus particles at 42 dpi in the hamster model following a 7-day culture of homogenized lung tissue[67]. Another study in macaques showed that replication-competent SARS-CoV-2 was detectable in cultured BAL macrophages isolated 6 months after infection, inducing IFNγ and NK cell dysregulation[66]. The discrepancies could be related to differences in study design, choice of animals, virus strain and dose as well as to the very low amount of persistent virus and the choice of detection methods. Thus, it seems that an initial propagation step and/or a targeted approach need to be applied to demonstrate the persistence of infectious virus, and that the negative results from studies using standard methods utilizing whole lung tissue have to be interpreted with caution. Therefore, we cannot

completely exclude the presence of minimal amounts of replicating virus below the limits of detection of immunohistochemistry. Interestingly, pulmonary viral persistence has been associated with chronic weight loss in the hamster model by others[67], and we also found that the animals with detectable viral RNA in the lung were the ones with the most severe weight loss and/or the ones that showed the least recovery of the initial body weight. In line with this, a recent study conducted as part of a national infection survey reported that patients with detection of SARS-CoV-2 RNA at high titer persisting for at least 30 days had more than 50% higher odds of self-reporting long COVID[102]. Therefore, even in the apparent absence of replicating viruses, a link between viral RNA persistence and clinical signs should be considered.

During the chronic phase of the disease, we observed mild to moderate but persistent pathomorphological changes like sub-pleural and interstitial pulmonary fibrosis as well as alveolar bronchiolization areas dominated by SCGB1A1+ club cells and fewer CK14+ basal cells. Club cells and bronchiolization have been associated with the maintenance of a pro-fibrotic milieu[107] and with idiopathic pulmonary fibrosis[108], respectively. Our transcriptome data analysis supports the presence of a pro-fibrotic environment with the prolonged presence of CD204+ M2-like macrophages at the later time points. In addition, the gene *Cyp2e1* was an upregulated DEG until 56 dpi in SARS-CoV-2-infected animals. *Cyp2e1* belongs to the P450 superfamily and in the lung it is mainly expressed by club cells[109]. It has been demonstrated that the upregulation of *Cyp2e1* is involved in pulmonary fibrosis via ER stress- and ROS-dependent mechanisms[110]. In our previous study[18], we postulated that the cause of the fibrosis was mainly attributed to the presence of ADI cells. However, it is clear now that the pathogenesis of SARS-CoV-2-induced pulmonary fibrosis appears to be more complex and involves multiple cell types. Taking into consideration that ADI cells are not the prominent cell type in the chronic phase of the disease, more attention should be focused on the club cells within the bronchiolization areas. It is worth noticing that our transcriptome data analysis displayed the presence of a number of DEGs that were either shared between or unique to the sub-acute and the chronic phases of the disease. As most of these DEGs are involved in cell migration, GABA-A, ephrin, or serotonin receptors, neuronal growth, plasticity, and signaling, as well as trans-synaptic signaling, this could hint towards the presence of a recently described niche of club cells residing in the so-called neuroepithelial bodies[111]. However, future immunohistochemical and in situ hybridization studies are warranted to confirm such postulation.

The prolonged presence of bronchiolization areas could increase the risk of developing other conditions. For instance, it is known that cigarette smokers are prone to develop peribronchiolar alveolar bronchiolization[112]. Smokers affected by long-COVID could potentially be at higher risk for more extensive alveolar bronchiolization due to the combined action of cigarette smoking and SARS-CoV-2 infection. As a consequence, this could increase the risk of pulmonary fibrosis due to the relationship between bronchiolization and fibrosis[108]. In addition, pulmonary fibrosis and alveolar bronchiolization are considered risk factors for lung cancer and premalignant change, respectively[113,114]. In light of the above, these patients should be particularly monitored for a possible increased risk for lung cancer. Interestingly, high-mobility group box-1 (HMGB1) protein, which is expressed by club cells in the lung, promotes airway hyperresponsiveness and an asthma-like type 2 response. For this reason, airway hypersensitivity could represent another condition potentially promoted by the persistence of alveolar bronchiolization rich in club cells in long-COVID patients[115]. This has already been demonstrated for other viruses. For instance, respiratory syncytial virus infection in early life is considered an important risk factor for the development of asthma[116].

A contribution of airway progenitors to alveolar regeneration has been reported in COVID-19 patients, and their presence has been associated with a pro-fibrotic gene signature and the presence of fibroblastic areas and bronchiolization. We have shown here and in our previous publication[18], that the hamster recapitulates this pattern. The question remains, whether any of this response is SARS-CoV-2 specific or whether it could be observed in any other viral pneumonia. The current understanding of alveolar regeneration is that AT2 cells are mainly responsible for AT1 cell regeneration in homeostatic turnover and following mild injury, while airway progenitors are recruited after severe injury with marked AT1 cell loss[73,117,118]. The participation of airway progenitors and ADI cells in alveolar regeneration has been demonstrated in different viral respiratory diseases (SARS-CoV-2, IAV, Sendai virus infection)[18,71,72,88,119], and non-infectious mouse models of lung injury (e.g., bleomycin, neonatal hypoxia and hyperoxia, LPS)[72,73,120–124]. The most comparable infectious disease is IAV infection. Hamsters are naturally susceptible to IAV and are considered a good model for respiratory as well as systemic disease[87]. A number of studies have been published on the comparisons of SARS-CoV-2 and IAV infection in hamsters[88,125–127]. Many of these demonstrate a higher disease severity in SARS-CoV-2 infection compared to IAV infection when using similar infection doses[125,127,128]. However, in one published study with infection doses adapted to reach equivalent viral loads, disease severity, histopathological lesions, inflammation and transcriptomic changes in the lung were comparable between the models during acute and chronic disease[88]. Our analysis of the published transcriptome dataset obtained from SARS-CoV-2 and IAV-infected young hamsters and comparison with our transcriptome data in old SARS-CoV-2-infected hamsters confirmed similarities of the two diseases regarding cell types participating in alveolar regeneration. In addition, we demonstrated here that Plet-1, which has been recently identified as an important mediator of macrophage-epithelial cross-talk in alveolar regeneration in the IAV model[68], is also upregulated during the regeneration phase of SARS-CoV-2 infection. One discrepancy between the dataset from young and old hamsters was found regarding pro-fibrotic gene expression. In young SARS-CoV-2- and IAV-infected hamsters, pro-fibrotic genes were only upregulated in the acute disease, while we additionally noted upregulation of collagen-encoding genes in the chronic phase. This discrepancy is in line with the lack of obvious collagen deposition reported by Frere et al., as opposed to the significant interstitial collagen deposition reported in this study. This discrepancy is most likely related to the age difference, since the aged lung shows an impaired regeneration capacity and increased pro-fibrotic changes following alveolar damage[129]. Combining all the evidence, we conclude that the morphologic and transcriptomic features of lung regeneration we observe in hamsters reflect a stereotypical response to severe alveolar damage and not a distinct SARS-CoV-2-specific phenomenon. The fact that the response overlaps with that of other respiratory diseases highlights the usefulness of the model also beyond COVID-19 and expands the relevance of the findings beyond the COVID-19 research field.

Interestingly, mobilization of airway progenitors appears not to be a feature of SARS-CoV-2 infection in the mouse models. For instance, sublethal SARS-CoV-2 infection of K18-hACE2 mice failed to induce airway progenitor cell proliferation in contrast to IAV infection of B6 mice[71]. The lack of airway progenitor cell proliferation has also been highlighted in BALB/c mice infected with the mouse-adapted SARS-CoV-2 strain MA10[71]. We believe that the apparent qualitative differences between viral models observed by some are not indicative of a true virus-specific response pattern, but most likely related to the severity of alveolar damage, which in turn is dependent on the virus dose and strain as well as the choice of animal species and age group. It appears that SARS-CoV-2-induced alveolar damage is more severe in hamsters, whereas IAV is provoking more severe damage in mice.

The authors recognize that the study has some limitations. First, we used only a single SARS-CoV-2 variant. We recognize that in the planning and execution of this study, several other variants occurred globally. However, our choice on SARS-CoV-2 Delta variant was based on the fact that the risk of developing long-COVID is most often associated with Delta variant infection[130,131]. By the middle of 2021, Delta variant was the most prevalent variant of concern[132]. Compared to SARS-CoV-2 Omicron BA1, the Delta variant is also known to cause more severe pulmonary lesions in hamsters[31]. Second, we used only old male animals. We are aware that in an optimal condition the use of male and female as well as young and aged hamsters would have been preferable and that the results of this study cannot be extrapolated to females and young hamsters. However, our aim was to offer a model for a population at high risk of respiratory long-COVID. Risk factors for the development of respiratory long-term sequelae include severe acute disease[8,21,23,133,134], advanced age[3,135] and sex, with males being more likely to develop severe acute disease[8,20–23,26]. Moreover, male sex is associated with higher risk for respiratory long-COVID with diffusion impairment and restriction in humans[21]. Similarly, male hamsters show more severe disease, lung function impairment, and histological lesions in the acute phase[26,28–30,67]. The few long-term studies extending beyond the first 2 weeks also report males as recovering more slowly than females[26,29,30]. Thus, to increase the likelihood of observing long-lasting pulmonary long-COVID, we used one-year-old male hamsters. Future studies should explore whether there are sex-specific differences in the long-term consequences of infection in the hamster model. Third, the conclusions regarding cell fate of CK14+ basal cells in this work are based on double-labeling and expression of genes interpreted in the context of published data. Therefore, they have to be interpreted as preliminary observations. Fourth, we did not sub-classify club cells in all the sub-populations that are reported in the literature. This is mostly due to the fact that several tested antibodies failed to specifically recognize club cell sub-populations in hamsters.

In conclusion, our study shows that functional long-term respiratory consequences of SARS-CoV-2 infection can be modeled in aged male hamsters, using functional tests in combination with exercise. Alveolar bronchiolization and interstitial pulmonary fibrosis persist up to 112 dpi and are likely to represent potential risk factors for other diseases in long-COVID survivors, and this should be addressed in future studies. Altogether, these results represent further steps toward a more complete understanding of long-term pathogenesis.

## Methods

### Ethics statement
The animal experiment was in accordance with the EU directive 2010/63/EU and approved by the state office for consumer protection and food safety of Lower Saxony (Niedersächsisches Landesamt für Verbraucherschutz und Lebensmittelsicherheit, LAVES, protocol code TV22-00088, approval 16.09.2022).

### Animal experiment
During the experiment, the animals were under veterinary observation, and all efforts were made to minimize distress. Eleven to twelve-month-old male Syrian golden hamsters (*Mesocricetus auratus*, RjHan:AURA) purchased from Janvier Labs (strain origin: Zentralinstitut für Versuchstierzucht (Hanover)) were housed under BSL-3 conditions for at least 2 weeks prior to the experiment for acclimatization. A total of 112 hamsters were randomly assigned to groups of 8 animals per time-point and treatment ($n = 16$ per dpi). They were housed in isolated ventilated cages under standardized conditions ($21 \pm 2\,°C$, $40$–$50\%$ humidity, 12:12 inversed light-dark cycle, food and water ad libitum) at the Research Center for Emerging Infections and Zoonoses (RIZ) in Hannover, Germany. The animals were injected intranasally with either a suspension of $10^4$ TCID$_{50}$ of SARS-CoV-2 (SARS-CoV-2 Delta variant) or phosphate-buffered saline (PBS, control) as previously

described[136,137] under general anesthesia. After infection, the animals were monitored with daily scoring and weighing until 21 days post infection (dpi). At chosen time points, the animals underwent lung function measurements as described below. At 1, 3, 6, 14, 28, 56, and 112 dpi, groups of 16 animals were put in deep general anesthesia by inhalation of isoflurane in an induction chamber followed by an intraperitoneal administration of 100 mg/kg ketamine and 0.25 mg/kg medetomidine. Final euthanasia was performed by exsanguination through an abdominal aorta incision. Subsequently, the lung, heart, kidneys, liver, and spleen were collected and fixed in 10% neutral-buffered formalin. In particular, the left lung lobe was pre-fixed by injections of 10% buffered formalin. Furthermore, tissue from the right lung lobes was collected and suspended in 500 µl RNAlater (Invitrogen).

### In vivo lung function measurement, treadmill, and exercise scoring
To measure lung function parameters, a WBP (Buxco small animal WBP, DSI) connected to a gas analyzer (ADInstruments) was used under BSL-3 conditions. From 21 dpi, the animals were additionally placed on a rodent treadmill (Ugo Basile), and lung function was measured before and after exercise. No exercise was performed on animals during the acute and sub-acute phases of the disease due to animal welfare reasons. During the housing phase, the animals were familiarized with the measuring procedure three times. The training sessions consisted of 10 minutes of plethysmography followed by increasing durations of the treadmill. During the first training, the treadmill was switched off, and the animals were free to explore. Treats (sunflower seeds or waffles) were placed inside as a reward. Subsequently, the treadmill was switched on and animals were running for 5 minutes at a slow speed. The speed and duration were increased in subsequent training sessions. The treadmill training was only conducted in animals of the 28, 56, or 112 dpi groups. All groups underwent the training with the plethysmograph. After infection, all animals were measured in the plethysmograph at 0, 3, 6, 10, and 14 dpi, followed by weekly measurements in combination with exercise on the treadmill from 21 dpi until 112 dpi. Due to space and time restraint reasons, 6 out of the 8 animals from each group (SARS-CoV-2 and mock-infected) were chosen to undergo exercise according to the behavior on the treadmill during the training sessions. Since no negative reinforcement was used in this experiment, the hamsters could stop running at any time and rest on the grid positioned at the lower end of the treadmill. The behavior was scored with the following, semi-quantitative score: 1: Animal stays on the treadmill´s grid most of the time, very reluctant to move, 2: animal needs some time to start running, but runs quite constantly towards the end; 3: animal runs constantly throughout the exercise. The animals with the best score during the training sessions were selected for exercise during the chronic infection. A similar scoring system was then used to assess and compare the animals' performance after infection (1: Animal frequently stops (more than 5 times) and it may manifest labored breathing pattern, 2: Animal occasionally makes brief stops (up to 5 times). When it stops it manifests normal breathing pattern, 3: Animal runs throughout the exercise). The measurement in the plethysmograph before exercise lasted at least 25 minutes, which included a ≥10-minute acclimatization period. If exercise was performed, the treadmill was set to an accelerating speed starting at 10 m/min and ending at 15 m/min at a 5° angle for 10 minutes. Immediately afterwards, the animals were transferred back to the plethysmograph to be measured for another 15 minutes without any acclimatization. The animals running on the treadmill were under constant veterinary observation. Hamsters are considered night animals, therefore all the procedures of lung function measurements and exercise were performed during the dark period of the light cycle. Measurements were collected with the software FinePointe Respiratory Software

(Version 3.0.1.13370). After the acquisition, data were checked for breathing artifacts like sniffing patterns or moments when the animals were scratching or moving. These patterns were identified and marked by the software and subsequently reviewed and manually excluded from recordings in order to avoid artifacts. The lung function parameters that were chosen to be analyzed were: frequency, TV, EF50, Ti, and Te. Parameters evaluated by respiratory gas analysis were $vO_2$, $vCO_2$, MR, and RQ. Description and unit of the parameters are listed in Supplementary Table 2. Statistical analysis and graphs design were performed using the statistical software SAS® version 9.4 (SAS Institute Inc., Cary, NC, USA) with the SAS Enterprise Guide 7.1 Client and GraphPad Prism 9.3.1 (GraphPad Software, San Diego, CA, USA) for Windows™. The assumption of normal distribution was tested using the Kolmogorov-Smirnov test. Due to the rejection of the assumption of normal distribution, non-parametric methods were used. Pairwise comparisons between SARS-CoV-2-infected hamsters and the control group at each time point were tested with a Wilcoxon test. Statistical significance was accepted at exact $p$ values of ≤0.05.

### Virus

SARS-CoV-2 (isolate hCoV-19/USA/PHC658/2021, lineage B.1.617.2 Delta variant, NR-55611) was obtained from BEI Resources, NIAID, NIH, and cultured in Vero cells (ATCCR, CRL-1586) using DMEM (Sigma-Aldrich) with added 2% FBS, 1% penicillin-streptomycin, and 1% L-glutamine at 37 °C. All experiments involving SARS-CoV-2 infection were conducted in BSL-3 laboratories and stables at the Research Center for Emerging Infections and Zoonoses (RIZ), University of Veterinary Medicine Hannover, Germany.

### Histological examination

For histological evaluation, formalin fixed-paraffin embedded (FFPE) lung tissue was used. The entire left lung lobe was collected and embedded as a whole, with the bronchus facing down in the mold, resulting in longitudinal sections parallel to the main bronchi. 2 μm thick serial sections were prepared and stained with hematoxylin and eosin (HE) and Azan trichrome. Qualitative evaluations with special emphasis on inflammatory and epithelial regenerative processes (HE) as well as on fibrosis (Azan) were performed in a blinded fashion by veterinary pathologists (FA, LH) and subsequently reviewed by board-certified veterinary pathologists (MCI, WB). Lung samples were scored by a veterinary pathologist (FA) in a blinded manner with a semi-quantitative scoring system with special emphasis on inflammation, degeneration, and regeneration as previously described[31].

FFPE samples of the heart, kidney, liver, and spleen were also cut into 2 μm serial sections and stained with HE and Azan trichrome (heart, kidney). Qualitative evaluations were performed in a blinded fashion by veterinary pathologists (IZ, TS) and subsequently reviewed by board-certified veterinary pathologist (MC, WB). Examined heart section included 2 transversal sections, including both the right and the left chamber, one right above the atrio-ventricular valves, and one in the middle of the ventricles. Kidneys were examined in toto in longitudinal sections while 2–3 cross-sections of liver and spleen were used.

### Azan trichrome quantification

Azan trichrome stained tissue sections were digitized using an Olympus VS200 Digital slide scanner (Olympus Deutschland GmbH). Image analysis software (Visiopharm Software Version 2023.01, Hoersholm Denmark) was used to measure the Azan-positive areas by thresholding (contrast red-blue filter with threshold −50) and tissue area (green filter with threshold 220). Azan-positive area per the whole lung was calculated by division. Statistical analysis and graph design were performed using GraphPad Prism 9.3.1 (GraphPad Software, San Diego, CA, USA) for Windows™. Single comparisons between SARS-CoV-2-infected hamsters and the control group were tested with a two-tailed

Mann–Whitney $U$ test. Statistical significance was accepted at exact $p$ values of ≤0.05.

### Immunohistochemistry

Immunohistochemistry was performed on consecutive lung sections from all animals to detect SARS-CoV-2 antigen (SARS-CoV-2 nucleoprotein and spike protein), macrophages, and dendritic cells (ionized calcium-binding adapter molecule 1, Iba-1), T-cells (CD3), B-cells (pax5), neutrophils (myeloperoxidase, MPO), ADI cells (cytokeratin 8), airway basal cells (cytokeratin 14, cytokeratin 5 and ΔNP63), club cells (secretoglobin 1A1), bronchial cells (MUC5AC, MUC5B), senescent cells (p21, p53), proliferating cells (Ki67), M2-like macrophages (CD204), and Plet1. Furthermore, SARS-CoV-2 NP was stained on heart, kidney, liver, and spleen slides for all animals. Additionally immunohistochemistry of Iba-1, MPO, and CD3 was performed on heart sections from all animals. Immunolabeling was visualized either using the Dako EnVision+ polymer system (Dako Agilent Pathology Solutions) and 3,3´-Diaminobenzidine tetrahydrochloride (DAB, Carl Roth) as previously described[31] or using avidin–biotin complex (ABC) peroxidase kit (Vector Labs) and DAB (Carl Roth) as previously described[138]. Nuclei were counterstained with hematoxylin. Further details about primary and secondary antibodies, visualization methods, and dilutions used can be found in Supplementary Table 3. For negative controls, the primary antibodies were replaced with adjusted protein concentrations of rabbit serum or BALB/cJ mouse ascitic fluid, respectively. Antibodies were tested on murine and hamster lung tissue to confirm specificity for the cells of interest. Subsequently, hamster tissue was used as a positive control.

### Immunofluorescence

Double and triple-labeling immunofluorescence was performed on consecutive lung sections from selected animals to investigate the presence of alveolar pneumocytes type 1 (Podoplanin), ADI cells, and alveolar pneumocytes type 2 (pro-SPC). In addition, we also investigated CK14$^+$ basal cell fate (cytokeratin 14), cell cycle activity (Ki67), and cell senescence (p53). The reaction was carried out as previously described with minor modifications[18,139]. Briefly, after deparaffinization, heat-induced epitope retrieval, and serum blocking, washing with PBS in between each step, a dilution containing two primary antibodies was added and incubated for 120 minutes at room temperature. Afterwards, a dilution containing two secondary antibodies was incubated for 90 minutes at room temperature in the dark. Additionally, on triple-labeled slides, fluorescent conjugated antibodies (Alexa-Fluor®488 CK8, AlexaFluor®488 Scgb1A1) were incubated for 120 minutes. After washing with PBS and distilled water, sections were counterstained and mounted using an anti-fade mounting medium containing DAPI (Vectashield®HardSet™, Biozol). Further details about primary and secondary antibodies, visualization methods and dilutions used can be found in Supplementary Table 3. For negative controls, the primary antibodies were replaced with rabbit serum or BALB/cJ mouse ascitic fluid, respectively, with the dilution chosen according to the protein concentration of the exchanged primary antibody.

### Digital image analysis

To quantify immunolabeled cells in pulmonary tissue slides were digitized using an Olympus VS200 Digital slide scanner (Olympus Deutschland GmbH). Image analysis was performed using QuPath (version 0.5.0), an open-source software package for digital pathology image analysis[140]. For all animals, immunolabellings for cytokeratin 8 (CK8), cytokeratin 14 (CK14), and secretoglobin 1A1 (SCGB1A1) were evaluated on digital whole slide images of the entire left lung as previously described[18]. Briefly, total lung tissue was detected automatically using digital thresholding. Afterwards, regions of interest (ROI) were defined. The ROIs airways (bronchi, bronchioles, terminal bronchioles), blood vessels, epithelial proliferation (alveoli that were

involved in an epithelial regenerative process), and artifacts were manually outlined. The area denoted as total alveoli was defined by subtraction of the blood vessels, airways, and artifacts ROIs from the total lung tissue using an automated script. For p53, p21, ki67, and ΔNP63, only epithelial proliferation was marked manually as a ROI. Using tissue- and marker-specific thresholding parameters, quantification of immunolabeled cells was achieved by automated positive cell detection in all ROI. To analyze SARS-CoV-2 NP, Iba-1, MPO Pax5, CD3, and CD204 immunolabeling, total lung tissue was automatically detected using digital thresholding. Afterwards, only blood vessels and artifacts were indicated as ROIs and subtracted from the total lung tissue. Based on tissue and marker-specific thresholding parameters, quantification of immunolabeled cells was then achieved by automated positive cell detection. All procedures (tissue detection, indication of ROIs, positive cell detection) were performed and subsequently reviewed by at least two veterinary pathologists (FA, GB, LH, MC). Statistical analysis and graph design were performed using GraphPad Prism 9.3.1 (GraphPad Software, San Diego, CA, USA) for Windows™. Single comparisons between SARS-CoV-2-infected hamsters and the control group were tested with a two-tailed Mann–Whitney $U$ test. For multiple comparisons among different time-points data were tested for significant differences using Kruskal–Wallis tests and corrected for multiple group comparisons using the Benjamini–Hochberg correction. Statistical significance was accepted at exact $p$ values of ≤0.05.

## RNA-isolation

Lung tissues from infected hamsters were homogenized in 1 mL of DMEM with penicillin and streptomycin (Gibco) using the TissueLyser-II (Qiagen). RNA was then extracted from 100 μL of the homogenized lung sample using the NucleoMag® RNA kit (MACHERY-NAGEL) following the manufacturer's protocol.

## RNAseq

The quality and integrity of the total RNA were assessed using the Agilent Technologies 2100 Bioanalyzer (Agilent Technologies; Waldbronn, Germany). To prepare the RNA-sequencing library, 500 ng of total RNA was utilized. mRNA purification was carried out using the Dynabeads® mRNA DIRECT™ Micro Purification Kit (Thermo Fisher), followed by library preparation using the NEBNext® Ultra™ II Directional RNA Library Prep Kit (New England BioLabs) in accordance with the manufacturer's protocols. The resulting libraries were sequenced on the Illumina NovaSeq 6000 platform, employing the NovaSeq 6000 S1 Reagent Kit (100 cycles) for paired-end runs. On average, each RNA sample yielded ~3–5 × $10^7$ reads. Subsequently, each FASTQ file underwent quality assessment using the FASTQC tool[141]. Before alignment to the reference genome, raw FASTQ files were processed to trim sequences with low base call quality and remove sequencing adapter contamination (fastq-mcf, http://expressionanalysis.github.io/ea-utils/). Reads shorter than 15 bp were excluded from the FASTQ files. The trimmed reads were then aligned to the reference genome using the open-source short read aligner STAR[142], with default settings. The reference file for Golden Hamster (MesAur1.0) was taken from Ensemble database release 110. The Sars-CoV-2 reference file was obtained from Ensembl COVID-19 database (Genome assembly: ASM985889v3). The following steps of analysis were performed using R software package (version 4.2.1) and RStudio (version 2022.02.0 Build 443)[143]. Feature counts were obtained by Rsubread v2.16.0 followed by feature annotation using biomaRt_2.54.1. In cases where the gene symbol information for hamster genes was missing, the orthologous human gene symbols were used instead. For the transcriptome analysis, only genes with a biotype of 'protein coding' and counts greater than 10 in at least four samples were included. Differential gene expression analysis was performed using DESeq2 (version 1.36.0). Normalized values were generated with the rlog function (regularized log transformation) in

DESeq2 and expressed as $\log_2$ values. Volcano plots were generated with the package EnhancedVolcano (version 1.14.0)[144]. Functional analyses of DEGs were performed using the R software package cluster Profiler (version 4.4.4)[145]. Heatmaps were generated with the function heatmap2 of package gplots (version 3.1.3; https://github.com/talgalili/gplots) and package pheatmap (version 1.0.12). Upset plots were prepared with package UpSetR (version 1.4.0)[146]. For beeswarm graphs of expression levels, package beeswarm (version 0.4.0)[147] was used. Data from Frere et al.[88], was downloaded from the GEO public database as raw counts and normalized with the rlog function (regularized log transformation) in DESeq2.

## Real-time PCR

RNA was isolated from lung tissue as described above. The RNA was then amplified using RT-qPCR (quantitative real-time reverse transcription PCR) targeting the RNA-dependent RdRp and the subgenomic sequence of the envelope protein (subgenomic E) of SARS-CoV-2. In more detail, isolated RNA was processed with the Luna® Universal One-Step RT-qPCR Kit (NEB #E3005, New England Biolabs GmbH) in a CFX96-Touch Real-Time PCR system (Bio-Rad). The following RdRp-targeting primers were used: SARS-2-IP4 forward (5′-GGTAACTGGTA TGATTTCG-3′), reverse (5′-CTGGTCAAGGTTAATATAGG-3′), and probe (5′-TCATACAAACCACGCCAGG-3′ [5′ FAM, 3′ BHQ-1]). Subgenomic E was amplified using the following primers: Subgenomic leader sequence sgLead-SARS-CoV-2 forward (5′-CGATCTCTTGTAGATCGTTCTC-3′), E_Sarbeco reverse (3′-ATATTGCAGCAGTACGCACACA-5′), and probe (5′-ACACTAGCCATCCTTACTGCGCTTCG-3′ [5′ FAM, 3′ BBQ]). The PCR program includes the reverse transcription at 55 °C for 10 minutes with a subsequent denaturation at 95 °C for 1 minute. This step is followed by 44 cycles of denaturation (95 °C, 20 seconds) and annealing and elongation (56 °C, 30 seconds) of the samples. Relative fluorescence units (RFUs) were measured at the end of the elongation phase. A standard RNA transcript was used to convert the obtained Ct values into copy numbers per μl of total RNA. Ct values > 37 were evaluated as negative results.

## Serum analysis

Analysis of oxidative stress was carried out in serum samples. Blood was collected at necropsy from the abdominal aorta into Eppendorf tubes and left to clot for RT. Following centrifugation, serum was collected in cryo tubes and stored at −80 °C.

**Enhancer of filamentation 1, Peroxiredoxin-3.** To determine the amount of EF1 and Prdx3 in serum samples, a quantitative sandwich ELISA was used (MyBioSource MBS9364488 and MBS9391144). The kits are based on EF1 antibody-EF1 antigen interactions or PRDX3 antibody-PRDX3 antigen interactions (immunosorbency) and an HRP colorimetric detection system to detect EF1 antigen targets in samples. The assay procedure was performed according to the manufacturers' instructions. After 5 minutes of adding the stop solution, absorbance was measured once at 450 nm with the Tecan Spark Multimode Microplate Reader. Sample ODs were plotted against the standard curve to determine EF1 and Prdx3 contents. The kits have a detection range of 0.25 ng/ml−8 ng/ml (EF1) and 0.625 ng/ml−20 ng/ml (Prdx3).

**Glutathione (GSH/GSSG).** To determine the glutathione content of serum samples a quantitative colorimetric assay kit (MedChemExpress HY-K0311) with a detection range from 1.57 μM−50 μM was used. The kit uses the enzymatic method that utilizes Ellman's Reagent (DTNB) and glutathione reductase (GR). DTNB is able to react with the reduced glutathione and forms a yellow product, which can be measured at 412 nm. The change in the OD is directly proportional to the glutathione concentration. After the addition of the Enzyme/Coenzyme working solution, the absorbance was measured 20 times with intervals of 25 seconds between the readings in a kinetic fashion with the

Tecan Spark Multimode Microplate Reader. The standard curve obtained from the linear fitting of standard samples was used to determine the total glutathione content in samples.

Statistical analysis and graphs design were performed using GraphPad Prism 9.3.1 (GraphPad Software, San Diego, CA, USA) for Windows™. The assumption of normal distribution was tested using the Kolmogorov-Smirnov test. Due to the rejection of the assumption of normal distribution, non-parametric methods were used. Pairwise comparison between SARS-CoV-2-infected hamsters and control group at each time-point were tested with a two-tailed Mann–Whitney $U$ test. Statistical significance was accepted at exact $p$ values of ≤0.05.

### Statistic and reproducibility

The animal experiment, immunolabellings, quantifications, and data analysis were performed once. The sample size was determined with ANOVA and Kruskal–Wallis test. Allocation to experimental groups was randomized. No data was excluded from the analysis. The investigators were not blinded to group allocation during the animal experiment. For morphologic analysis, samples were blinded.

### Reporting summary

Further information on research design is available in the Nature Portfolio Reporting Summary linked to this article.

## Data availability

The data generated in this study are provided in the Supplementary Information and Source Data file. The bulk RNAseq data generated in this study have been deposited in the GEO database under accession code GSE271365. Source data are provided with this paper.

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

## Acknowledgements

The authors are grateful to Julia Baskas, Petra Grünig, Jana-Svea Harre, Caroline Schütz, Kerstin Schöne, and Melanie Woischnik for their excellent technical assistance. Furthermore, the authors would like to thank Henrieke Meyer-Sievers, Katrin Wirz, Claudia Schulz, Monika Berg, Mathias Herberg, Bernd Vollbrecht, and Darren Markillie for their assistance during the animal experiment. We thank Cristina-Luminita Baciu from Data Sciences International (DSI) for her excellent technical support regarding the use of FinePointe Respiratory Software. This project was supported by the COVID-19 Research Network of the State of Lower Saxony (COFONI) with funding from the Ministry of Science and Culture of Lower Saxony, Germany (14-76403-184, W.B., M.C., A.V., M.vK.B., K.S., R.G., and F.A.). Additionally, this research was partially supported by the Deutsche Forschungsgemeinschaft (DFG; German Research Foundation, grant number 398066876/GRK 2485/2, L.H., T.S., and G.B.). T.S. is also a recipient of a PhD fellowship from the National Research Fund, Luxembourg (AFR15686728). We acknowledge financial support by the Open Access Publication Fund of the University of Veterinary Medicine Hannover, Foundation.

## Author contributions

The study was designed by F.A., W.B. and M.C. The animal experiments were performed by F.A., M.C., L.H., T.S., K.H., K.M.G., L.M.M. and W.R. The virus stock was prepared, and real-time PCR was performed by A.V., T.T., C.M.zN., L.M.S. and S.C. Histology and immunolabeling was performed and evaluated by F.A., M.C., L.H., G.B., W.B., T.S. and I.Z. The antibody podoplanin was prepared and provided by M.K. and Y.K. Digital image analysis and quantification of immunolabeling were performed by L.H., G.B., D.S. and I.Z. RNA was isolated by T.T., C.M.zN., L.M.S. and S.C. Bulk RNA sequencing, and subsequent data preparation was performed by R.G. and K.S. Serum analysis was performed by C.G., G.A., A.P., M.vK.B., T.H. and F.A. Data analysis and interpretation were performed by F.A., L.H., M.C., D.S., K.S. and R.G. Statistical analysis was performed by K.R., F.A. and M.C. Figures were prepared by L.H., M.C., and K.S. The original draft of the manuscript was written by L.H., M.C., and F.A. The manuscript was reviewed, edited and approved by all authors. Funding was acquired by W.B., M.C., K.S., R.G., A.V. and M.vK.B. The project was jointly supervised by F.A. and W.B.

## Funding

## Competing interests

The authors declare no competing interests.

## Additional information

Laura Heydemann [1,14], Małgorzata Ciurkiewicz [1,14], Theresa Störk [1], Isabel Zdora[1], Kirsten Hülskötter[1], Katharina Manuela Gregor[1], Lukas Mathias Michaely[1], Wencke Reineking [1], Tom Schreiner [1], Georg Beythien [1], Asisa Volz [2,3], Tamara Tuchel[2,3], Christian Meyer zu Natrup[2,3], Lisa-Marie Schünemann[2,3], Sabrina Clever[2,3], Timo Henneck[3,4], Maren von Köckritz-Blickwede [3,4], Dirk Schaudien[5], Karl Rohn[6], Klaus Schughart [7,8], Robert Geffers [9], Mika K. Kaneko [10], Yukinari Kato [10], Carina Gross[11], Georgios Amanakis [11], Andreas Pavlou[12], Wolfgang Baumgärtner [1,14] ✉ & Federico Armando[13,14]

[1]Department of Pathology, University of Veterinary Medicine Foundation, Hanover, Germany. [2]Department of Virology, University of Veterinary Medicine Foundation, Hanover, Germany. [3]Research Center for Emerging Infections and Zoonoses (RIZ), University of Veterinary Medicine Foundation, Hanover, Germany. [4]Department of Biochemistry, University of Veterinary Medicine Foundation, Hanover, Germany. [5]Fraunhofer Institute for Toxicology and Experimental Medicine (ITEM), Hanover, Germany. [6]Department of Biometry, Epidemiology and Data Management, University of Veterinary Medicine Foundation, Hanover, Germany. [7]Department of Microbiology, Immunology and Biochemistry, University of Tennessee Health Science Center, Memphis, TN, USA. [8]Institute of Virology Münster, University of Münster, Münster, Germany. [9]Helmholtz Centre for Infection Research (HZI), Brunswick, Germany. [10]Department of antibody drug development, Tohoku University, Sendai, Miyagi, Japan. [11]Department of Cardiology and Angiology, Hanover Medical School (MHH), Hanover, Germany. [12]Institute for Experimental Infection Research, TWINCORE, Centre for Experimental and Clinical Infection Research, a joint venture between the Helmholtz Centre for Infection Research and the Hannover Medical School, Hannover, Germany. [13]Pathology Unit, Department of Veterinary Science, University of Parma, Parma, Italy. [14]These authors contributed equally: Laura Heydemann, Małgorzata Ciurkiewicz, Wolfgang Baumgärtner, Federico Armando. ✉e-mail: Wolfgang.baumgaertner@tiho-hannover.de

