## [Transparent Peer Review file · Nature Communications]

Respiratory long COVID in aged hamsters features impaired lung function post-exercise with bronchiolization and fibrosis

Corresponding Author: Professor Wolfgang Baumgärtner

Version 0:

Reviewer comments:

Reviewer #1

(Remarks to the Author)

This manuscript reports a very interesting in-depth long-term study on clinical aspects as well as the pathological and functional processes in the lung after intranasal SARS-CoV-2 infection, over a prolonged time course, from 1 to 112 dpi, using the Syrian hamster model of COVID-19. This model has been shown to be suitable to investigate the pulmonary response to SARS-CoV-2 infection in humans.

The study has taken a multidirectional approach, combining model-adapted clinical with morphological, transcriptional and translational approaches. It provides very interesting results that offer descriptive and mechanistic explanations for several of the respiratory aspects of long COVID in human patients. It is of substantial significance not only for the work on long COVID but potentially also for other respiratory virus infections and pulmonary regenerative and reparative processes in general. In its depth and multifaceted approach, it is rather unique and adds substantially to the existing literature on COVID-19.

The methodology is sound, the work meets the expected standards in the field; the material and methods section with the respective supplementary material provides sufficient detail to allow reproduction of the study. Results are appropriately described, illustrated and discussed, and any conclusions drawn are well founded.

There are a few language issues, typos and inconsistencies which require editing; these are not specifically commented on. Some specific comments are listed in a consecutive order (NB: Line numbers refer to the PDF document.); please see below:

Introduction

Line 61-63: Suggest to change the sentence to past tense (“declined”, “shifted”...).

Line 63: Reword as symptoms are not really observed in organ systems but are rather based on changes in organ systems.

Line 93: Replace “pathology” by “pathological changes” or “pathological features” or similar.

Line 94: Reword “most diffuse variants”.

Results

Line 128: Replace “symptoms” by “(clinical) signs” here and whenever referring to clinical features in hamsters, as the term “symptoms” is generally only used for human patients/medicine.

Line 147: Suggest to replace “mock animals” by “mock infected animals” here and whenever the term might have been used in the manuscript.

Line 159: Not sure “could be underlined” is the best wording here. Suggest to reword.

Line 170: Replace “pathology” by “lesions” or similar.

Line 192: Since the clinical examination alone does not prove that the pulmonary changes underlying long COVID and the clinical signs in the hamsters are identical, it might be better to resist the use of “confirmed” and write s.th. along the lines of “provide further evidence”.

Lines 246-247: The statements “cytokeratin 14 (Krt14 gene), mainly expressed by airway basal cells” and “Scgb3a2, Scgb3a1 genes mainly expressed by club cells” should each be followed by a reference.

Line 253: Reword that Sox9 “was exclusive at 56 dpi” (i.e. complete the sentence).

Lines 254/255: Although it is more or less common knowledge, the statement “... Mmp9 and Mmp25, which are involved in extracellular matrix remodeling/deposition” should be followed by a reference. Similarly, references should also be added to the statements in lines 270-275 as well as in lines 281/282.

Line 276: Delete “gene”.

Line 332: Reword “alveoli and airways cells” to make clear that this refers to alveolar and airway epithelial cells.

Line 333: Make clear what “peak” refers to (presumably number of pos cells).

Line 334: Specify what the conclusion “virus clearance was completed” at 6 dpi is based upon, in particular when considering that “SARS-CoV-2 S and N transcripts ... were marginally present until 6 dpi” (line 338).

Line 335: Specify what “Mock animals showed no positive signal at any time-point” is based upon.

Lines 367/368: A reference should be added to the statement “since these changes ... could be associated with long lasting functional impairment.” or it should be reworded to make it sound more like a hypothesis.

Line 439: Delete the comma after “Rare”.

Line 450: Have the abbreviations “AT1” and “AT2” been introduced and their meaning been explained??

Line 461: Reword “about ADI cells presence”.

Line 473: Replace “do” by “does”.

Line 537: Suggest to reword “multipotent potential”.

Line 557 and 558: Delete “gene”.

Line 636: Replace “far from being” by “not”.

Line 638/639: Reword “immediately connected” and specify “this organ”.

Line 643: Replace “histopathology” by “histological examination”.

Discussion

Line 719: Replace “display a” by “cause” or similar.

Line 757: Reword “this change recovered”.

Line 769: Reword “persistent inflammatory foci in the alveoli”.

Line 771: Replace “wih” by “with”.

Lines 794-797: Please check the sentence, as a verb is lacking.

Lines 819-820: Please check and reword the sentence.

Materials and methods

Line 904: Please check the sentence (“manually exclude”).

Line 925: Replace “Histopathology” by “Histological examination”.

Line 926: Replace “histopathological” by “histological”. Also, it is suggested to specify how the left lung lobe was trimmed for the histological examination (cross or longitudinal sections?). Furthermore, it is preferred to state here that the left lung was examined (move up from line 934)

Line 937: Does it really make sense to state “special emphasis on ... epithelial regenerative processes” in this context?

Line 956: Please state that the IHC was done on consecutive (?) FFPE sections from all/selected (?) lungs (and other tissues?).

Lines 958-960: Check for correct placement and inclusion of commata.

Line 974: Please state that IF was done on consecutive (?) FFPE sections from all/selected (?) lungs (and other tissues?).

Line 976/977: List which markers were used to determine “CK14+ basal cells fate, cell cycle activity, and cell senescence”.

Acknowledgements

Line 1087/1088: Please check the structure/sentence.

Reviewer #2

(Remarks to the Author)

The authors intended to study the post-acute sequelae of COVID-19 (PASC), also referred to as “long COVID,” by characterizing the post-acute sequelae of SARS-CoV-2 infection in a hamster model that they have previously characterized in the acute setting out to 14 days post infection (dpi), but this time out to 112 dpi. They did this through serial evaluation of lung function using whole body plethysmography (WBP) plus respiratory gas analysis (pre- and post-exercise in the chronic phase), lung tissue histology, and lung tissue bulk transcriptomics through 112 dpi. They used only male, aged (~1 year) hamsters for these experiments, their justification being that older and male hamsters are more adversely affected by SARS-CoV-2 infection, which was also true in humans during the pandemic. In this study, they again observed the animals developed acute lung impairment, characterized by peribronchiolar epithelial proliferation derived from airway progenitors and cytokeratin 8 (CK8+) alveolar differentiation intermediate (ADI) cells (intermediate between the AT2-to-AT1 transition), which resolved by 10 dpi. Beyond that timepoint, altered lung function is only detected after exercise through 7 weeks (49 dpi), but histomorphological findings can persist out to 16 weeks (112 dpi), including sub-pleural and interstitial fibrosis as well as bronchiolization, associated with prolonged presence of M2-like macrophages, pro-fibrotic transcriptomic changes, and Ki67 expression consistent with ongoing proliferative activity. Notably, they observe that while the hamster model can emulate a protracted lung injury phenotype, it does not manifest any extra-pulmonary sequelae.

In terms of key findings to highlight, they observed:

- Acute lung function impairments in the infected animals including decreased respiratory rate (f) and mid-expiratory flow (EF50) and an increased enhanced pause (Penh), suggestive of abnormal bronchiolar airflow, and decreased O₂ uptake and CO₂ production, suggesting either decreased gas exchange capacity, perfusion, or metabolic activity. The changes in both WBP mechanics and gas analyses resolved after 10 dpi.
- After the resolution of the acute infection, the animals did not show any respiratory or behavioral signs compared to the mock animals.
- After 21 dpi, the animals were exercised and they found the infected animals had a lower RR, increased Penh, and

decreased EF50 after exercise with the most abnormal difference in O₂ uptake and CO₂ production (both decreased), all of which resolved around 50 dpi.

- They performed bulk RNA sequencing of lung tissues at multiple time points from acute infection out to 112 dpi and analyzed DEGs at each time point to perform pathway enrichment analysis. Notable findings include the expected upregulation of immune activation and signaling pathways in the early phase, with transition to ECM deposition and remodeling in the chronic phase. This will be a key resource for investigators to be able to explore.
- At the histological level, viral antigen was present in the alveoli airway cells and alveolar macrophages and cleared by 6 dpi, confirmed by RNAseq. Peak lung injury was 6 dpi which associated with the peak M1-like macrophage signature. After 28 dpi, histological abnormalities were minimal across animals, with the most prominent features being areas of alveolar epithelial proliferation (“bronchiolization”) and sub-pleural fibrosis. Overall lung tissue in the infected animals showed increased Azan+ collagen deposition in the alveolar septa from 14 to 112 dpi with increased pro-fibrotic M2-like macrophages, though no longer significant by 112 dpi.
- CK8+ ADI cells are abundantly expressed in the alveolar epithelial proliferation foci, peaking at 6 dpi and slowly decreasing through 112 dpi, which was corroborated by analyzing the ADI cell-specific gene transcriptome with variable down-regulation over time observed. Overall, this shows ADI cells are prominent features of alveolar regeneration after SARS-CoV-2 infection.
- Within the alveolar epithelial proliferation foci, airway progenitors (CK14+ basal cells and SCGB1A1+ club cells) were the dominant cell types, with the CK14+ basal in the earlier chronic phase and the SCGB1A1+ club cells predominating in the later bronchiolization foci to 112 dpi. Further, both basal and club cell types demonstrated Ki67 expression through 112 dpi suggesting ongoing proliferative activity.
- The investigators evaluated for extra-pulmonary sequelae, but observed no histological changes in the heart, liver, kidneys, and spleen compared to mock animals.

The authors are commended for completing this comprehensive phenotyping of post-acute respiratory sequelae of SARS-CoV-2 infection in hamsters. There are several strengths, including the lung function and respiratory gas analysis pre- and post-exercise in the post-acute and chronic phase, and the rich transcriptional dataset from lung tissue over the protracted phase of recovery which will be a resource for the research community. However, there are several significant limitations that would need to be addressed by the authors to make this a complete work that offers novel insight into PASC.

Major comments

1. The authors specifically chose to use only male, aged hamsters because human males were disproportionately affected by acute critical illness related to COVID in the early phase of the pandemic, and hence why males were reported to have worse post-ARDS fibrotic sequelae compared to females (reference 55). However, epidemiologically, females are more likely than males to experience PASC (PMID 36951832). Thus, to be complete, the authors need to include female hamsters in this study, especially since we do not know if the extra-pulmonary sequelae may have been observable in female vs. male hamsters. This is addressed as a weakness in the manuscript, but this certainly seems like one that can be surmounted.
2. PASC is a heterogeneous disorder. The authors narrow down on respiratory PASC, but clinically, respiratory PASC can range from organizing lung injury and post-ARDS fibrotic sequelae to bronchiolitis, phenotypes that can be observed following many viral causes of lung injury. Thus, to be complete, the authors should compare the post-acute sequelae of another viral agent, such as influenza, to better characterize post-viral syndromes.
3. While the animals did not demonstrate observable behavioral differences in the post-acute phase, it is not clear that formal assessments of memory, attention, and behavior were performed, nor of more advanced neurologic phenotyping, as conducted in a rodent model of long COVID, which demonstrated objective differences in these parameters that would not otherwise be observable (PMID 37848036).

Minor comments

1. Penh is a flawed measure of pulmonary function in animal models, and it is largely recommended to avoid reporting it in studies (PMID 15317683).
2. Can the authors please provide a rationale for why exercise precipitates a decline in respiratory rate in the infected animals? This is incongruous from what we observe in some PASC patients, where there is often ventilatory inefficiency (hyperventilation relative to the VCO₂) at rest and with exercise, though a recent meta-analysis is less conclusive on the matter (PMID 39022666).
3. Similarly, can the authors please provide a rationale for why exercise precipitates a decline in EF50? That would suggest an increase in dynamic airways resistance, but the histologic lesions described in these animals are not airway-centered or suggestive of bronchiolitis.
4. The observation of decreased O₂ uptake and CO₂ production suggests more of an impairment in the mitochondria (or muscle mass in the infected animals), though abnormal dead space ventilation could contribute. Understanding the metabolic derangements through metabolomics, mitochondrial and skeletal muscle density, and assessments of mitochondrial function in vitro would add rigor to this assessment. Further, the pulmonary vasculature is abnormal in patients with acute COVID and is associated with increased dead space ventilation (PMID 35671465, 35584345) as well as in patients with PASC, even without preceding critical illness (PMID 38500738, 37094097), and micro-CT of the hamster lungs to assess for micro- and macro-vascular changes would also be valuable.
5. Prior studies have shown dysregulated CD8+ T cells in the lung and GI tract are highly activated compared to circulating T cells and associate with organ-specific dysfunction, suggesting that organ-specific PASC may be due to dysregulated local immune cells (PMID 34591653). It may be worth considering FACS on the bronchoalveolar lavage fluid compared to circulating blood to assess the burden of locally dysregulated CD8+ T cells.

Reviewer #3

(Remarks to the Author)

NCOMMS-24-37593

The authors utilize a hamster model of COVID-19 to demonstrate chronic or long term effects of infection on lung pathology and on the activity of the infected animals. Besides the observation of pulmonary fibrosis, the authors demonstrate a shift from alveolar epithelial cell progenitors to club cell progenitors, further demonstrating unique, long term changes in lung pathology that may contribute to prolonged pulmonary dysfunction. A particular strength of the study is the inclusion of exercise-induced physiological measurements. A particular weakness of the study is its use of only male animals, as that makes the study conclusions only relevant to males. A final shortcoming is the lack of a control group infected with another respiratory virus, which would have strengthened the conclusions that the observed changes in pathology were specific to COVID-19 and not a result of infection with any respiratory virus. There are very interesting observations that come from the clear and detailed analysis of the data, but the shortcomings do impact significance.

While I understand that using males only skews the population to ones that have more severe disease, it does limit the impact of the study because any conclusions drawn can only be applied to males.

I realize this is a critique that involves significant expense, but the lack of a control group infected with another virus prevents the authors from stating that the pulmonary effects and damage they see are specific to COVID and not a result of infection with any respiratory virus.

Lines 337-339 - Interesting. Others speculate that long term persistence of viral RNA contributes to chronic disease. More discussion of this is needed.

Lines 358-386 - Technically, if its not statistically significant then its not really increased.

Version 1:

Reviewer comments:

Reviewer #1

(Remarks to the Author)

The manuscript has improved further with revision in particular also due to the additional work done (like the PCR for RdRP and Esub, ad the comparison with IAV infection) which adds further value. This is indeed a very interesting, relevant and impressive piece of work! As a consequence of the additions the discussion is very extensive, and it is the reviewer's impression that it is a bit redundant in places...

There are some editing issues, typos, spaces, capital letters, hyphens etc., and commata in wrong places; this is likely a consequence of the extensive changes made to the manuscript and can be sorted easily by a thorough final clean-up. Some specific comments are listed in a consecutive order (NB: Line numbers refer to the Word document with Track Changes); please see below:

Introduction

Line 71: Delete comma after "persistent".

Line 74/75: The authors might consider to refer to the term suggested by the WHO: "post-COVID-19 condition" as the most relevant/suitable one?

Results

Line 172: Change to "differences".

Line 189/190: Suggest to reword as follows "The data indicate slower and partially deeper breathing in SARS-CoV-2-infected hamsters,...".

Line 230: It might be good to say again what "frequency" refers to.

Line 237: Maybe replace "until 112 dpi" by "thereafter"?

Line 564: Replace "it" by "CK8".

Lines 807: Delete "marked"?

Lines 863 and 866: Change to "cell".

Discussion

Line 975: Suggest to replace "... by marked transient".

Line 962/963: It is not clear what you want to emphasise here, that patients report dyspnoe or coughing during both the acute and chronic phase, or that patients report dyspnoe or coughing not only during the acute phase of COVID-19 but also in long COVID. Suggest to reword.

Line 996: Delete "in".

Line 1004: Delete "the".

Lines 1012/1013: Suggest to move "again" after "we"..

Lines 1020: Change to "histologically observed foci".

Line 1033/1034: Change to "spent" or "were spending".

Line 1117: Changes to "sampled for a transcriptome" or similar.

Line 1135: Suggest to reword and not speak about "positive signal" when referring to IHC (but rather restrict this to, eg. ISH signals).

Line 1159: Change to "cell".

Line 1181: Delete one "had".

Line 1244: Add "infection" after "virus".

Line 1246: Are you really talking about "repair" here and not about "regeneration" or both?

Materials and methods

Line 1471: Replace "of" by "for".

Reviewer #2

(Remarks to the Author)

The authors have revised their initial manuscript in which they aim to study the post-acute sequelae of COVID-19 (PASC), also referred to as "long COVID," by characterizing the post-acute sequelae of SARS-CoV-2 infection in a hamster model that they have previously characterized in the acute setting out to 14 days post infection (dpi), but this time out to 112 dpi. To reiterate, there are several strengths, including the lung function and respiratory gas analysis pre- and post-exercise in the post-acute and chronic phase, and the rich transcriptional dataset from lung tissue over the protracted phase of recovery which will be a resource for the research community.

The authors have responded in detail to my comments on their first draft. While they were unable to conduct experiments to address the limitations of only using male, aged hamsters (major comment 1) or only one viral acute lung injury model (major comment 2), they did provide substantial justification and included new secondary analyses to support their choices. They also included new data to address my comments regarding more detailed behavioral assessment (major comment 3) in addition to all the minor comments. Overall, the authors were appropriately responsive and improved the quality of the manuscript.

In terms of actionable changes:

With respect to their response to my minor comment regarding dysregulated T cells which they address on page 32 of their Response to Review, I agree that the limitations should preclude inclusion in the manuscript. They should feel free to remove the provisional revision, though they can keep their findings regarding Plet-1 if desired.

Otherwise, I might just modify the statement to be more declarative that future studies should (instead of "might have to explore") whether there are sex-specific differences in the long-term consequences of infection in the hamster model (which would provide insight into sex-based differences in resilience to post-viral sequelae) (page 49, lines 1245-47).

Reviewer #3

(Remarks to the Author)

The primary critiques I made - the use of only male hamsters and the lack of a control group infected with a respiratory pathogen not associated with "long respiratory COVID" - have not been directly addressed. I realize these are not trivial critiques but they are important critiques nonetheless. The authors spend much time in their response arguing for why females and other respiratory infections are not needed controls, but those are not convincing arguments. Sex differences in long COVID exist and should be factored into analyses. The fact that the authors argue that they never claim the differences they see are "SARS-CoV-2 specific" and would refrain from doing so is a bit laughable, as the only variable they are testing is SARS-CoV-2 infection compared to mock infected animals. All their results are SARS-CoV-2 specific because of their experimental design.

line 3 (title) - insert "male" before "hamster"

line 46,50 - insert "male" before "hamster"

Reviewer comments

Reviewer #1 (Remarks to the Author):

This manuscript reports a very interesting in-depth long-term study on clinical aspects as well as the pathological and functional processes in the lung after intranasal SARS-CoV-2 infection, over a prolonged time course, from 1 to 112 dpi, using the Syrian hamster model of COVID-19. This model has been shown to be suitable to investigate the pulmonary response to SARS-CoV-2 infection in humans.

The study has taken a multidirectional approach, combining model-adapted clinical with morphological, transcriptional and translational approaches. It provides very interesting results that offer descriptive and mechanistic explanations for several of the respiratory aspects of long COVID in human patients. It is of substantial significance not only for the work on long COVID but potentially also for other respiratory virus infections and pulmonary regenerative and reparative processes in general. In its depth and multifaceted approach, it is rather unique and adds substantially to the existing literature on COVID-19.

The methodology is sound, the work meets the expected standards in the field; the material and methods section with the respective supplementary material provides sufficient detail to allow reproduction of the study. Results are appropriately described, illustrated and discussed, and any conclusions drawn are well founded.

There are a few language issues, typos and inconsistencies which require editing; these are not specifically commented on. Some specific comments are listed in a consecutive order (NB: Line numbers refer to the PDF document.); please see below:

RESPONSE: the authors are thankful to Reviewer 1 for their time and the effort taken to revise our manuscript. We are particularly thankful for the appreciations and for the thorough revision and suggestions. We hope that the revised version of our manuscript will meet Reviewer 1 approval.

Introduction

Line 61-63: Suggest to change the sentence to past tense (“declined”, “shifted”...).

Line 63: Reword as symptoms are not really observed in organ systems but are rather based on changes in organ systems.

Line 93: Replace “pathology” by “pathological changes” or “pathological features” or similar.

Line 94: Reword “most diffuse variants”.

RESPONSE: the authors are thankful to Reviewer 1 for their thorough revision and the insightful suggestions that improved the quality of our manuscript. All the reported lines have been revised in the main text accordingly.

Results

Line 128: Replace “symptoms” by “(clinical) signs” here and whenever referring to clinical features in hamsters, as the term “symptoms” is generally only used for human patients/medicine.

Line 147: Suggest to replace “mock animals” by “mock infected animals” here and whenever the term might have been used in the manuscript.

Line 159: Not sure “could be underlined” is the best wording here. Suggest to reword.
 Line 170: Replace “pathology” by “lesions” or similar.
 Line 192: Since the clinical examination alone does not prove that the pulmonary changes underlying long COVID and the clinical signs in the hamsters are identical, it might be better to resist the use of “confirmed” and write s.th. along the lines of “provide further evidence”.
 Lines 246-247: The statements “cytokeratin 14 (Krt14 gene), mainly expressed by airway basal cells” and “Scgb3a2, Scgb3a1 genes mainly expressed by club cells” should each be followed by a reference.
 Line 253: Reword that Sox9 “was exclusive at 56 dpi” (i.e. complete the sentence).
 Lines 254/255: Although it is more or less common knowledge, the statement “... Mmp9 and Mmp25, which are involved in extracellular matrix remodeling/deposition” should be followed by a reference. Similarly, references should also be added to the statements in lines 270-275 as well as in lines 281/282.

Line 276: Delete “gene”
 Line 332: Reword “alveoli and airways cells” to make clear that this refers to alveolar and airway epithelial cells.
 Line 333: Make clear what “peak” refers to (presumably number of pos cells).
 Line 334: Specify what the conclusion “virus clearance was completed” at 6 dpi is based upon, in particular when considering that “SARS-CoV-2 S and N transcripts ... were marginally present until 6 dpi” (line 338).
 Line 335: Specify what “Mock animals showed no positive signal at any time-point” is based upon.
 Lines 367/368: A reference should be added to the statement “since these changes ... could be associated with long lasting functional impairment.” or it should be reworded to make it sound more like a hypothesis.
 Line 439: Delete the comma after “Rare
 Line 450: Have the abbreviations “AT1” and “AT2” been introduced and their meaning been explained??
 Line 461: Reword “about ADI cells presence
 Line 473: Replace “do” by “does”.
 Line 537: Suggest to reword “multipotent potential”
 Line 557 and 558: Delete “gene”.
 Line 636: Replace “far from being” by “not”.
 Line 638/639: Reword “immediately connected” and specify “this organ”.
 Line 643: Replace “histopathology” by “histological examination”

RESPONSE: the authors are thankful to Reviewer 1 for their thorough revision and the insightful suggestions that improved the quality of our manuscript. All the reported lines have been revised in the main text accordingly.

Discussion

Line 719: Replace “display a” by “cause” or similar.
 Line 757: Reword “this change recovered”.
 Line 769: Reword “persistent inflammatory foci in the alveoli”
 Line 771: Replace “wih” by “with”
 Lines 794-797: Please check the sentence, as a verb is lacking.
 Lines 819-820: Please check and reword the sentence.

RESPONSE: the authors are thankful to Reviewer 1 for their thorough revision and the insightful suggestions that improved the quality of our manuscript. All the reported lines have been revised in the main text accordingly.

Materials and methods

Line 904: Please check the sentence (“manually exclude”).

Line 925: Replace “Histopathology” by “Histological examination”.

Line 926: Replace “histopathological” by “histological”. Also, it is suggested to specify how the left lung lobe was trimmed for the histological examination (cross or longitudinal sections?). Furthermore, it is preferred to state here that the left lung was examined (move up from line 934)

Line 937: Does it really make sense to state “special emphasis on ... epithelial regenerative processes” in this context?

Line 956: Please state that the IHC was done on consecutive (?) FFPE sections from all/selected (?) lungs (and other tissues?).

Lines 958-960: Check for correct placement and inclusion of commata

Line 974: Please state that IF was done on consecutive (?) FFPE sections from all/selected (?) lungs (and other tissues?).

Line 976/977: List which markers were used to determine “CK14+ basal cells fate, cell cycle activity, and cell senescence”.

RESPONSE: the authors are thankful to Reviewer 1 for their thorough revision and the insightful suggestions that improved the quality of our manuscript. All the reported lines have been revised in the main text accordingly.

Acknowledgements

Line 1087/1088: Please check the structure/sentence.

RESPONSE: the authors are thankful to Reviewer 1 for their thorough revision and the insightful suggestions that improved the quality of our manuscript. The section has been revised in the main text accordingly.

Reviewer #2 (Remarks to the Author):

The authors intended to study the post-acute sequelae of COVID-19 (PASC), also referred to as “long COVID,” by characterizing the post-acute sequelae of SARS-CoV-2 infection in a hamster model that they have previously characterized in the acute setting out to 14 days post infection (dpi), but this time out to 112 dpi. They did this through serial evaluation of lung function using whole body plethysmography (WBP) plus respiratory gas analysis (pre- and post-exercise in the chronic phase), lung tissue histology, and lung tissue bulk transcriptomics through 112 dpi. They used only male, aged (~1 year) hamsters for these experiments, their justification being that older and male hamsters are more adversely affected by SARS-CoV-2 infection, which was also true in humans during the pandemic. In this study, they again observed the animals developed acute lung impairment, characterized by peribronchiolar epithelial proliferation derived from airway progenitors and cytokeratin 8 (CK8+) alveolar differentiation intermediate (ADI) cells (intermediate between the AT2-to-AT1 transition), which resolved by 10 dpi. Beyond that timepoint, altered lung function is only detected after exercise through 7 weeks (49 dpi), but histomorphological findings can persist out to 16 weeks (112 dpi), including sub-pleural and interstitial fibrosis as well as bronchiolization, associated with prolonged presence of M2-like macrophages, pro-fibrotic transcriptomic changes, and Ki67 expression consistent with ongoing proliferative activity. Notably, they observe that while the hamster model can emulate a protracted lung injury phenotype, it does not manifest any extra-pulmonary sequelae.

In terms of key findings to highlight, they observed:

- Acute lung function impairments in the infected animals including decreased respiratory rate (f) and mid-expiratory flow (EF50) and an increased enhanced pause (Penh), suggestive of abnormal bronchiolar airflow, and decreased O₂ uptake and CO₂ production, suggesting either decreased gas exchange capacity, perfusion, or metabolic activity. The changes in both WBP mechanics and gas analyses resolved after 10 dpi.
- After the resolution of the acute infection, the animals did not show any respiratory or behavioral signs compared to the mock animals.
- After 21 dpi, the animals were exercised and they found the infected animals had a lower RR, increased Penh, and decreased EF50 after exercise with the most abnormal difference in O₂ uptake and CO₂ production (both decreased), all of which resolved around 50 dpi.
- They performed bulk RNA sequencing of lung tissues at multiple time points from acute infection out to 112 dpi and analyzed DEGs at each time point to perform pathway enrichment analysis. Notable findings include the expected upregulation of immune activation and signaling pathways in the early phase, with transition to ECM deposition and remodeling in the chronic phase. This will be a key resource for investigators to be able to explore.
- At the histological level, viral antigen was present in the alveoli airway cells and alveolar macrophages and cleared by 6 dpi, confirmed by RNAseq. Peak lung injury was 6 dpi which associated with the peak M1-like macrophage signature. After 28 dpi, histological abnormalities were minimal across animals, with the most prominent features being areas of alveolar epithelial proliferation (“bronchiolization”) and sub-pleural fibrosis. Overall lung tissue in the infected animals showed increased Azan+ collagen deposition in the alveolar septa from 14 to 112 dpi with increased pro-fibrotic M2-like macrophages, though no longer significant by 112 dpi.
- CK8+ ADI cells are abundantly expressed in the alveolar epithelial proliferation foci, peaking at 6 dpi and slowly decreasing through 112 dpi, which was corroborated by analyzing the ADI cell-specific gene transcriptome with variable down-regulation over time observed. Overall, this shows ADI cells are prominent features of alveolar regeneration after SARS-CoV-2 infection.
- Within the alveolar epithelial proliferation foci, airway progenitors (CK14+ basal cells and SCGB1A1+

club cells) were the dominant cell types, with the CK14+ basal in the earlier chronic phase and the SCGB1A1+ club cells predominating in the later bronchiolization foci to 112 dpi. Further, both basal and club cell types demonstrated Ki67 expression through 112 dpi suggesting ongoing proliferative activity.

- The investigators evaluated for extra-pulmonary sequelae, but observed no histological changes in the heart, liver, kidneys, and spleen compared to mock animals.

The authors are commended for completing this comprehensive phenotyping of post-acute respiratory sequelae of SARS-CoV-2 infection in hamsters. There are several strengths, including the lung function and respiratory gas analysis pre- and post-exercise in the post-acute and chronic phase, and the rich transcriptional dataset from lung tissue over the protracted phase of recovery which will be a resource for the research community. However, there are several significant limitations that would need to be addressed by the authors to make this a complete work that offers novel insight into PASC.

RESPONSE: The authors are thankful to Reviewer 2 for their time and the effort taken to revise our manuscript. We are particularly thankful for the insightful comments that helped us to improve the scientific value of our manuscript. We have tried to comply with all the requests made by Reviewer 2. Where not possible, we provided a rationale or a justification (please see point by point). Unfortunately, some points could not be addressed exactly as Reviewer 2 proposed, due to initial study design issues that prevented us to comply with the request. However, we offered, when possible, alternative solutions. Thanks to Reviewers comments, we believe that the revised version of this manuscript is of improved value for the scientific community and we hope that will meet Reviewer 2 approval.

1. The authors specifically chose to use only male, aged hamsters because human males were disproportionately affected by acute critical illness related to COVID in the early phase of the pandemic, and hence why males were reported to have worse post-ARDS fibrotic sequelae compared to females (reference 55). However, epidemiologically, females are more likely than males to experience PASC (PMID 36951832). Thus, to be complete, the authors need to include female hamsters in this study, especially since we do not know if the extra-pulmonary sequelae may have been observable in female vs. male hamsters. This is addressed as a weakness in the manuscript, but this certainly seems like one that can be surmounted.

We thank Reviewer 2 for their comments. However, we would like to explain in more detail our choice of male animals and refine some statements in the main text that might have been misunderstood due to lack of clarity. We believe that our rationale to only include male old hamsters was justified considering the aims of this project. Our initial concept (the study was designed in the 2021 and performed in 2022) was to offer a model for a population at high risk of respiratory long-COVID, namely people that experienced severe acute COVID-19, which are older male patients. Old age and male sex are considered to be risk factors for severe acute-COVID19 (PMID: 34454673; PMID: 35891696; PMID: 38652535; PMID: 32411652; PMID: 33298944). Severe acute COVID-19 was at that time and still is today considered a risk factor for respiratory long-COVID (PMID: 35891696; PMID 35429399; PMID: 34454673; PMID: 38652535). According to the current literature, female sex is considered to be a risk factor for long-COVID in general (PMID: 33692530; PMID: 36626183; PMID 35429399; PMID: 36555931). However, it should be noted that most clinical studies, reviews and meta-analyses do not specifically investigate respiratory long-COVID but focus instead on the overall presence of PASC in any

organ system (long-COVID *in sensu lato*), with some of the most frequently reported symptoms being extra-pulmonary (e.g. brain fog and fatigue), rather than only respiratory (e.g. shortness of breath, typical of respiratory long-COVID). In contrast, some studies that are focusing on respiratory long-term sequelae, have found that especially disease severity, but also male sex and age are risk factors for developing respiratory PASC (PMID: 38652535). When looking at the disease severity and pulmonary lesions in the Syrian golden hamster model for SARS-CoV-2, the literature reports male hamsters as being more severely affected in the acute stage (PMID: 34253053; PMID: 33790236; PMID: 36851642; PMID: 34870132). The few long-term studies extending beyond the first 2 weeks also report males as recovering more slowly (PMID: 37572667; PMID: 34253053; PMID: 34870132). One of these studies (PMID: 37572667), where some of us are co-authors, demonstrated that males (humans and hamsters) have a worse outcome of the disease than females up to 21 dpi, comparing lung function and histopathology of lung lesion. Therefore, since our focus was respiratory long-COVID and not long-COVID *in sensu lato*, and since the probability of more severe pulmonary disease with functional impairment was higher in male hamsters, we consider the choice of aged males is substantiated by the aforementioned references. We have prepared a revised version of the manuscript, outlining the reasoning for the choice of males and the resulting limitations more clearly, and including a modified title underlining that our main focus is respiratory long-COVID and not long-COVID *in sensu lato*. We believe that the manuscript provides novel and highly valuable information that deserves to be made available to the scientific community in a reasonable time, given the relevance of the long-COVID situation. We feel that the significance of the manuscript is not compromised by the lack of females, since it offers a comprehensive and extensive investigation relevant for a very large group of patients and has several strengths. We sincerely hope that Reviewer 2 will accept our rationale and consider the revised manuscript for publication.

Based on this the **title**, **introduction** and **discussion** have been revised accordingly in the manuscript as follows:

TITLE

Persistent alveolar bronchiolization, interstitial fibrosis and impaired lung function post-exercise are features of respiratory long-COVID in SARS-CoV-2-Delta variant-infected aged hamsters

INTRODUCTION

new line number 98-108

Only male, aged (~ 1-year-old) hamsters were chosen for the study. The rationale behind this choice was that we wanted to characterize in detail a respiratory long COVID-19 model for a better understanding of the pathogenetic processes associated with this syndrome. The choice of our study design was based on the following observations i) old age and male sex are risk factors for severe acute COVID-19^{8,19-23}, ii) severe acute disease is a risk factor for respiratory long-COVID^{8,21,23,24}, and iii) male hamsters show more severe disease course with more prominent lung function alterations and slower recovery, as well as more severe histological pulmonary lesions compared to females^{22,25-30}. SARS-CoV-2 Delta-variant was chosen since it showed the most prominent pulmonary pathological changes among the most common variants worldwide at the time^{31,32}.

DISCUSSION

new line number 1234-1248

Second, we used only old male animals. We are aware that in an optimal condition the use of male and female as well as young and aged hamsters would have been preferable and that the results of this study cannot be extrapolated to females and young hamsters. However, our aim was to offer a

model for a population at high risk of respiratory long-COVID. Risk factors for the development of respiratory long-term sequelae include severe acute disease^{8,21,23,137,138}, advanced age^{3,139} and sex, with males being more likely to develop severe acute disease^{8,20-23,26}. Moreover, male sex is associated with higher risk for respiratory long-COVID with diffusion impairment and restriction in humans²¹. Similarly, male hamsters show more severe disease, lung function impairment, and histological lesions in the acute phase^{26,28-30,67}. The few long-term studies extending beyond the first 2 weeks also report males as recovering more slowly than females^{26,29,30}. Thus, to increase the likelihood of observing long lasting pulmonary long-COVID, we used one year old male hamsters. Future studies might have to explore whether there are sex-specific differences in the long-term consequences of infection in the hamster model.

2. **PASC is a heterogeneous disorder. The authors narrow down on respiratory PASC, but clinically, respiratory PASC can range from organizing lung injury and post-ARDS fibrotic sequelae to bronchiolitis, phenotypes that can be observed following many viral causes of lung injury. Thus, to be complete, the authors should compare the post-acute sequelae of another viral agent, such as influenza, to better characterize post-viral syndromes.**

We thank Reviewer 2 for their comment. We entirely agree with the fact that these phenotypes can be observed following many viral causes of lung injury. As a matter of fact, we did not claim that the changes that we have shown are SARS-CoV-2 specific, and we would refrain from doing so. As stated in our previous work characterizing the pulmonary regeneration response in the acute phase of SARS-CoV-2 infection (PMID: 37277327), we believe that the response overlaps with that of other diseases.

The current understanding is that AT2 cells are mainly responsible for AT1 cell regeneration in homeostatic turnover and following mild injury, while airway progenitors are recruited after severe injury with marked AT1 cell loss (PMID: 31978363; PMID: 3214265; PMID: 25533958). The participation of airway progenitors and ADI cells in alveolar regeneration has been demonstrated in many publications, independent of the virus (e.g. SARS-CoV-2, IAV, Sendai virus) or animal model (hamster, mouse; PMID: 37277327; PMID: 38092883; PMID: 35857629; PMID: 34343135; PMID: 32678092). Moreover, the same processes have been shown to be partaking in alveolar regeneration also in non-infectious mouse models of lung injury (e.g. bleomycin, neonatal hypoxia and hyperoxia, LPS, PMID: 32678092; PMID: 31978363; PMID: 32073903; PMID: 30913038). We have demonstrated previously, and in this manuscript, that homologous patterns of lung regeneration are active in the SARS-CoV-2 infected hamster (PMID: 37277327).

The most comparable infectious disease is influenza A virus (IAV) infection. Hamsters are naturally susceptible to IAV and are considered a good model for the respiratory as well as systemic disease (PMID: 29212926). A number of studies has already been published on the comparisons of SARS-CoV-2 and IAV infection in hamsters (e.g. PMID: 35857629; PMID: 35648595; PMID: 33216851; PMID: 38400021). Most of the works demonstrate a higher disease severity and more pronounced inflammation in SARS-CoV-2 infection compared to IAV infection when using similar infection doses (PMID: 35648595; PMID: 33216851; PMID: 38400021). Apart from these quantitative differences, qualitative morphological features and the antiviral response appear to be comparable between these models. In one published study, with infection of young hamsters with SARS-CoV-2 and IAV with inoculation doses adapted to reach equivalent viral loads, the disease severity, histopathological lesions, inflammatory infiltrates and transcriptomic changes in the lung were qualitatively and quantitatively comparable during acute (3, 7 dpi) and chronic disease (31 dpi; PMID: 35857629). Thus, it appears that hamsters might be more susceptible to SARS-CoV-2 than to IAV, but once the infections doses are adapted to cause the same level of lung damage, the response patterns are the same.

Interestingly, sublethal SARS-CoV-2 infection of K18-hACE2 mice failed to induce airway progenitor cell proliferation in contrast to IAV infection of B6 mice (PMID: 38092883). The lack of airway progenitor cell proliferation has also been highlighted in BALB/c mice infected with the mouse-adapted SARS-CoV-2 strain MA10 (PMID: 35857635). However, airway basal cell proliferation occurs in SARS-CoV-2- and IAV-infected hamsters (PMID: 35857629). Therefore, we believe that the apparent qualitative differences observed are not indicative of a true virus-specific response pattern, but related to the severity of alveolar damage, which in turn is dependent on the virus dose and strain as well as the choice of animal species and age group. It appears that SARS-CoV-2-induced alveolar damage is more severe in hamsters whereas IAV is provoking a more severe damage in mice.

Altogether, since the studies mentioned above already demonstrate that the lung response in hamsters to viral infection appears to be stereotypical, we did not see a strong rationale to include another virus as a control group in the original study design. However, since most of the published work addressed above did not look into epithelial cells participating in alveolar regeneration in detail, we decided to perform additional analyses targeting this aspect using a published RNAseq dataset from the hamster model. We selected the experiment published by Frere et al. (PMID: 35857629), since it showed similar responses to SARS-CoV-2 and IAV infection, contained time-points during acute and chronic disease, and described chronic bronchiolization lesions similar to the changes found in our study. We selected hallmark gene sets that were associated with certain cell types (ADI cells, club cells, airway basal cells) or the pro-fibrotic environment that showed differential expression in different disease phases of our experiment. Then, we compared the relative expression of these genes in SARS-CoV-2 infected and IAV-infected lungs from Frere et al., and identified similarities and differences to our own data. The analysis showed mostly overlapping patterns in the expression of epithelial cell genes, with comparable changes in both viral diseases and both experiments. Some differences between our data and the data from Frere et al. were observed in the pro-fibrotic gene expression in the chronic phase, with lasting changes in our hamster model as opposed to the transient upregulation in the SARS-CoV-2 and IAV-infected hamsters in Frere et al. This discrepancy could be related to the animal age, since we used old males, while the published dataset was obtained in young animals.

Following the suggestion of Reviewer 2 to investigate immune cell interactions (see also response to Point No. 8), we added an additional evaluation of immune cells implicated in pulmonary regenerative processes, that further supports similarities between SARS-CoV-2 and IAV infection. It has been recently demonstrated in a murine IAV model, that an important mediator of macrophage-epithelial cross-talk in alveolar repair is Placenta expressed transcript-1 (Plet-1) (PMID: 38167746). Thus, we decided to investigate *Plet1* and related genes on a protein level (IHC for Plet-1 with digital quantitative analysis) and on the transcriptome level (*Plet1*, *Mertk*, *Siglecf*). Interestingly, we found out that there were significantly higher numbers of pulmonary Plet-1 immunolabelled cells in SARS-CoV-2-infected compared to mock-infected hamsters at 3, 6, and 14 dpi. Transcriptome data analysis revealed an increased expression of *Mertk* and *Plet1* at 6 dpi while at 14 and 28 dpi only *Plet1* was upregulated.

Combining all this evidence, we conclude that the morphologic and transcriptomic features of lung regeneration we observe in our study reflect a stereotypical response to alveolar damage, and not a distinct SARS-CoV-2 specific phenomenon. The fact that the lung response to SARS-CoV-2 infection in hamsters overlaps with that of other respiratory diseases highlights the usefulness of the model also beyond COVID-19 and expands the relevance of these findings to the wider field of lung regeneration. Compared to mice, the hamster model has not been used so extensively for viral respiratory diseases and the response to lung damage has not been characterized in detail. Altogether, we feel that since the hamster has the potential to be used more widely in this field, the new information we provide for the scientific community (longitudinal study design, characterization of cell types, rich transcriptome dataset, functional readout strategy) should be considered useful and significant.

We feel that the re-analysis of publicly available RNAseq data represents a suitable alternative approach to answer this comment raised by Reviewer 2 with respect to include another respiratory pathogen in the current study. Performing an additional long-term experiment with another virus would require new ethical committee permits, which is associated with very long processing times in Germany, which, combined with the time needed to perform a four-month experiment in BSL3 facilities, would delay the publication of this data for at least one year. As already pointed above, we believe that the information provided in this work deserves to be made available sooner. The results described above have been added to the revised manuscript and the discussion has been expanded to highlight that the findings could apply to other diseases as well. We hope that this approach, which is also in the spirit of the 3R-principle, meets the approval of Reviewer 2.

Based on this the **results, discussion, materials and methods, and supplementary material** have been revised accordingly in the manuscript as follows:

RESULTS

new line number 809-866

10. Transcriptome dynamics of ADI and airway basal cell signature as well as pro-fibrotic genes are comparable between SARS-CoV-2 and IAV infection in hamsters

One of the main questions when analyzing data from COVID-19 patients and animal models is whether an observed morphologic, functional or transcriptomic change is specific to the disease or a common feature of respiratory infections. The most comparable infectious disease is influenza A virus (IAV) infection. Hamsters are naturally susceptible to IAV and are considered a good model for the respiratory as well as systemic disease⁹³. In one published study, with infection of young hamsters with SARS-CoV-2 and IAV with inoculation doses adapted to reach equivalent viral loads, the disease severity, histopathological lesions, inflammatory infiltrates and transcriptomic changes in the lung were qualitatively and quantitatively comparable during acute (3, 7 dpi) and chronic disease (31 dpi). In particular, both viruses induced chronic bronchiolization lesions as observed in our study⁷⁹. Since this study did not look into epithelial cells participating in alveolar regeneration in detail, we decided to analyze selected gene signatures as determined in our own analysis in this published RNAseq dataset.

We selected the datasets obtained at 3 and 31 dpi as representative for acute and chronic infection, respectively, and compared the relative expression of gene sets in SARS-CoV-2 infected and IAV-infected lungs. Subsequently, we compared them to the relative expression in our hamsters at 3, 6 and 56 dpi, representing the timepoints with earliest, most prominent and latest significant changes, respectively. We chose to focus on genes expressed by early-stage ADI cells (*S100a6*, *Krt8*, *Anxa1*, *Tp53*, and *Hbgef*), late-stage ADI cells (*Sparc*, *Sox4*, and *Wwtr1*), club cells (*Gss* and *Pigr*), airway basal cells (*Ngfr*, *Pou2f3*, *Krt14*, and *Krt5*) and genes indicating a pro-fibrotic signature (*Mmp12*, *Mmp14*, *Tgfb1*, *Col5a1*, *Col1a1*, and *Col3a1*), which were differentially expressed in our own experiment.

In our dataset, early ADI cells genes were upregulated in the acute phase, whereas late ADI genes are variably upregulated in the chronic phase from 14 dpi onwards, as described above. In the study by Frere et al., SARS-CoV-2-infected animals showed a similar pattern of expression for ADI cells genes, whereas IAV-infected hamsters displayed a mostly homogeneous ADI cell gene expression already in the acute phase of the disease (Supplementary Fig. 10). This could indicate differences in the dynamics of ADI cell trajectories, suggesting a slight delay of ADI cell maturation in SARS-CoV-2 compared to IAV infection.

Club and airway basal cell progenitor genes were upregulated mainly in the chronic phase in our hamsters. In the study by Frere et al., SARS-CoV-2-infected animals showed only a slight upregulation of typical club and airway basal cell progenitor genes in the chronic phase of the disease, whereas IAV-infected hamsters displayed a more prominent upregulation of these genes at this phase (Supplementary Fig. 10). Of note, the authors reported that bronchiolization areas appeared more prominent in SARS-CoV-2 infected animals compared to IAV-infected ones in histology, albeit no significant difference was observed upon morphometry.

In our dataset, early pro-fibrotic environment genes involved in ECM-remodeling were upregulated in the acute phase, while collagen encoding genes were upregulated in the chronic phase of the disease, as stated above. The study by Frere et al showed a comparable pattern of expression in both SARS-CoV-2- and IAV-infected hamsters, but upregulation of both ECM-remodeling and collagen encoding genes was only noted in the acute phase of the disease (Supplementary Fig. 10). This discrepancy is in line with the lack of obvious collagen deposition reported by Frere et al., as opposed to the significant interstitial collagen deposition reported in this study.

In summary, comparison of our data with a published dataset from SARS-CoV-2 and IAV-infected hamsters showed mostly overlapping patterns in the expression of epithelial cell genes, with comparable changes in both viral diseases and both experiments. Differences between our data and the data from Frere et al. were observed in the pro-fibrotic gene expression in the chronic phase, with lasting changes in our hamster model as opposed to the transient upregulation in the SARS-CoV-2 and IAV-infected hamsters in Frere et al.

DISCUSSION

new line number 1168-1226

The current understanding of alveolar regeneration is that AT2 cells are mainly responsible for AT1 cell regeneration in homeostatic turnover and following mild injury, while airway progenitors are recruited after severe injury with marked AT1 cell loss^{75,121,122}. The participation of airway progenitors and ADI cells in alveolar regeneration has been demonstrated in different viral respiratory diseases (SARS-CoV-2, IAV, Sendai virus)^{18,73,74,79,123}, and non-infectious mouse models of lung injury (e.g. bleomycin, neonatal hypoxia and hyperoxia, LPS)^{74,75,124,125}. A contribution of airway progenitors to alveolar repair has been reported in COVID-19 patients and their presence has been associated with a profibrotic gene signature and presence of fibroblastic areas and bronchiolization. We have shown here and in our previous publication¹⁸, that the hamster recapitulates this pattern. The question remains, whether any of this response is SARS-CoV-2 specific or whether it could be observed in any other viral pneumonia. The most comparable infectious disease is IAV infection. Hamsters are naturally susceptible to IAV and are considered a good model for the respiratory as well as systemic disease⁹³. A number of studies have been published on the comparisons of SARS-CoV-2 and IAV infection in hamsters^{79,129-131}. Many of these publications demonstrate a higher disease severity and more pronounced inflammation in SARS-CoV-2 infection compared to IAV infection, when using similar infection doses. However, apart from these quantitative differences, qualitative morphological features and the antiviral response appear to be comparable between both viruses^{129,131,132}. In one published study, the infection doses of SARS-CoV-2 and IAV were adapted to reach equivalent viral loads, resulting in comparable disease severity, histopathological lesions, inflammatory infiltrates and transcriptomic changes in the lung during acute and chronic disease⁷⁹. Thus, it appears that hamsters might be more susceptible to SARS-CoV-2 than to IAV, but once the infections doses are adapted to cause the same level of alveolar damage, the response patterns are the same. Our analysis of the published transcriptome dataset obtained from SARS-CoV-2 and IAV infected young hamsters and comparison with our transcriptome data in old SARS-CoV-2 infected hamsters confirmed similarities of

the two diseases regarding cell types participating in alveolar regeneration. In addition, we demonstrated here that Plet-1, which has been recently identified as an important mediator of macrophage-epithelial cross-talk in alveolar regeneration in the IAV model⁷⁰, is also upregulated during the regeneration phase of SARS-CoV-2 infection. Combining all the evidence, we conclude that the morphologic and transcriptomic features of lung regeneration we observe in hamsters reflect a stereotypical response to severe alveolar damage, and not a distinct SARS-CoV-2 specific phenomenon. The fact that the response overlaps with that of other respiratory diseases highlights the usefulness of the model also beyond COVID-19 and expands the relevance of the findings beyond the COVID-19 research field.

Interestingly, mobilization of airway progenitors appears not to be a feature of SARS-CoV-2 infection in the mouse models. For instance, sub-lethal SARS-CoV-2 infection of K18-hACE2 mice failed to induce airway progenitor cell proliferation in contrast to IAV infection of B6 mice⁷³. The lack of airway progenitor cell proliferation has also been highlighted in BALB/c mice infected with the mouse-adapted SARS-CoV-2 strain MA10⁷¹. We believe that the apparent qualitative differences between viral models observed by some are not indicative of a true virus-specific response pattern, but most likely related to the severity of alveolar damage, which in turn is dependent on the virus dose and strain as well as the choice of animal species and age group. It appears that SARS-CoV-2-induced alveolar damage is more severe in hamsters whereas IAV is provoking a more severe damage in mice.

One discrepancy between the dataset from young and old hamsters was found regarding pro-fibrotic gene expression. In young SARS-CoV-2- and IAV-infected hamsters, pro-fibrotic genes were only upregulated in the acute disease, while we additionally noted upregulation of collagen-encoding genes in the chronic phase. This discrepancy is in line with the lack of obvious collagen deposition reported by Frere et al., as opposed to the significant interstitial collagen deposition reported in this study. This discrepancy is most likely related to the age difference, since the aged lung shows an impaired regeneration capacity and increased pro-fibrotic changes following alveolar damage¹³³.

MATERIAL AND METHODS

new line number 1575-1577

Data from Frere et al.⁷⁹ was downloaded from the GEO public database as raw counts and normalized with the rlog function (regularized log transformation) in DESeq2.

SUPPLEMENTARY MATERIAL

new Supplementary Figure 10

a Comparison SARS-CoV-2 / IAV

Supplementary figure 10: comparison SARS-CoV-2 and IAV-infection in hamsters.

Genes associated with a pro-fibrotic environment, ADI, club, or basal cells expressed in the lung of SARS-CoV-2 infected hamsters in this experiment were selected and the expression pattern was compared with a published dataset obtained in SARS-CoV-2- and IAV-infected hamsters (Frere et al., 2022). Heatmaps (pheatmap) show log₂ fold changes of normalized expression values (means per group) to the respective mock control means. Expression values are scaled by row. Red/orange indicates higher and blue/lightblue lower relative expression levels.

- 3. While the animals did not demonstrate observable behavioral differences in the post-acute phase, it is not clear that formal assessments of memory, attention, and behavior were performed, nor of more advanced neurologic phenotyping, as conducted in a rodent model of long COVID, which demonstrated objective differences in these parameters that would not otherwise be observable (PMID 37848036).**

We thank Reviewer 2 for their comment and apologize for the misleading statement. In the paragraph Reviewer 2 is referring to (Results section 2, first sentence), we stated that “after the resolution of acute SARS-CoV-2 infection, hamsters did not show any respiratory or behavioral signs at the daily clinical evaluation”. The behavioral signs we were referring to included potential changes detectable within the routine monitoring during the acute disease (reduced grooming and reduced activity) that were assessed in the standard daily clinical scoring. We did not perform any specific formal assessment of memory or attention. Since our aim was to focus on respiratory long-COVID, we considered these kind of measurements outside the scope of the aims of this study. We chose instead to investigate morphological, immunohistochemical and transcriptomic changes in the lung combined with a functional read-out using plethysmography combined with a treadmill. Our rationale behind this choice was to correlate lung function data, rather than behavioral changes with lung pathology analysis.

However, we did perform an additional behavioral assessment as a readout for exercise tolerance that was not mentioned in the original submitted version of the manuscript. In order to avoid bias in lung function measurements following treadmill exercise, we selected 6 out of 8 hamsters per group based on their performance on the treadmill in the training phase. The basis for this selection was a three-tiered scoring system that considered the animals’ attitude to run constantly and the frequency of breaks. Since no negative reinforcement was used in this experiment, the hamsters could stop running anytime and rest on the grid positioned at the lower end of the treadmill. The majority of animals was running constantly throughout the measurements. A similar scoring system was also applied during the experimental phase to ensure that animals were performing the same amount of exercise. Triggered by the comment of Reviewer 2, we realized that the scoring results could also be used as a readout of infection-induced changes in exercise tolerance. Interestingly, infected hamsters showed a slight decrease of the group mean score from 21 to 56 dpi compared to the mock-infected group. This result could point towards a higher reluctance to perform movements or prolonged activity and a putative correspondence to exercise intolerance reported in humans with respiratory long-COVID. We think that this additional information could be of value for future research with this model. In response to the comments of Reviewer #2, and in view of the potential usefulness of this information for the reader, we decided to add the results of the scoring in the manuscript.

Based on this the **results, discussion, materials and methods, and supplementary material** have been revised accordingly in the manuscript as follows:

RESULTS

new line number 209-214

The running behavior was scored with a scoring system that considered the animals’ disposition to run constantly and the frequency of breaks. Infected hamsters showed a slight decrease of the group mean score from 21 to 56 dpi compared to the mock-infected group. This result could point towards a higher reluctance to perform movements or prolonged activity which could be an indicator of exercise intolerance reported in humans with respiratory long-COVID (Supplementary Figure 1)

DISCUSSION

new line number 993-1000

As for the acute phase, we assume that behavioral differences also contributed to the changes in parameters measured by plethysmography after exercise. Infected hamsters showed a slight decrease of the running score in the treadmill from 21 to 56 dpi compared to the mock-infected group which could point towards a higher reluctance to running and a higher level of exhaustion, reminiscent of exercise intolerance reported in humans with respiratory long-COVID. The observed reduced respiratory rate in SARS-CoV-2 infected hamsters could indicate that the animals were spend more time resting.

new line number 1029-1032

In the chronic phase, SARS-CoV-2-infected hamsters were performing slightly worse on the treadmill compared to controls and this slightly reduced physical activity may have contributed to the differences in metabolism between the groups after exercise.

MATERIALS AND METHODS**new line number 1308-1319**

Since no negative reinforcement was used in this experiment, the hamsters could stop running anytime and rest on the grid positioned at the lower end of the treadmill. The behavior was scored with the following, semi-quantitative score: 1: Animal stays on the treadmill's grid most of the time, very reluctant to move, 2: animal needs some time to start running, but runs quite constantly towards the end; 3: animal runs constantly throughout the exercise. The animals with the best score during the training sessions were selected for exercise during the chronic infection. A similar scoring system was then used to assess and compare the animals' performance after infection (1: Animal frequently stops (more than 5 times) and it may manifest labored breathing pattern, 2: Animal occasionally makes brief stops (up to 5 times). When it stops it manifests normal breathing pattern, 3: Animal runs throughout the exercise).

SUPPLEMENTARY MATERIAL**Supplementary Figure 1**

Supplementary figure 1: Experimental design, exercise tolerance and virus quantification

a Schematic drawing of the experimental design. Male, 1-year old Syrian hamsters were infected with SARS-CoV-2 Delta variant and sacrificed at 1, 3, 6, 14, 28, 56 and 112 dpi. The study design allowed to distinguish three phases of the disease: acute phase (infection – 6 dpi), sub-acute phase (6 – 28 dpi), and chronic phase (28 dpi – 112 dpi). During the experiment, repeated lung function measurements were conducted using whole-body plethysmography (WBP) with respiratory gas analysis. Physical exercise on a rodent treadmill was used to exacerbate possible latent respiratory impairment from 21 dpi onwards and was repeated weekly. **b** Scoring of exercise tolerance (running behavior on the treadmill, mean and SEM). *N* = 18 (Training 1 until 28 dpi), 12 (35-56 dpi), or 6 (63-112 dpi) animals/group. **c-d** Immunohistochemistry for SARS-CoV-2 nucleoprotein (NP, c), and spike protein (SP, d). For each staining, an overview, a high magnification and the quantification in the whole section of the left lung lobe is shown. Quantitative data is shown as box and whisker plots. The bounds of the box plot indicate the 25th and 75th percentiles, the bar indicates medians, and the whiskers indicate minima and maxima. Dots show individual values. Data was tested by two-tailed Mann–Whitney *U* test. A *p* value of ≤ 0.05 was chosen as the cutoff for statistical significance. *N* = 7-8 animals/group/time-point. **e** Beeswarm plots for SARS-CoV-2 genes. The bounds of the box plot indicate the 25th and 75th percentiles, the bar indicates medians, and the whiskers indicate minima and maxima. Dots show individual values. *N* = 4 animals/group/time-point. **f** PCR for SARS-CoV-2 RNA-dependent RNA polymerase (RdRp) and subgenomic viral RNA in the chronic phase. Graphs show mean, SEM, and individual values. *N* = 7-8 animals/group/time-point. Negative results were set to -1 to enhance readability. Source data is provided as a Source Data file.

4. Penh is a flawed measure of pulmonary function in animal models, and it is largely recommended to avoid reporting it in studies (PMID 15317683).

We thank Reviewer 2 for this comment. We acknowledge that the utility of PenH as an indicator for respiratory resistance in rodents has been questioned. As correctly summarized in the reference provided by the Reviewer, PenH does not always correlate with changes in pulmonary resistance and might rather represent an unspecific parameter related to an altered breathing pattern. Our decision to include this parameter was based on the fact that it has been consistently reported as a sensitive readout of respiratory disease severity in mouse and hamster models of SARS-CoV-2 (PMID:35062015; PMID: 33031744; PMID: 35857635; PMID: 38416804), SARS-CoV (PMID: 26115403) and MERS-CoV infection (PMID: 27892925) as well as the LPS model of ARDS (PMID: 27892925). In these publications, the parameter was frequently shown to closely correspond, or even precede other classical measures of disease (e.g. clinical signs, peak of viral replication, histological lesions) and its application could differentiate between different viral doses and/or strains with differences in virulence (PMID: 26115403; PMID:35062015; PMID: 38416804). In the LPS ARDS model, PenH was shown the best indicator of the disease phenotype, compared to EF₅₀, Ti/Te ratio, frequency, tidal volume, and minute ventilation (PMID: 27892925). Particularly in the recent SARS-CoV-2 studies, the constructed variables PenH and Rpef are frequently the only reported parameters (PMID: 35062015; PMID: 33031744; PMID: 35857635; PMID: 35104835). For those reasons, we initially decided to include the parameter, since it was consistent with the results of others and allow a direct comparison to published data. However, we agree that we have incorrectly stated in the manuscript that PenH is an indicator of bronchoconstriction and did not provide a critical interpretation of the potential causes leading to its alteration after infection. Since, to our knowledge, the mechanisms driving PenH changes rodent models of viral disease have not been formally investigated yet, we decided to remove the measurement from the manuscript, rather than to provide a mere speculation.

Based on this and other comments from Reviewer 2, we also decided to provide more information and a more comprehensive, in-depth discussion of the plethysmography results. We therefore added the measurements of Tidal volume (TV), Te (Expiratory Time), and Ti (inspiratory Time) in the Results section and an additional paragraph in the discussion section (see also response to the following Point).

Based on this the **results, discussion, and materials and methods** have been revised accordingly in the manuscript as follows:

PenH graph has been removed From **Figure 1**

RESULTS

new line number 166-172

WBP revealed marked mechanical lung function changes in the acute phase of the disease. Here, we focused on specific mechanical metrics such as respiratory rate (Frequency), tidal volume (TV), EF₅₀ (mid expiratory flow), inspiration time (Ti), and expiration time (Te). Infected hamsters showed a significantly decreased frequency and EF₅₀ at 3 and 6 dpi compared to mock-infected animals. TV and Te were significantly higher compared to mock-infected animals at 3 dpi (Fig. 1 b). No significant difference were noted in Ti.

new line number 186-192

To summarize, the acute disease induced by SARS-CoV-2 Delta variant infection was characterized by weight loss, nasal discharge, and reduced activity. While no signs of lower respiratory tract distress

were detected by clinical monitoring, significant lung function changes were detected by WBP. The data indicate that SARS-CoV-2-infected hamsters were breathing slower and partially deeper, with a prolonged expiration and reduced expiratory airflow. This breathing pattern is typically observed in conditions with airflow limitation due to obstruction of airways^{35,36}.

4. **Can the authors please provide a rationale for why exercise precipitates a decline in respiratory rate in the infected animals? This is incongruous from what we observe in some PASC patients, where there is often ventilatory inefficiency (hyperventilation relative to the VCO₂) at rest and with exercise, though a recent meta-analysis is less conclusive on the matter**
5. **Similarly, can the authors please provide a rationale for why exercise precipitates a decline in EF50? That would suggest an increase in dynamic airways resistance, but the histologic lesions described in these animals are not airway-centered or suggestive of bronchiolitis.**

We thank Reviewer 2 for raising these insightful points, that stimulated a deeper dive into the interpretation of lung function data after exercise, which we believe improves the quality of the manuscript. We would like to answer both of the questions together since the changes in respiratory rate and EF50 are related. As stated in the answer to the previous comment (Point No. 4), we decided to provide additional parameters measured by plethysmography in order to obtain a more complete picture and allow a better interpretation. We expanded the results section by adding the additional parameters Tidal volume (TV), Te (Expiratory Time), and Ti (inspiratory Time), in addition to the already provided respiratory rate and EF50. Before we discuss the changes occurring after exercise in the chronic phase, we would like to comment on some observations during the acute phase. Our data show that at 3 and 6 dpi, SARS-CoV-2 infected hamsters breathe slower, but only partially deeper, since changes in TV are less pronounced than the ones in respiratory rate and they are only observed at 3 dpi. The breathing curve is characterized by a prolonged, slower expiration phase with increased Te and reduced EF50, while Ti remains unchanged (see new Figure 1, also pasted below). This breathing pattern is typically observed in conditions with airflow limitation due to obstruction of airways, e.g. rodent airway hyperresponsiveness models (PMID: 22973226; PMID: 16309547). During acute SARS-CoV-2 infection, this most likely corresponds to the damage to airway epithelium and obstruction of the airways by exudate, debris and inflammatory cells. Additionally, from 6 dpi onwards, we observe histologically a prominent hyperplasia and migration of airway progenitor cells in the terminal bronchioles, that potentially contributes to airflow restriction due to luminal narrowing, as described in detail in our previous publication (PMID: 37277327). Interestingly, there are some differences between our data and the changes reported in SARS-CoV-1 and SARS-CoV-2 infected mice (PMID: 26115403; PMID: 33031744; PMID: 35062015), which show an increase of EF50 after infection. The authors of the SARS-CoV-1 study reported that this goes along with a prolonged Te and lower Rpef, indicating a rapid exhalation of the majority of the volume, but a prolonged time needed to exhale the remainder volume. The authors concluded that this breathing pattern combines elements of both restrictive (rapid exhalation due to potentially reduced compliance) and obstructive (slow expiration with reduced flow rate in late expiration) lung disease patterns. Our hamster model does not recapitulate the element indicative of restrictive disease and is rather consistent with a purely obstructive phenotype. Of note, the prominent epithelial proliferation in terminal bronchioles and alveolar bronchiolization that occurs in SARS-CoV-2 infected humans and hamsters (PMID: 37277327; PMID: 36868468) are not observed in mice (PMID: 35857635). This could potentially explain differences in plethysmography changes between the two species.

Another possible contributor of the reduced respiratory rate in the acute phase is a behavioral difference with reduced activity of the hamsters in the plethysmograph. Recent publication evaluating the respiratory rate in unrestrained, mechanically restrained and chemically restrained hamsters argued that the reduced respiratory rate observed after SARS-CoV-2 infection is driven by behavioral changes such as reduced exploratory behavior, grooming or chewing, related to general malaise (PMID: 39066185; PMID: 38416804). As a matter of fact, during acute disease, the SARS-CoV-2 infected hamsters in our experiment tended to sit quietly in the plethysmograph, while some mock-infected animals showed intermittent phases of activity. However, in order to reduce this bias, we used a relatively long acclimatization times (at least 10 minutes), during which most animals calmed down. In addition, the software used for our experiment detects and rejects abnormal breathing patterns that occur during chewing or sniffing. On top of that, all recordings were subsequently reviewed manually and only segments showing a quiet, regular and physiologic breathing curve were used. However, since this cannot fully explain all the changes observed in the T_i/T_e ratio or EF50, we believe that a combination of lung function impairment and behavioral differences produce the observed phenotype.

In the chronic phase, no differences were observed in respiratory rate, TV, T_e , T_i or EF50 before exercise. After exercise, we observed changes with a pattern similar to the acute disease, with reduced respiratory rate, slight increase of TV, reduced EF50 and prolonged T_e in SARS-CoV-2- vs. mock-infected hamsters. Again, this pattern points towards an obstructive phenotype. We assume that this obstruction is caused by the space-occupying effect of persistent proliferation foci at the bronchioalveolar junction areas. These foci were present in 8/8 animals at 28 dpi and 7/8 animals at 56 dpi. Thereafter, the incidence and size of the foci decreased, which could explain the lack of differences in plethysmography after 56 dpi. In contrast, the fibrotic changes observed histologically in the chronic disease appear not to affect lung compliance, since the changes in lung function were not indicative of a restrictive disease phenotype.

Furthermore, as for the acute phase, we assume that behavioral differences also contributed to the changes in parameters measured by plethysmography after exercise. As also described in an answer above (see answer to point No. 3, Reviewer 2), we performed a behavioral assessment based on the animals' performance on the treadmill using a scoring system. Since hamsters are natural runners, the majority of animals was running constantly throughout the measurements, before and after infection. However, infected hamsters showed a slight decrease of the group mean score from 21 to 56 dpi compared to the mock-infected group (see new Supplementary Figure 1, also pasted above). After exercise, animals were immediately placed in the plethysmograph chambers and the measurements were started. Immediately after exercise, all mock-and SARS-CoV-2-infected animals were sitting calmly in the chambers, seemingly recovering from the effort as if "catching their breath". At some point, they would variably restart activities such as grooming and exploring. The observed reduced respiratory rate in SARS-CoV-2 infected hamsters could indicate that the animals spend more time resting. The scoring results and changes in breathing parameters after exercise could point towards a higher reluctance to running and a higher level of exhaustion during and after exercise, which could be an indicator of exercise intolerance reported in humans with respiratory long-COVID.

Unfortunately, assessment of breathing changes after exercise is not feasible with other methods. Mechanical restraint does not represent a useful option, since it does not prevent the animals from sniffing or chewing of the restrainer (PMID: 39066185). While it is known that invasive lung function techniques offer more reliable results, they represent terminal, one-point readouts and cannot provide data on awake animals or be combined with exercise.

In order to share these results and interpretation with the reader as well, we added the additional plethysmography readouts and the treadmill scoring results in the revised version of the manuscript and included an additional discussion paragraph.

Based on these new results the **results**, **discussion** as well as **materials and methods** have been revised accordingly in the manuscript as follows:

New graphs have been added in **Fig 1**

RESULTS

new line number 159-186

WBP revealed marked mechanical lung function changes in the acute phase of the disease. Here, we focused on specific mechanical metrics such as respiratory rate (Frequency), tidal volume (TV), EF50 (mid expiratory flow), inspiration time (Ti), and expiration time (Te). Infected hamsters showed a significantly decreased frequency and EF50 at 3 and 6 dpi compared to mock-infected animals. TV and Te were significantly higher compared to mock-infected animals at 3 dpi (Fig. 1 b). No significant difference were noted in Ti.

WBP measurement was coupled with a respiratory gas analyzer to assess metabolic function. vO_2 (O_2 uptake), vCO_2 (CO_2 production), MR (metabolic rate) and RQ (respiratory quotient) were measured. vO_2 and vCO_2 were significantly reduced at 3 and 6 dpi in SARS-CoV-2-infected hamster compared to mock-infected animals, whereas MR was significantly reduced only at 3 dpi. No significant changes were observed in RQ. Similar to mechanical WBP changes, these differences between SARS-CoV-2 and mock-infected animals were not observed from 10 dpi onwards (Fig. 1 b).

To summarize, the acute disease induced by SARS-CoV-2 Delta variant infection was characterized by weight loss, nasal discharge, and reduced activity. While no signs of lower respiratory tract distress were detected by clinical monitoring, significant lung function changes were detected by WBP. The data indicate that SARS-CoV-2-infected hamsters were breathing slower and partially deeper, with a prolonged expiration and reduced expiratory airflow. This breathing pattern is typically observed in conditions with airflow limitation due to obstruction of airways^{35,36}. Moreover, respiratory gas analysis revealed a decreased metabolic activity. These changes were marked, transient and disappeared around 10 dpi. However, based on studies in long-COVID-patients²⁶, we hypothesized that an underlying alteration in lung function and metabolism in the chronic phase of the disease could be exacerbated in a situation of physical stress. To substantiate our hypothesis, we evaluated lung function at later time-points before and after inducing physical stress, i.e. running on a rodent treadmill.

new line number 215-237

Interestingly, after mild exercise, SARS-CoV-2-infected animals showed a lower frequency compared to the mock-infected group until around 7 weeks after infection. A mild significant increase of TV, decrease of EF50, and an increase of Te was observed in SARS-CoV-2-infected animals compared to controls around similar time-points (Fig. 1c). No changes were observed Ti. (Fig. 1c). vO_2 , vCO_2 , and MR showed marked and significant differences from 21 dpi until 49 or 56 dpi. Differences in mechanical and metabolic values became less prominent around 56 dpi (Fig. 1c) and were no longer detectable until 112 dpi. Of note, the lack of differences in metabolic values was not caused by a recovery of SARS-CoV-2 infected animals, which showed lower values as compared to pre-infection until the end of the experiment, but rather caused by the gradual decrease of vO_2 , vCO_2 and MR in mock-infected animals

starting at 63 dpi (Fig. 1c). This decrease was closely associated with a gradual body weight loss starting at the same time in these animals and was considered related to aging. In summary, these results demonstrate that SARS-CoV-2-infected hamsters in the chronic phase i) do not recover their pre-infection body weight, ii) show lung function alterations after exercise persisting up to 7 weeks after infection iii) reduced running behavior in the treadmill, and iv) reduced metabolic activity values at rest, that become more pronounced after exercise. These findings are in line with changes in lung function and metabolic parameters observed in long-COVID-patients after physical exercise^{26,30}, which provides further evidence that the hamster is a suitable model for studying long-term pulmonary effects of SARS-CoV-2 infection. Our next aim was to characterize the underlying pathomorphological and transcriptomic changes associated with these findings.

DISCUSSION

new line number 926-1005

The early phase of the disease was characterized morphologically by pneumonia with high numbers of inflammatory cells, and subsequent features of alveolar regeneration with high numbers of ADI cells and airway cell proliferation. This was reflected by a markedly and transiently altered lung function at 3 and 6 dpi. It is interesting to note that, both during the acute phase of the disease and long-COVID, patients often report dyspnea or coughing⁹⁵. However, hamsters had very mild to moderate respiratory signs detectable by clinical scoring, limited only to the acute phase of the disease despite a marked alteration of lung function. This indicates that clinical scoring based on periodic visual inspection alone might be a sub-optimal method to assess respiratory impairment in hamsters and that more sensitive methods like WBP have to be applied to accurately assess the impact on the lower respiratory tract. The WBP alterations are in agreement with previous studies using the method to measure lung function in acute SARS-CoV-2 infection in hamsters^{32,71,94}. The changes were characterized by a lower frequency and partly increased TV, with a prolonged, slower expiration phase characterized by increased T_e and reduced EF50, but unchanged inspiration phase. This breathing pattern is typically observed in conditions with airflow limitation due to obstruction of airways, e.g. rodent airway hyperresponsiveness models^{35,36}. During acute SARS-CoV-2 infection, this most likely corresponds to the damage to airway epithelium and obstruction of the airways by exudate, debris and inflammatory cells. Additionally, from 6 dpi onwards, a prominent hyperplasia and migration of airway progenitor cells in the terminal bronchioles is observed, as we reported in detail previously¹⁸. This process putatively contributes to airflow restriction due to luminal narrowing.

Interestingly, there are some differences between our data and the changes reported in SARS-CoV-1 and SARS-CoV-2 infected mice^{32,94,96}, which show an increase of EF50 after infection. The authors of the SARS-CoV-1 study reported that this goes along with a prolonged T_e and lower R_pef, indicating a rapid exhalation of the majority of the volume, but a prolonged time needed to exhale the remainder volume. This breathing pattern combines elements of both restrictive (rapid exhalation due to potentially reduced compliance) and obstructive (slow expiration with reduced flow rate in late expiration) lung disease patterns. Our hamster model does not recapitulate the element indicative of restrictive disease and is rather consistent with an obstructive phenotype. Of note, the prominent epithelial proliferation in terminal bronchioles and alveolar bronchiolization that occurs in SARS-CoV-2 infected humans and hamsters^{18,97} are not observed in mice⁷¹. This could potentially explain differences in plethysmography changes between the two species.

Another possible contributor of the reduced respiratory rate in the acute phase is a behavioral difference with reduced activity of the hamsters in the plethysmograph. Recent publications evaluating in the hamster model argued that the reduced respiratory rate observed after SARS-CoV-2 infection is

driven by behavioral changes such as reduced exploratory behavior, grooming or chewing, related to general malaise^{98,99}. As a matter of fact, during acute disease, the SARS-CoV-2 infected hamsters in our experiment tended to sit quietly in the plethysmograph, while some mock-infected animals showed intermittent phases of activity. However, in order to reduce this bias, we used long acclimatization times, and abnormal breathing patterns indicative of sniffing or grooming were removed from the measurements.

In the chronic phase, no differences were observed in respiratory rate, TV, Te, Ti or EF50 before exercise. This suggests that the observed changes in the acute phase are directly related to the acute damage to alveoli and airways and the associated inflammatory process obliterating large areas of the lung, which are mostly resolved within 14 days. Although areas of alveolar proliferation are still prominent after this time-point, they do not obliterate the alveolar space, creating a more subtle alteration that does not seem to affect the measured lung function parameters in a resting state. Interestingly, after exercise, we observed marked lung function changes again. The use of a rodent treadmill was inspired by the use of similar tests in human medicine¹⁰⁰ and we think that its use should be taken into consideration in future animal studies modeling long-term respiratory signs induced by SARS-CoV-2. The changes showed the same pattern as observed in the acute disease, indicative of an obstructive phenotype. We assume that this obstruction is caused by the space-occupying effect of persistent proliferation foci at the bronchiolo-alveolar junction areas. These foci were present in almost all animals until 56 dpi. Thereafter, the incidence and size of the histological foci decreased, which could explain the lack of differences in plethysmography at later time points. In contrast, the pulmonary interstitial fibrosis arising from 14 dpi and persisting until 112 dpi does not appear to alter lung function in a resting state, since changes in lung function were not indicative of a restrictive disease phenotype with reduced lung compliance. Apparently, the relatively low degree of fibrosis observed does not impact breathing or can be compensated by remaining unaltered lung tissue. As for the acute phase, we assume that behavioral differences also contributed to the changes in parameters measured by plethysmography after exercise. Infected hamsters showed a slight decrease of the running score in the treadmill from 21 to 56 dpi compared to the mock-infected group which could point towards a higher reluctance to running and a higher level of exhaustion, reminiscent of exercise intolerance reported in humans with respiratory long-COVID. The observed reduced respiratory rate in SARS-CoV-2 infected hamsters could indicate that the animals were spend more time resting. Unfortunately, assessment of breathing changes after exercise is not feasible with other methods. Mechanical restraint does not represent a useful option since it does not prevent the animals from sniffing or chewing of the restrainer⁹⁸. While it is known that invasive lung function techniques offer more reliable results, they represent terminal, one-point readouts and cannot provide data on awake animals or be combined with exercise.

MATERIALS AND METHODS

new line number 1403-1405

The lung function parameters that were chosen to be analyzed were: frequency, TV, EF50, Ti, and Te.

6. **The observation of decreased O₂ uptake and CO₂ production suggests more of an impairment in the mitochondria (or muscle mass in the infected animals), though abnormal dead space ventilation could contribute. Understanding the metabolic derangements through metabolomics, mitochondrial and skeletal muscle density, and assessments of mitochondrial function *in vitro* would add rigor to this assessment. Further, the pulmonary vasculature is abnormal in patients with acute COVID and is associated with increased dead space ventilation (PMID 35671465, 35584345) as well as in patients with PASC, even without preceding critical illness (PMID 38500738,), and micro-CT of the hamster lungs to assess for micro- and macro-vascular changes would also be valuable.**

RESPONSE: We are thankful to Reviewer 2 for their insightful comments and suggestions. We agree that the changes in O₂ uptake and CO₂ production could be indicative of an altered metabolism, reduced muscle mass or related to vascular abnormalities, rather than being a readout of gas exchange impairment. We also acknowledge that the original manuscript did not offer an in-depth interpretation of these potential changes and decided to address this in the revised version. Since the focus of the project was to investigate respiratory function parameters and pathomorphological changes with a focus on epithelial regeneration, our original study design unfortunately did not include specific investigations like micro-CT or skeletal muscle density measurements. Moreover, we did not collect fresh tissue samples suitable for a comprehensive analysis of mitochondrial function, like heart, muscle, and liver. The fresh lung samples collected were stabilized in RNAlater and subsequently frozen, making them unsuitable for assessments of mitochondrial function *in vitro*. However, since we agree that the respiratory gas analysis results should be investigated further, we adapted our manuscript and substantially expanded it by adding new results (additional respiratory gas analysis parameters, transcriptome analysis of genes related to mitochondrial function and vascular remodeling, measurement of serum biomarkers for mitochondrial dysfunction and vascular remodeling, scoring results of treadmill performance). Moreover, we expanded the discussion to offer different potential explanations for changes in metabolic parameters.

First, we adapted the phrasing in the results section by clearly stating that the measurements of vO₂ and vCO₂ reflect a readout of metabolic activity. Additionally, we expanded the results by adding the metabolic rate (MR) and respiratory quotient (RQ) in order to add more information. Interestingly, the additional readouts revealed that in the acute phase, the metabolic rate is reduced in infected hamsters, while no alterations are detected in the RQ, indicating that the energy consumption is reduced, but that the balance of carbohydrate to lipid metabolism is unaltered by the infection. In the chronic phase, mild alterations of the metabolic rate were present even before the exercise, indicating that infection causes a lasting effect on resting metabolic rate. After exercise, this difference became more pronounced. vO₂, vCO₂, and MR showed marked and significant differences from 21 dpi until 49- or 56 dpi. From 63 dpi onwards, no differences were observed between SARS-CoV-2- and mock-infected animals. Of note, this lack of differences was not due to a recovery of SARS-CoV-2 infected animals, which showed lower values as compared to pre-infection until the end of the experiment, but rather caused by the gradual decrease of vO₂, vCO₂ and MR in mock-infected animals starting at 63 dpi. This decrease was closely associated with a gradual body weight loss starting at the same time in these animals and was considered related to aging.

Next, we formulated a putative list of factors potentially contributing to a reduction of vO₂, vCO₂ and MR following SARS-CoV-2 infection: i) reduced metabolic rate due to reduced physical activity, ii) reduced muscle mass, iii) altered mitochondrial function, iv) increased dead space ventilation due to vascular remodeling.

As discussed above (see answer to point 6), infected animals were calmer in the plethysmograph in the acute disease phase due to general malaise, and this could have contributed to the reduced metabolism. In the chronic phase, SARS-CoV-2-infected hamsters were performing slightly worse on the treadmill compared to controls between 21 and 56 dpi, as assessed by semi-quantitative behavioral scoring (see answers to point 3 and 6). This result could point towards a higher reluctance to perform movements or prolonged activity and a putative correspondence to exercise intolerance reported in humans with respiratory long-COVID. This slightly reduced physical activity certainly contributed to the differences in metabolism between the groups after exercise. However, $\dot{V}O_2$ and MR were already lower in SARS-CoV-2 infected hamsters before exercise, pointing towards a lower resting metabolic rate. Since hamsters did not show any signs of malaise in this disease stage, the differences could not be attributed to observable differences in physical activity. A reduced resting metabolic rate could be explained by reduced skeletal muscle mass resulting in lower oxygen consumption. Unfortunately, we can only offer a speculation for this point since skeletal muscle density measurement is not possible retrospectively. The infected animals lost weight after infection and did not recover their initial weight until the end of the study. We assume that a lasting loss of skeletal muscles contributed to the body weight loss and failure to regain the initial body weight, since the acute disease goes along with a reduced activity and movement, which would lead to muscle atrophy. Loss of fat-free mass, skeletal muscle mass and reduced resting metabolic is also observed in COVID patients with mild disease (PMID: 38057036). The results of the scoring were added to the revised manuscript and the line of argument was added into the discussion section.

As reviewer 2 pointed out, an alternative explanation for the reduced values is an impairment of mitochondria. Since there is evidence of mitochondrial dysfunction and metabolic reprogramming in SARS-CoV-2 infection of humans and mouse models (PMID: 38668888; PMID: 38614374; PMID: 35901960; PMID: 37556555; PMID: 39008677), we decided to investigate this possibility in our hamster model in more detail. First, we analyzed our lung transcriptome data using a published list (PMID: 37556555) of hamster hallmark genes involved in metabolic pathways and mitochondrial function. Transcriptome changes were only detectable in the acute, but not in the chronic phase of the disease. We found that acute SARS-CoV-2 infection is associated with a downregulation of some genes involved in the β -oxidation of fatty acids and upregulation of genes involved in ketone metabolism, indicating a metabolic switch. Other changes in acute disease were an upregulation of genes involved in TCA cycle, pyruvate metabolism, ROS scavenging systems and most genes belonging to Complex I to V of the mitochondrial OXPHOS system. The results are in line with what has already been reported in SARS-CoV-2-infected hamsters (PMID: 37556555). To investigate whether the transcriptomic changes would have a functional correlate on the systemic level, we also quantified levels of GSH and PRDX3 in serum samples of hamsters. Increased PRDX3 in the serum has been recently reported in long-COVID patients and is considered a potential biomarker of mitochondrial dysfunction (PMID: 38668888). However, serum analysis results did not reveal differences between Mock- and SARS-CoV-2-infected hamsters. Overall, the results indicate that mitochondrial dysfunction and metabolic alterations were only present during the acute disease in hamsters, correlating with the peak of inflammation and viral replication. In contrast, we did not find evidence of mitochondrial dysfunction or metabolic derangement in the chronic disease.

As pointed out by Reviewer 2, decreased $\dot{V}O_2$, $\dot{V}CO_2$ and MR in chronic disease could also be linked to increased dead space ventilation due to pulmonary vascular abnormalities, resulting in decreased oxygen uptake in the lung (PMID 35671465, 35584345, PMID 38500738). Pulmonary vascular remodeling in COVID-19 is characterized by an angiogenesis of the intussusceptive type which is induced by increased expression of Cxcl12 (stromal derived factor-1) and Cxcr4. Another hallmark of this peculiar SARS-CoV-2 induced vascular remodeling is the expression of Ccl12, Gdf15, Cd163, Col3a1 which underline the unique vascular etiology of COVID-19 distinguishing it from other forms of

interstitial lung disease (PMID: 33008942; PMID: 32437596; PMID: 38580869). Since we unfortunately could not perform micro-CT in this experiment to demonstrate vascular remodeling, we decided to analyze the expression of hallmark genes driving COVID-19 vascular remodeling from our RNAseq data. We found that *Ccl12*, *Gdf15*, and *Cd163* were upregulated during the acute phase, while *Col3a1* was markedly upregulated from 14 dpi to 56 dpi. Despite a marked downregulation during the acute phase, *Cxcl12* was upregulated at 28 and 56 dpi. Increased Neural precursor cell expressed developmentally down-regulated protein 9 (*Nedd9*) also known as enhancer of filamentation 1 (EF1) serum levels has been recently reported in long-COVID patients and is considered a potential biomarker of vascular remodeling (PMID: 33523764). However, we found no differences in the serum levels of EF between Mock- and SARS-CoV-2-infected hamsters at any time-point (Supplementary Fig. 9). Altogether, these results are suggestive of activation of pathways driving pulmonary vascular remodeling typical of COVID-19 during the acute phase.

In summary, we postulate that multiple factors could contribute to the reduction of vO_2 , vCO_2 and MR at rest and after exercise, including putative loss of muscle mass, reduced activity on the treadmill due to lower exercise tolerance. A possible increased dead space ventilation due to vascular remodeling cannot be confirmed or ruled out at this point, although we did observe markers of vascular remodeling mostly in the early phase. In contrast, we found no evidence of mitochondrial dysfunction in chronic disease.

Based on these new results the **results**, **discussion**, **materials and methods**, and **supplementary material** have been revised accordingly in the manuscript as follows:

RESULTS

new line number 166-186

WBP measurement was coupled with a respiratory gas analyzer to assess metabolic function. vO_2 (O_2 uptake), vCO_2 (CO_2 production), MR (metabolic rate) and RQ (respiratory quotient) were measured. vO_2 and vCO_2 were significantly reduced at 3 and 6 dpi in SARS-CoV-2-infected hamster compared to mock-infected animals, whereas MR was significantly reduced only at 3 dpi. No significant changes were observed in RQ. Similar to mechanical WBP changes, these differences between SARS-CoV-2 and mock-infected animals were not observed from 10 dpi onwards (Fig. 1 b).

To summarize, the acute disease induced by SARS-CoV-2 Delta variant infection was characterized by weight loss, nasal discharge, and reduced activity. While no signs of lower respiratory tract distress were detected by clinical monitoring, significant lung function changes were detected by WBP. The data indicate that SARS-CoV-2-infected hamsters were breathing slower and partially deeper, with a prolonged expiration and reduced expiratory airflow. This breathing pattern is typically observed in conditions with airflow limitation due to obstruction of airways^{35,36}. Moreover, respiratory gas analysis revealed a decreased metabolic activity. These changes were marked, transient and disappeared around 10 dpi. However, based on studies in long-COVID-patients²⁶, we hypothesized that an underlying alteration in lung function and metabolism in the chronic phase of the disease could be exacerbated in a situation of physical stress. To substantiate our hypothesis, we evaluated lung function at later time-points before and after inducing physical stress, i.e. running on a rodent treadmill.

new line number 190- 237

After the resolution of acute SARS-CoV-2 infection, hamsters did not show any respiratory signs at the daily clinical evaluation. However, SARS-CoV-2-infected hamsters did not recover their initial body

weight, which remained stable between 12 and 63 dpi. A gradual weight loss was observed in mock- and SARS-CoV-2-infected animals, starting from 63 dpi (Fig. 1 a). One animal died unexpectedly due to atrial thrombosis. The weight loss at the later time-points and atrial thrombosis were interpreted as age-related or spontaneous background lesions and considered unrelated to the infection²⁷⁻²⁹. In line with the lack of clinical signs related to infection, no differences in breathing pattern between the two groups were detected by WBP in a resting state. However, vO_2 and MR were slightly but significantly lower in SARS-CoV-2-infected hamsters between 21 and 56 dpi, indicating lasting effects on metabolism. No differences were observed in vCO_2 and RQ.

Long-COVID-patients often suffer from reduced pulmonary function, which can be exacerbated by physical exercise^{26,30}. To reproduce this condition experimentally, we used a combination of WBP and exercise on a rodent treadmill in the chronic phase of the disease. From 21 dpi onwards, the animals underwent mild physical exercise once a week, consisting of 10 minutes of running on a treadmill, with a slight upward slope and gradually increasing speed (10 m/min to 15 m/min). The intensity of training was well tolerated by the animals, with none of the animals displaying signs of exhaustion. The running behavior was scored with a scoring system that considered the animals' disposition to run constantly and the frequency of breaks. Infected hamsters showed a slight decrease of the group mean score from 21 to 56 dpi compared to the mock-infected group. This result could point towards a higher reluctance to perform movements or prolonged activity which could be an indicator of exercise intolerance reported in humans with respiratory long-COVID (Supplementary Figure 1).

Interestingly, after mild exercise, SARS-CoV-2-infected animals showed a lower frequency compared to the mock-infected group until around 7 weeks after infection. A mild significant increase of TV, decrease of EF50, and increase of Te was observed in SARS-CoV-2-infected animals compared to controls around similar time-points (Fig. 1c). No Ti changes were observed. (Fig. 1c). vO_2 , vCO_2 , and MR showed marked and significant differences from 21 dpi until 49 or 56 dpi. Differences in mechanical and metabolic values became less prominent around 56 dpi (Fig. 1c) and were no longer detectable until 112 dpi. Of note, the lack of differences in metabolic values was not caused by a recovery of SARS-CoV-2 infected animals, which showed lower values as compared to pre-infection until the end of the experiment, but rather caused by the gradual decrease of vO_2 , vCO_2 and MR in mock-infected animals starting at 63 dpi (Fig. 1c). This decrease was closely associated with a gradual body weight loss starting at the same time in these animals and was considered related to aging. In summary, these results demonstrate that SARS-CoV-2-infected hamsters in the chronic phase i) do not recover their pre-infection body weight, ii) show lung function alterations after exercise persisting up to 7 weeks after infection iii) reduced running behavior in the treadmill, and iv) reduced metabolic activity values at rest, that become more pronounced after exercise. These findings are in line with changes in lung function and metabolic parameters observed in long-COVID-patients after physical exercise^{26,30}, which provides further evidence that the hamster is a suitable model for studying long-term pulmonary effects of SARS-CoV-2 infection. Our next aim was to characterize the underlying pathomorphological and transcriptomic changes associated with these findings.

new line number 737-807

9. Transcriptome analysis of SARS-CoV-2 Delta infected aged hamsters does not provide evidence of mitochondrial dysfunction but shows activation of pathways involved in pulmonary vascular remodeling

Analysis of metabolic parameters by respiratory gas analysis revealed decreased vO_2 , vCO_2 and MR, which could be suggestive of a mitochondrial dysfunction in SARS-CoV-2-infected animals. Mitochondrial dysfunction and metabolic reprogramming have been demonstrated during SARS-CoV-

2 infection in humans and mouse models^{69,83-86}. For instance, it is reported that SARS-CoV-2 inhibits mitochondrial oxidative phosphorylation (OXPHOS) to increase reactive oxygen species (ROS) production in mice⁸⁵. To investigate metabolic reprogramming and mitochondrial dysfunction on a transcriptome level, we used a published list⁸⁴ of hallmark genes involved in metabolic pathways like β -oxidation of fatty acids, ketone metabolism, pyruvate metabolism, and tricarboxylic acid (TCA) cycle, ROS scavenging system, and all the complexes involved (complex I to V) in OXPHOS.

During the acute phase, SARS-CoV-2 infected hamsters showed a downregulation of some genes involved in the β -oxidation of fatty acids (*Acad11*, *Acss1*, *Acsl1*, *Acam3*, and *Decr1*), whereas some genes involved in ketone metabolism were upregulated (*Gpd2*, *Bdh1*, *Rpia*). This metabolic switch has been reported in humans and in a murine model for COVID-19. Expression of genes involved in the TCA cycle (*Fh1*, *Idh2*, *Idh3g*) was upregulated in the acute disease phase. Some genes involved in pyruvate metabolism also showed upregulation (*Rpla*, *Pdhhb*, *Bdh1*) at 3 and 6 dpi. However, none of these changes were present in the chronic phase of the disease (Supplementary Fig. 7; Supplementary Data 7).

SARS-CoV-2 infected hamsters showed a marked upregulation of genes involved in the ROS scavenging systems especially at 3 and 6 dpi. Of note, genes like *Sod2*, *Gpx1*, *Gsr*, *Prdx1*, *Prdx3* were markedly upregulated. Importantly, some of these genes are playing a crucial role in the mitochondrial ROS scavenging. However, none of these genes were differentially upregulated at the later time-points.

Similarly, SARS-CoV-2 infected hamsters showed an upregulation of most genes belonging to the Complex I to Complex V of the mitochondrial OXPHOS only during the acute phase of disease (Supplementary Fig. 8; Supplementary Data 7). These results of upregulated ROS scavenging system and upregulated OXPHOS complexes-related genes are similar to what has been reported in a recent publication that investigated this same gene set in SARS-CoV-2 infected hamsters during the acute phase of the disease⁸⁴.

To investigate whether the transcriptomic changes would have a functional correlate on the systemic level, we also quantified levels of Glutathione peroxidase (GSH/GSSG) and peroxiredoxin-3 (PRDX3) in serum samples of hamsters. GSH/GSSG was performed to verify the systemic oxidative stress level, while increased PRDX3 in the serum has been recently reported in long-COVID patients and is considered a potential biomarker of mitochondrial dysfunction⁸⁶. No differences were found for either marked between Mock- and SARS-CoV-2-infected hamsters (Supplementary Fig. 8).

In summary, transcriptomic data obtained from lung tissues indicate that mitochondrial dysfunction and metabolic alterations were only present briefly during the acute disease in hamsters, correlating with the peak of inflammation and viral replication. In contrast, we did not find evidence of mitochondrial dysfunction or metabolic derangement in the chronic disease.

The decreased vO_2 , vCO_2 and MR in chronic disease could also be linked to increased dead space ventilation due to pulmonary vascular abnormalities, resulting in decreased oxygen uptake in the lung⁸⁷⁻⁸⁹. Pulmonary vascular remodeling in COVID-19 is characterized by an angiogenesis of the intussusceptive type which is induced by increased expression of *Cxcl12* (stromal derived factor-1) and *Cxcr4*. Another hallmark of this peculiar SARS-CoV-2 induced vascular remodeling is the expression of *Ccl12*, *Gdf15*, *Cd163*, *Col3a1* which underline the unique vascular etiology of COVID-19 distinguishing it from other forms of interstitial lung disease⁹⁰⁻⁹². For these reasons we sought to investigate the hallmark genes of COVID-19 vascular remodeling in the hamster model. Transcriptome data analysis revealed that *Ccl12*, *Gdf15*, *Cd163* were markedly upregulated during the acute phase at 3 and 6 dpi. *Col3a1* was markedly upregulated from 14 dpi to 56 dpi. Despite a marked downregulation during the acute phase at 3 and 6 dpi, *Cxcl12* was upregulated at 28 and 56 dpi (Supplementary Fig. 9; Supplementary Data 7).

Increased Neural precursor cell expressed developmentally down-regulated protein 9 (Nedd9) also known as enhancer of filamentation 1 (EF1) serum levels has been recently reported in long-COVID patients and is considered a potential biomarker of vascular remodeling^{89,90}. However, the serum level of EF 1 was not changed in the hamster species regardless of the infectious status (Supplementary Fig. 9). Altogether, these results are suggestive of activation of pathways driving pulmonary vascular remodeling typical of COVID-19 during the acute phase.

DISCUSSION

new line number 1022-1064

Besides the alterations of breathing parameters, SARS-CoV-2-infected hamsters also showed decreased vO_2 , vCO_2 and MR in the acute phase. We hypothesize that these alterations could have been caused by one or more of the following: i) reduced metabolic rate due to reduced physical activity, ii) reduced muscle mass, iii) impaired mitochondrial function, iv) increased dead space ventilation, e.g. due to vascular remodeling. As discussed above, SARS-CoV-2-infected animals were calmer in the plethysmograph in the acute disease phase due to general malaise and this could have contributed to reduced metabolic rate. In the chronic phase, SARS-CoV-2-infected hamsters were performing slightly worse on the treadmill compared to controls and this slightly reduced physical activity may have contributed to the differences in metabolism between the groups after exercise. However, vO_2 and MR were already lower in SARS-CoV-2 infected hamsters before exercise, pointing towards a lower resting metabolic rate. The reduced resting metabolic rate could be explained by reduced skeletal muscle mass resulting in lower oxygen consumption. Infected animals lost weight after infection and did not recover their initial weight until the end of the study. We assume that this phenomenon is at least partly due to lasting loss of skeletal muscle mass, since the acute disease goes along with a reduced activity and movement, which would lead to muscle atrophy. Loss of fat-free mass, skeletal muscle mass and reduced resting metabolic rate is observed in COVID-19 patients, even those with mild disease not requiring hospitalization⁴².

Since there is evidence of mitochondrial dysfunction and metabolic reprogramming during SARS-CoV-2 infection in humans and mouse models^{69,83-86}, we also analyzed expression of hallmark genes involved in metabolic pathways and mitochondrial function and measured serum biomarkers of mitochondrial dysfunction. Transcriptomic data indicate that mitochondrial dysfunction and metabolic alterations were only present briefly during the acute disease in hamsters, correlating with the peak of inflammation and viral replication, which is in line with the published evidence in COVID-19⁸⁴. In contrast, we did not find evidence of mitochondrial dysfunction or metabolic derangement in the chronic disease. However, it should be mentioned that for a complete evaluation of mitochondrial dysfunction causing a metabolic change observed in this study, organs like skeletal muscles, heart and liver should be investigated in depth. Since that this metabolic alteration was not expected, these organs were not collected to be destined to a transcriptome analysis for this current study.

Lastly, increased dead space ventilation due to pulmonary vascular abnormalities, resulting in decreased oxygen uptake in the lung, could have contributed to decreased vO_2 , vCO_2 and MR in this experiment⁸⁷⁻⁸⁹. In our study, some markers of the intussusceptive vascular remodeling typical for COVID-19 were upregulated during the acute (*Ccl12*, *Gdf15*, and *Cd163*) or chronic phase (*Col3a1*, *Cxcl12*). These results could be suggestive of the activation of pathways driving pulmonary vascular remodeling. However, further studies including micro-CT analysis are needed to assess for micro- and macro-vascular changes in the lung of SARS-CoV-2 infected hamsters and to substantiate the molecular findings and interpretations.

MATERIALS AND METHODS

new line number 1406-1407

Parameters evaluated by respiratory gas analysis were vO_2 , vCO_2 , MR and RQ.

new line number 1600

13. Serum analysis

Analysis of oxidative stress was carried out in serum samples. Blood was collected at necropsy from the abdominal aorta into Eppendorf tubes and left to clot for at RT. Following centrifugation, serum was collected in cryo tubes and stored at $-80^{\circ}C$.

Enhancer of filamentation 1 (EF1), Peroxiredoxin 3 (Prdx3)

To determine the amount of EF1 and Prdx3 in serum samples, a quantitative sandwich ELISA was used (MyBioSource MBS9364488 and MBS9391144). The kits are based on EF1 antibody-EF1 antigen interactions or PRDX3 antibody-PRDX3 antigen interactions (immunosorbency) and an HRP colorimetric detection system to detect EF1 antigen targets in samples. The assay procedure was performed according to the manufacturers' instructions. After 5 minutes of adding the stop solution, absorbance was measured once at 450 nm with the Tecan Spark Multimode Microplate Reader. Sample ODs were plotted against the standard curve to determine EF1 and Prdx3 contents. The kits have a detection range of 0.25 ng/ml – 8 ng/ml (EF1) and 0.625 ng/ml – 20 ng/ml (Prdx3).

Glutathione (GSH/GSSG)

To determine the glutathione content of serum samples a quantitative colorimetric assay kit (MedChemExpress HY-K0311) with a detection range from 1.57 μM – 50 μM was used. The kit uses the enzymatic method that utilizes Ellman's Reagent (DTNB) and glutathione reductase (GR). DTNB is able to react with the reduced glutathione and forms a yellow product, which can be measured at 412 nm. The change in the OD is direct proportional to the glutathione concentration. After addition of the Enzyme/Coenzyme working solution the absorbance was measured 20 times with intervals of 25 seconds between the readings in a kinetic fashion with the Tecan Spark Multimode Microplate Reader. The standard curve obtained from linear fitting of standard samples was used to determine total glutathione content in samples.

Statistical analysis and graphs design were performed using GraphPad Prism 9.3.1 (GraphPad Software, San Diego, CA, USA) for Windows™. The assumption of normal distribution was tested using the Kolmogorov-Smirnov test. Due to the rejection of the assumption of normal distribution, non-parametric methods were used. Pairwise comparison between SARS-CoV-2-infected hamsters and control group at each time-point were tested with a two-tailed Mann-Whitney-U test. Statistical significance was accepted at exact p-values of ≤ 0.05 .

SUPPLEMENTARY MATERIAL

Excel file with all the new analyzed gene sets (supplementary data 7).

new Supplementary Figures 7, 8, and 9

Supplementary figure 7: Metabolism.

Heatmaps of normalized expression values and beeswarm plots of selected genes involved in different metabolic pathways at each dpi in mock- and SARS-CoV-2- infected hamsters. In the heatmaps, expression values are scaled by row. Red indicates higher and blue lower relative expression levels. In the beeswarm plots, the bounds of the box plot indicate the 25th and 75th percentiles, the bar indicates medians, and the whiskers indicate minima and maxima. Dots indicate individual values. N = 4 animals/group/time-point.

Supplementary figure 8: Mitochondrial Oxidative phosphorylation.

a-f Heatmaps of normalized expression values and beeswarm plots of selected genes involved in ROS scavenging and oxidative phosphorylation at each dpi in mock- and SARS-CoV-2-infected hamsters. In the heatmaps, expression values are scaled by row. Red indicates higher and blue lower relative expression levels. In the beeswarm plots, the bounds of the box plot indicate the 25th and 75th percentiles, the bar indicates medians, and the whiskers indicate minima and maxima. Dots indicate individual values. N = 4 animals/group/time-point. g, h Quantification of total Glutathione (GSH/GSSG, g) and Peroxiredoxin 3 (Prdx3, h) in serum samples of mock- and SARS-CoV-2-infected hamsters. Data is shown as box and whisker plots. The bounds of the box plot indicate the 25th and 75th percentiles, the bar indicates the median, and the whiskers indicate minima and maxima. Dots indicate individual values. In c, data was tested by two-tailed Mann–Whitney U test. A p value of ≤ 0.05 was chosen as the cutoff for statistical significance. N = 7-8 animals/group/time-point.

Supplementary figure 9: Vascular remodeling.

a Heatmaps of normalized expression values and beeswarm plots of genes involved in COVID-19-associated pulmonary vascular remodeling at each dpi in mock- and SARS-CoV-2-infected hamsters. In the heatmaps, expression values are scaled by row. Red indicates higher and blue lower relative expression levels. In the beeswarm plots, the bounds of the box plot indicate the 25th and 75th percentiles, the bar indicates medians, and the whiskers indicate minima and maxima. Dots indicate individual values. N = 4 animals/group/time-point. b Quantification of EF1, a marker for COVID-19-associated vascular dysfunction, in the serum of mock- and SARS-CoV-2-infected hamsters. Data is shown as box and whisker plots. The bounds of the box plot indicate the 25th and 75th percentiles, the bar indicates medians, and the whiskers indicate minima and maxima. Dots indicate individual values. Data was tested by two-tailed Mann–Whitney U test. A p value of ≤ 0.05 was chosen as the cutoff for statistical significance. N = 7-8 animals/group/time-point.

7. **Prior studies have shown dysregulated CD8⁺ T cells in the lung and GI tract are highly activated compared to circulating T cells and associate with organ-specific dysfunction, suggesting that organ-specific PASC may be due to dysregulated local immune cells (PMID 34591653). It may be worth considering FACS on the bronchoalveolar lavage fluid compared to circulating blood to assess the burden of locally dysregulated CD8⁺ T cells.**

RESPONSE: The authors are thankful to Reviewer 2 for their comments. We are aware of the association of dysregulated T cells with possible organ specific symptoms of COVID-19, and we agree with Reviewer 2 that it is an intriguing topic. However, the focus of the current experiment was the epithelial cell reaction to injury. Our main interest was to evaluate whether a complete regeneration of alveolar structures would occur over time and whether this would lead to a complete restoration of lung function in the hamster model. Since this goal already encompassed a wide and comprehensive analysis, an in-depth analysis of interactions among immune cells involved in these processes was not included in the original study design. For this reason, whole peripheral blood cells, PMBC or BAL were not collected and FACS analysis of these samples cannot be performed at this point.

However, since we value the comment of Reviewer 2, and we think that evaluating local immunologic dysregulation as a potential cause for persistent lung lesions deserves further investigations, we looked into the presence of dysregulated T cells based on our transcriptome data. From the literature, we selected a list of genes that are reported to be mainly expressed by dysregulated T cells that have been described in humans with respiratory long COVID-19 (PMID: 34591653). A marked up-regulation of these genes was noted only at the acute phase of infection (3 and 6 dpi), with no persistence during the chronic phase. This is in contrast to the results reported by Cheon and colleagues in respiratory long COVID-19 patients. They showed increased numbers of γ/δ T cells, B cells, and CD8⁺ T cells in the BAL fluid 60-90 days after infection. Thus, the hamster does not seem to recapitulate this phenotype of respiratory long-COVID-19. However, it is known that pulmonary long-COVID comprises different phenotypes with distinct histopathological, clinical and radiological features and not all patients show persistent inflammatory changes (PMID: 35301248).

The authors recognize that the analysis based on immunohistochemistry and transcriptome analysis has limitations. The gene lists that were provided in the paper were obtained by scRNAseq (PMID 34591653). Some of the genes are unspecific if taken singularly and the phenotype of distinct T cell types is determined by the combination of expressed genes and not a single specific marker. In our case, with bulk RNAseq, a detailed characterization of specific cell types was of course not possible. Therefore, we cannot exclude with certainty that the cell types described by other authors are present in low numbers in the chronic disease in the hamster but are indistinguishable or the gene expression is not strong enough to detect them. We also cannot exclude that dysregulated T cells have a different phenotype in hamsters. In this scenario, to approach this problem we would need to use BAL and peripheral blood cells in future studies as suggested by Reviewer 2 initially in their comment.

Given the limitations of this suggested approach, we leave it up to Reviewer 2 whether they think that this data should be included in the final version of this manuscript or not. The data has been provisionally included in the revised form to allow Reviewer 2 to visualize it.

On the other hand, following the suggestion to investigate immune cell interactions guiding chronic disease, we added an additional evaluation of immune cells implicated in pulmonary regenerative processes. It has been recently demonstrated in a murine model of influenza A virus (IAV) pneumonia that an important mediator of macrophage-epithelial cross-talk for alveolar proliferation is Placenta expressed transcript-1 (Plet-1) (PMID: 38167746). Thus, we decided to investigate *Plet1* and related genes on a protein level (IHC for Plet-1 with digital quantitative analysis) and on the transcriptome level (*Plet1*, *Mertk*, *Siglecf*). Interestingly, we found out that there were significantly higher numbers

of pulmonary Plet-1 immunolabelled cells in SARS-CoV-2-infected compared to mock-infected hamsters at 3, 6, and 14 dpi. Transcriptome data analysis revealed an increased expression of *Mertk* and *Plet1* at 6 dpi while at 14 and 28 dpi only *Plet1* was upregulated.

Based on these new results the **results, discussion, materials and methods, and supplementary material** have been revised accordingly in the manuscript as follows:

RESULTS

new line number 425-442

Different subtypes of dysregulated lung T cells have been reported to be associated with post-acute COVID-19 lung sequelae^{68,69}. For this reason, we wanted to further investigate T cell dysregulation in lungs of SARS-CoV-2 infected hamsters on the transcriptome level, using published gene signatures⁶⁸. Results showed an upregulation of genes characterizing dysregulated T cells (e.g. *Gzma*, *Tnf*, *Eomes*, *Nkg7*, *Gzmk*) only during the acute phase of disease, at 3 and 6 dpi (Supplementary Fig. 3). Thus, we could not confirm the presence of a persistent pool of activated T cells driving pathological changes in the chronic phase.

While inflammatory infiltrates were not prominent in the chronic lesions of SARS-CoV-2-infected hamsters, regenerative changes were present throughout the investigation period. It has been demonstrated in a murine model of influenza A virus (IAV) pneumonia that an important mediator of macrophage-epithelial cross-talk for alveolar proliferation is Placenta expressed transcript-1 (Plet-1)⁷⁰. Quantification of Plet-1 immunolabelling in the lungs of hamsters revealed significantly higher numbers of Plet-1⁺ cell in SARS-CoV-2-infected compared to mock-infected hamsters at 3, 6, and 14 dpi. Interestingly, transcriptome data analysis revealed an increased expression of *Mertk* and *Plet1* at 6 dpi while at 14 and 28 dpi only *Plet1* was upregulated (Supplementary Fig. 3).

DISCUSSION

new line number 1194-1204

Our analysis of the published transcriptome dataset obtained from SARS-CoV-2 and IAV infected young hamsters and comparison with our transcriptome data in old SARS-CoV-2 infected hamsters confirmed similarities of the two diseases regarding cell types participating in alveolar regeneration. In addition, we demonstrated here that Plet-1, which has been recently identified as an important mediator of macrophage-epithelial cross-talk in alveolar regeneration in the IAV model⁷⁰, is also upregulated during the regeneration phase of SARS-CoV-2 infection. Combining all the evidence, we conclude that the morphologic and transcriptomic features of lung regeneration we observe in hamsters reflect a stereotypical response to severe alveolar damage, and not a distinct SARS-CoV-2 specific phenomenon. The fact that the response overlaps with that of other respiratory diseases highlights the usefulness of the model also beyond COVID-19 and expands the relevance of the findings beyond the COVID-19 research field.

MATERIALS AND METHODS

Immunohistochemistry was performed on consecutive lung sections from all animals to detect SARS-CoV-2 antigen (SARS-CoV-2 nucleoprotein and spike protein), macrophages and dendritic cells (ionized calcium-binding adapter molecule 1, Iba-1), T-cells (CD3), B-cells (pax5), neutrophils (myeloperoxidase, MPO), alveolar differentiation intermediate cells (cytokeratin 8), airway basal cells (cytokeratin 14, cytokeratin 5 and Δ NP63), club cells (secretoglobin 1A1), bronchial cells (MUC5AC, MUC5B), senescent cells (p21, p53), proliferating cells (Ki-67), M2-like macrophages (CD 204), and Placenta-expressed transcript 1 (**Plet1**).

SUPPLEMENTARY DATA

Excel file with all the new analyzed gene sets (supplementary data 7)

SUPPLEMENTARY MATERIAL

new data in Supplementary Figure 2

Supplementary figure 2: Inflammatory cells in the lung.

a-d Immunohistochemistry for Iba-1 (macrophages, a), CD3 (T cells, b), myeloperoxidase (MPO, heterophils, c), and Pax5 (B cells, d). For each staining, an overview, a high magnification and the quantification in the whole section of the left lung lobe is shown. **e-j** Beeswarm plots for T-cell genes (e-h), and macrophage gene (i). In f there is a cybersortX deconvolution for M1-like macrophage signature genes. Quantitative data is shown as box and whisker plots. The bounds of the box plot

indicate the 25th and 75th percentiles, the bar indicates medians, and the whiskers indicate minima and maxima. Dots indicate individual values. In a-d, data was tested by two-tailed Mann–Whitney *U* test. A *p* value of ≤ 0.05 was chosen as the cutoff for statistical significance. *N* = 7-8 animals/group/time-point (a-d) or 4 animals/group/time-point (e-j). Source data is provided as a Source Data file.

new Supplementary Figure 3

Supplementary figure 3: T cells and macrophage-epithelial cross-talk.

a, b Heatmaps of normalized expression values for genes characterizing T cell dysregulation in COVID-19 at each dpi in mock- and SARS-CoV-2- infected hamsters. **c** Immunohistochemistry for Placenta expressed transcript-1 (Plet-1), a mediator of macrophage-epithelial crosstalk in alveolar proliferation. Quantification in the whole section of the left lung lobe, a low and a high magnification is shown. Positive alveolar macrophages are indicated by arrows. **d** Heatmap and beeswarms for *Plet1* and related genes. In a, b, and d, expression values are scaled by row. Red indicates higher and blue lower relative expression levels. In c, and d, data is shown as box and whisker plots. The bounds of the box plot indicate the 25th and 75th percentiles, the bar indicates medians, and the whiskers indicate minima and maxima. Dots indicate individual values. In c, data was tested by two-tailed Mann–Whitney *U* test. A *p* value of ≤ 0.05 was chosen as the cutoff for statistical significance. *N* = 7-8 animals/group/time-point (c) or 4 animals/group/time-point (a, b, d). Source data is provided as a Source Data file.

Reviewer #3 (Remarks to the Author):

NCOMMS-24-37593

1. **The authors utilize a hamster model of COVID-19 to demonstrate chronic or long term effects of infection on lung pathology and on the activity of the infected animals. Besides the observation of pulmonary fibrosis, the authors demonstrate a shift from alveolar epithelial cell progenitors to club cell progenitors, further demonstrating unique, long term changes in lung pathology that may contribute to prolonged pulmonary dysfunction. A particular strength of the study is the inclusion of exercise-induced physiological measurements.**

RESPONSE: The authors are thankful to Reviewer 3 for their time and the effort taken to revise our manuscript. We are particularly thankful for the insightful comments that helped us to improve the scientific value of our manuscript. We provide below a detailed point by point discussion of the comments made by Reviewer 3. Thanks to Reviewers comments, we believe that the revised version of this manuscript is of improved value for the scientific community, and we hope that will meet Reviewer 3 approval.

2. **A particular weakness of the study is its use of only male animals, as that makes the study conclusions only relevant to males. A final shortcoming is the lack of a control group infected with another respiratory virus, which would have strengthened the conclusions that the observed changes in pathology were specific to COVID-19 and not a result of infection with any respiratory virus. There are very interesting observations that come from the clear and detailed analysis of the data, but the shortcomings do impact significance. While I understand that using males only skews the population to ones that have more severe disease, it does limit the impact of the study because any conclusions drawn can only be applied to males. I realize this is a critique that involves significant expense, but the lack of a control group infected with another virus prevents the authors from stating that the pulmonary effects and damage they see are specific to COVID and not a result of infection with any respiratory virus.**

RESPONSE: The authors are thankful to Reviewer 3 for their comments. Our answer to this Reviewer 3 comment will be divided in two topics for sake of clarity. A “justification for only male” and a “justification for only SARS-CoV-2” sections will follow.

“justification for only male”

We thank Reviewer 3 for their comments. We are aware that in an optimal condition the use of both sexes would have been preferable and that the results of this study cannot be extrapolated to females. However, we believe that our rationale to include only male old hamsters was appropriate for the aims of this project. Our initial concept was to offer a model for a population at high risk of respiratory long-COVID, namely people that experienced severe acute COVID-19, which are older male patients. Thus, we chose to use animals that would have a higher probability of severe acute disease and slower

recovery. As Reviewer 3 rightfully pointed out, this skews the studies population to a certain subgroup. Old age and male sex are considered to be risk factors for severe acute-COVID19 (PMID: 34454673; PMID: 35891696; PMID: 38652535 PMID: 32411652, PMID: 33298944) and severe acute COVID-19 is a risk factor for respiratory long-COVID (PMID: 35891696; PMID 35429399; PMID: 34454673; PMID: 38652535). Additionally, some studies that are focusing specifically on respiratory long-term sequelae, have found that male sex and age are risk factors for developing pulmonary PASC (PMID: 38652535). When looking at the disease severity and pulmonary lesions in the Syrian golden hamster model for SARS-CoV-2, the literature reports male hamsters as being more severely affected in the acute stage (PMID: 34253053; PMID: 33790236; PMID: 36851642; PMID: 34870132). The few long-term studies extending beyond the first 2 weeks also report males as recovering more slowly (PMID: 37572667; PMID: 34253053; PMID: 34870132). One of these studies (PMID: 37572667), where some of us are co-authors, demonstrated that males (humans and hamsters) have a worse outcome of the disease than females up to 21 dpi, comparing lung function and histopathology of lung lesion. Therefore, since our focus was respiratory long-COVID and not long-COVID in *sensu lato*, and since the probability of more severe pulmonary disease with functional impairment was higher in male hamsters, we consider the choice of aged males sound and substantiated by the aforementioned references.

We have prepared a revised version of the manuscript, outlining the reasoning for the choice of males and the resulting limitations more clearly, and including a modified title underlining that our main focus is respiratory long-COVID and not long-COVID in *sensu lato*. We believe that the manuscript provides novel and highly valuable information that deserves to be made available to the scientific community in a reasonable time, given the relevance of the long-COVID situation. We feel that the significance of the manuscript is not compromised by the lack of females, since it offers a comprehensive and extensive investigation relevant for a very large group of patients and has several strengths. We sincerely hope that Reviewer 3 will accept our rationale and consider the revised manuscript for publication.

Based on this, the **title**, **introduction** and **discussion** have been revised accordingly in the manuscript as follows:

TITLE

Persistent alveolar bronchiolization, interstitial fibrosis and impaired lung function post-exercise are features of respiratory long-COVID in SARS-CoV-2-Delta variant-infected aged hamsters

INTRODUCTION

new line number 98-108

Only male, aged (~ 1-year-old) hamsters were chosen for the study. The rationale behind this choice was that we wanted to characterize in detail a respiratory long COVID-19 model for a better understanding of the pathogenetic processes associated with this syndrome. The choice of our study design was based on the following observations i) old age and male sex are risk factors for severe acute COVID-19^{8,19-23}, ii) severe acute disease is a risk factor for respiratory long-COVID^{8,21,23,24}, and iii) male hamsters show more severe disease course with more prominent lung function alterations and slower recovery, as well as more severe histological pulmonary lesions compared to females^{22,25-30}. SARS-CoV-2 Delta-variant was chosen since it showed the most prominent pulmonary pathological changes among the most common variants worldwide at the time^{31,32}.

DISCUSSION

new line number 1234-1248

Second, we used only old male animals. We are aware that in an optimal condition the use of male and female as well as young and aged hamsters would have been preferable and that the results of this study cannot be extrapolated to females and young hamsters. However, our aim was to offer a model for a population at high risk of respiratory long-COVID. Risk factors for the development of respiratory long-term sequelae include severe acute disease^{8,21,23,137,138}, advanced age^{3,139} and sex, with males being more likely to develop severe acute disease^{8,20-23,26}. Moreover, male sex is associated with higher risk for respiratory long-COVID with diffusion impairment and restriction in humans²¹. Similarly, male hamsters show more severe disease, lung function impairment, and histological lesions in the acute phase^{26,28-30,67}. The few long-term studies extending beyond the first 2 weeks also report males as recovering more slowly than females^{26,29,30}. Thus, to increase the likelihood of observing long lasting pulmonary long-COVID, we used one year old male hamsters. Future studies might have to explore whether there are sex-specific differences in the long-term consequences of infection in the hamster model.

“justification for only SARS-CoV-2”

We thank Reviewer 3 for their comment. As a matter of fact, we did not claim that the changes that we have shown are SARS-CoV-2 specific, and we would refrain from doing so. Post-COVID-19 lung changes in human patients can range from organizing lung injury and post-ARDS fibrotic sequelae to bronchiolitis, phenotypes that can be observed following many viral causes of lung injury. Indeed, the lung has a limited spectrum of reaction patterns in response to alveolar injury. The current understanding is that AT2 cells are mainly responsible for AT1 cell regeneration in homeostatic turnover and following mild injury, while airway progenitors are recruited after severe injury with marked AT1 cell loss (PMID: 31978363; PMID: 3214265; PMID: 25533958). The participation of airway progenitors and ADI cells in alveolar regeneration has been demonstrated in many publications, independent of the virus (e.g. SARS-CoV-2, IAV, Sendai virus) or animal model (hamster, mouse) (PMID: 37277327; PMID: 38092883; PMID: 35857629; PMID: 34343135; PMID: 32678092). Moreover, the same processes have been shown to be partaking in alveolar regeneration also in non-infectious mouse models of lung injury (e.g. bleomycin, neonatal hypoxia and hyperoxia, LPS, PMID: 32678092; PMID: 31978363; PMID: 32073903; PMID: 30913038). We have demonstrated previously, and in this manuscript, that homologous patterns of lung regeneration are active in the SARS-CoV-2 infected hamster (PMID: 37277327).

The most comparable infectious disease is influenza A virus (IAV) infection. Hamsters are naturally susceptible to IAV and are considered a good model for the respiratory as well as systemic disease (PMID: 29212926). A number of studies has already been published on the comparisons of SARS-CoV-2 and IAV infection in hamsters (e.g. PMID: 35857629; PMID: 35648595; PMID: 33216851; PMID: 38400021). Most of the works demonstrate a higher disease severity and more pronounced inflammation in SARS-CoV-2 infection compared to IAV infection when using similar infection doses (PMID: 35648595; PMID: 33216851; PMID: 38400021). Apart from these quantitative differences, qualitative morphological features and the antiviral response appear to be comparable between these models. In one published study, with infection of young hamsters with SARS-CoV-2 and IAV with inoculation doses adapted to reach equivalent viral loads, the disease severity, histopathological lesions, inflammatory infiltrates and transcriptomic changes in the lung were qualitatively and quantitatively comparable during acute (3, 7 dpi) and chronic disease (31 dpi; PMID: 35857629). Thus, it appears that hamsters might be more susceptible to SARS-CoV-2 than to IAV, but once the infections

doses are adapted to cause the same level of lung damage, the response patterns are the same. Interestingly, sublethal SARS-CoV-2 infection of K18-hACE2 mice failed to induce airway progenitor cell proliferation in contrast to IAV infection of B6 mice (PMID: 38092883). The lack of airway progenitor cell proliferation has also been highlighted in BALB/c mice infected with the mouse-adapted SARS-CoV-2 strain MA10 (PMID: 35857635). However, airway basal cell proliferation occurs in SARS-CoV-2- and IAV-infected hamsters (PMID: 35857629). Therefore, we believe that the apparent qualitative differences observed are not indicative of a true virus-specific response pattern, but related to the severity of alveolar damage, which in turn is dependent on the virus dose and strain as well as the choice of animal species and age group. It appears that SARS-CoV-2-induced alveolar damage is more severe in hamsters whereas IAV is provoking a more severe damage in mice.

Altogether, since the studies mentioned above already demonstrate that the lung response in hamsters to viral infection appears to be stereotypical, we did not see a strong rationale to include another virus as a control group in the original study design. However, since most of the published work addressed above did not look into epithelial cells participating in alveolar regeneration in detail, we decided to perform additional analyzes targeting this aspect using a published RNAseq dataset from the hamster model. We selected the experiment published by Frere et al. (PMID: 35857629), since it showed similar responses to SARS-CoV-2 and IAV infection, contained timepoints during acute and chronic disease, and described chronic bronchiolization lesions similar to the ones described by us. We selected hallmark gene sets that were associated with certain cell types (ADI cells, club cells, airway basal cells) or the pro-fibrotic environment that showed differential expression in different disease phases of our experiment. Then, we compared the relative expression of these genes in SARS-CoV-2 infected and IAV-infected lungs from Frere et al. and identified similarities and differences to our own data. The analysis showed mostly overlapping patterns in the expression of epithelial cell genes, with comparable changes in both viral diseases and both experiments. Some differences between our data and the data from Frere et al. were observed in the pro-fibrotic gene expression in the chronic phase, with lasting changes in our hamster model as opposed to the transient upregulation in the SARS-CoV-2 and IAV-infected hamsters in Frere et al. This discrepancy could be related to the animal age, since we used old males, while the published dataset was obtained in young animals.

Following the suggestion of Reviewer 2 to investigate immune cell interactions (see also response to Point No. 8, Reviewer 2), we added an additional evaluation of immune cells implicated in pulmonary regenerative processes, that further supports similarities between SARS-CoV-2 and IAV infection. It has been recently demonstrated in a murine IAV model, that an important mediator of macrophage-epithelial cross-talk in alveolar repair is Placenta expressed transcript-1 (Plet-1) (PMID: 38167746). Thus, we decided to investigate *Plet1* and related genes on a protein level (IHC for Plet-1 with digital quantitative analysis) and on the transcriptome level (*Plet1*, *Mertk*, *Siglecf*). Interestingly, we found out that there were significantly higher numbers of pulmonary Plet-1 immunolabelled cells in SARS-CoV-2-infected compared to mock-infected hamsters at 3, 6, and 14 dpi. Transcriptome data analysis revealed an increased expression of *Mertk* and *Plet1* at 6 dpi while at 14 and 28 dpi only *Plet1* was upregulated.

Combining all this evidence, we conclude that the morphologic and transcriptomic features of lung regeneration we observe reflect a stereotypical response to alveolar damage, and not a distinct SARS-CoV-2 specific phenomenon. The fact that the lung response to SARS-CoV-2 infection in hamsters overlaps with that of other respiratory diseases highlights the usefulness of the model also beyond COVID-19 and expands the relevance of these findings to the wider field of lung regeneration. Compared to mice, the hamster model has not been used so extensively for viral respiratory diseases and the response to lung damage has not been characterized in detail. Altogether, we feel that since the hamsters has the potential to be used more widely in this field, the new information we provide

for the scientific community (longitudinal study design, characterization of cell types, rich transcriptome dataset, functional readout strategy) should be considered useful and significant.

We feel that the re-analysis of publicly available RNAseq data represents a suitable alternative approach to answer this comment raised by Reviewer 3 with respect to include another respiratory pathogen in the current study. Performing an additional long-term experiment with another virus would require new ethical committee permits, which is associated with very long processing times in Germany, which, combined with the time needed to perform a four-moth experiment in BSL3 facilities, would delay the publication of this data for at least one year. As already pointed above, we believe that the information provided in this work deserves to be made available sooner. The results described above have been added to the revised manuscript and the discussion has been expanded to highlight that the findings could apply to other diseases as well. We hope that this approach, which is also in the spirit of the 3R-principle, meets the approval of Reviewer 3.

Based on this the **results, discussion, materials and methods, and supplementary material** have been revised accordingly in the manuscript as follows:

RESULTS

new line number 809-866

10. Transcriptome dynamics of ADI and airway basal cell signature as well as pro-fibrotic genes are comparable between SARS-CoV-2 and IAV infection in hamsters

One of the main questions when analyzing data from COVID-19 patients and animal models is whether an observed morphologic, functional or transcriptomic change is specific to the disease or a common feature of respiratory infections. The most comparable infectious disease is influenza A virus (IAV) infection. Hamsters are naturally susceptible to IAV and are considered a good model for the respiratory as well as systemic disease⁹³. In one published study, with infection of young hamsters with SARS-CoV-2 and IAV with inoculation doses adapted to reach equivalent viral loads, the disease severity, histopathological lesions, inflammatory infiltrates and transcriptomic changes in the lung were qualitatively and quantitatively comparable during acute (3, 7 dpi) and chronic disease (31 dpi). In particular, both viruses induced chronic bronchiolization lesions as observed in our study⁷⁹. Since this study did not look into epithelial cells participating in alveolar regeneration in detail, we decided to analyze selected gene signatures as determined in our own analysis in this published RNAseq dataset.

We selected the datasets obtained at 3 and 31 dpi as representative for acute and chronic infection, respectively, and compared the relative expression of gene sets in SARS-CoV-2 infected and IAV-infected lungs. Subsequently, we compared them to the relative expression in our hamsters at 3, 6 and 56 dpi, representing the timepoints with earliest, most prominent and latest significant changes, respectively. We chose to focus on genes expressed by early-stage ADI cells (*S100a6*, *Krt8*, *Anxa1*, *Tp53*, and *Hbgef*), late-stage ADI cells (*Sparc*, *Sox4*, and *Wwtr1*), club cells (*Gss* and *Pigr*), airway basal cells (*Ngfr*, *Pou2f3*, *Krt14*, and *Krt5*) and genes indicating a pro-fibrotic signature (*Mmp12*, *Mmp14*, *Tgfb1*, *Col5a1*, *Col1a1*, and *Col3a1*), which were differentially expressed in our own experiment.

In our dataset, early ADI cells genes were upregulated in the acute phase, whereas late ADI genes are variably upregulated in the chronic phase from 14 dpi onwards, as described above. In the study by Frere et al., SARS-CoV-2-infected animals showed a similar pattern of expression for ADI cells genes, whereas IAV-infected hamsters displayed a mostly homogeneous ADI cell gene expression already in the acute phase of the disease (Supplementary Fig. 10). This could indicate differences in the dynamics

of ADI cell trajectories, suggesting a slight delay of ADI cell maturation in SARS-CoV-2 compared to IAV infection.

Club and airway basal cell progenitor genes were upregulated mainly in the chronic phase in our hamsters. In the study by Frere et al., SARS-CoV-2-infected animals showed only a slight upregulation of typical club and airway basal cell progenitor genes in the chronic phase of the disease, whereas IAV-infected hamsters displayed a more prominent upregulation of these genes at this phase (Supplementary Fig. 10). Of note, the authors reported that bronchiolization areas appeared more prominent in SARS-CoV-2 infected animals compared to IAV-infected ones in histology, albeit no significant difference was observed upon morphometry.

In our dataset, early pro-fibrotic environment genes involved in ECM-remodeling were upregulated in the acute phase, while collagen encoding genes were upregulated in the chronic phase of the disease, as stated above. The study by Frere et al showed a comparable pattern of expression in both SARS-CoV-2- and IAV-infected hamsters, but upregulation of both ECM-remodeling and collagen encoding genes was only noted in the acute phase of the disease (Supplementary Fig. 10). This discrepancy is in line with the lack of obvious collagen deposition reported by Frere et al., as opposed to the significant interstitial collagen deposition reported in this study.

In summary, comparison of our data with a published dataset from SARS-CoV-2 and IAV-infected hamsters showed mostly overlapping patterns in the expression of epithelial cell genes, with comparable changes in both viral diseases and both experiments. Differences between our data and the data from Frere et al. were observed in the pro-fibrotic gene expression in the chronic phase, with lasting changes in our hamster model as opposed to the transient upregulation in the SARS-CoV-2 and IAV-infected hamsters in Frere et al.

DISCUSSION

new line number 1168-1226

The current understanding of alveolar regeneration is that AT2 cells are mainly responsible for AT1 cell regeneration in homeostatic turnover and following mild injury, while airway progenitors are recruited after severe injury with marked AT1 cell loss^{75,121,122}. The participation of airway progenitors and ADI cells in alveolar regeneration has been demonstrated in different viral respiratory diseases (SARS-CoV-2, IAV, Sendai virus)^{18,73,74,79,123}, and non-infectious mouse models of lung injury (e.g. bleomycin, neonatal hypoxia and hyperoxia, LPS)^{74,75,124,125}. A contribution of airway progenitors to alveolar repair has been reported in COVID-19 patients and their presence has been associated with a profibrotic gene signature and presence of fibroblastic areas and bronchiolization. We have shown here and in our previous publication¹⁸, that the hamster recapitulates this pattern. The question remains, whether any of this response is SARS-CoV-2 specific or whether it could be observed in any other viral pneumonia. The most comparable infectious disease is IAV infection. Hamsters are naturally susceptible to IAV and are considered a good model for the respiratory as well as systemic disease⁹³. A number of studies have been published on the comparisons of SARS-CoV-2 and IAV infection in hamsters^{79,129-131}. Many of these publications demonstrate a higher disease severity and more pronounced inflammation in SARS-CoV-2 infection compared to IAV infection, when using similar infection doses. However, apart from these quantitative differences, qualitative morphological features and the antiviral response appear to be comparable between both viruses^{129,131,132}. In one published study, the infection doses of SARS-CoV-2 and IAV were adapted to reach equivalent viral loads, resulting in comparable disease severity, histopathological lesions, inflammatory infiltrates and transcriptomic changes in the lung during acute and chronic disease⁷⁹. Thus, it appears that hamsters might be more susceptible to SARS-CoV-2 than to IAV, but once the infections doses are adapted to cause the same level of alveolar damage, the response patterns are the same. Our analysis of the

published transcriptome dataset obtained from SARS-CoV-2 and IAV infected young hamsters and comparison with our transcriptome data in old SARS-CoV-2 infected hamsters confirmed similarities of the two diseases regarding cell types participating in alveolar regeneration. In addition, we demonstrated here that Plet-1, which has been recently identified as an important mediator of macrophage-epithelial cross-talk in alveolar regeneration in the IAV model⁷⁰, is also upregulated during the regeneration phase of SARS-CoV-2 infection. Combining all the evidence, we conclude that the morphologic and transcriptomic features of lung regeneration we observe in hamsters reflect a stereotypical response to severe alveolar damage, and not a distinct SARS-CoV-2 specific phenomenon. The fact that the response overlaps with that of other respiratory diseases highlights the usefulness of the model also beyond COVID-19 and expands the relevance of the findings beyond the COVID-19 research field.

Interestingly, mobilization of airway progenitors appears not to be a feature of SARS-CoV-2 infection in the mouse models. For instance, sub-lethal SARS-CoV-2 infection of K18-hACE2 mice failed to induce airway progenitor cell proliferation in contrast to IAV infection of B6 mice⁷³. The lack of airway progenitor cell proliferation has also been highlighted in BALB/c mice infected with the mouse-adapted SARS-CoV-2 strain MA10⁷¹. We believe that the apparent qualitative differences between viral models observed by some are not indicative of a true virus-specific response pattern, but most likely related to the severity of alveolar damage, which in turn is dependent on the virus dose and strain as well as the choice of animal species and age group. It appears that SARS-CoV-2-induced alveolar damage is more severe in hamsters whereas IAV is provoking a more severe damage in mice.

One discrepancy between the dataset from young and old hamsters was found regarding pro-fibrotic gene expression. In young SARS-CoV-2- and IAV-infected hamsters, pro-fibrotic genes were only upregulated in the acute disease, while we additionally noted upregulation of collagen-encoding genes in the chronic phase. This discrepancy is in line with the lack of obvious collagen deposition reported by Frere et al., as opposed to the significant interstitial collagen deposition reported in this study. This discrepancy is most likely related to the age difference, since the aged lung shows an impaired regeneration capacity and increased pro-fibrotic changes following alveolar damage¹³³.

MATERIAL AND METHODS

new line number 1575-1577

Data from Frere et al.⁷⁹ was downloaded from the GEO public database as raw counts and normalized with the rlog function (regularized log transformation) in DESeq2.

new Supplementary Figure 10

a Comparison SARS-CoV-2 / IAV

Supplementary figure 10: comparison SARS-CoV-2 and IAV-infection in hamsters.

Genes associated with a pro-fibrotic environment, ADI, club, or basal cells expressed in the lung of SARS-CoV-2 infected hamsters in this experiment were selected and the expression pattern was compared with a published dataset obtained in SARS-CoV-2- and IAV-infected hamsters (Frere et al., 2022). Heatmaps (pheatmap) show log₂ fold changes of normalized expression values (means per group) to the respective mock control means. Expression values are scaled by row. Red/orange indicates higher and blue/lightblue lower relative expression levels.

3. Lines 337-339 - Interesting. Others speculate that long term persistence of viral RNA contributes to chronic disease. More discussion of this is needed.

RESPONSE: The authors are thankful to Reviewer 3 for their insightful comment that helped us to improve the scientific value of our manuscript. Prompted by the request to extend the discussion regarding the persistence of viral RNA and the relationship to the chronic disease, we reanalyzed the published evidence in more detail and, in addition to extending the discussion, decided to perform additional analysis on the topic. Therefore, in addition to the immunohistochemistry for SARS-CoV-2 nucleoprotein and the RNAseq data that was already included in the manuscript, we immunolabelled lung sections for viral spike protein (SP) and performed quantitative digital analysis. Moreover, since RNAseq is not the most sensitive method of choice to detect viral RNA, we added PCR for viral RNA-dependent RNA polymerase (RdRp) and subgenomic viral RNA (Esub) of lung samples from the chronic phase, from 28 dpi onward. Our quantitative analysis of SARS-CoV-2 SP on the lung parenchyma yielded similar results to SARS-CoV-2 NP quantification. However, the PCR provided interesting new insights. RdRp RNA was detected in all animals at 28 dpi, 6/7 animals at 56 dpi and 6/8 animals at 112 dpi. Esub RNA was present in 6/8 animals at 28 dpi, but was not detected at later time-points. In view of the positive results for RdRp RNA, we stained the sections of positive animals with double fluorescence for SARS-CoV-2-NP and the histiocytic marker Iba-1, since persistence of viral protein has been mostly found in lung macrophages (PMID: 34491552; PMID: 37919524). We could not detect viral protein in any of the animals (the data has not been included in the main text). The presence of Esub until 28 dpi could point to a prolonged presence of replicating virus (PMID: 32235945), although the correlation of subgenomic RNA and virus replication has been contested (PMID: 33247099). After 28 dpi, it seems that only viral RNA, but not actively replicating virus, persisted in the lungs of recovered hamsters in our study.

In animal models, only few long-term studies have been performed so far and there is limited and partly inconsistent data on virus persistence. Most studies failed to demonstrate infectious virus or viral protein beyond the acute disease phase in hamsters (PMID: 34253053; PMID: 35857629). In contrast, one study found minimal amounts of infectious virus in hamsters at 42 dpi (PMID: 34491552). It needs to be pointed out that in this particular hamster study, negative results were initially obtained by a standard CPE based titration method, and that demonstration of infectious virus was only achieved following a 7-day culture of homogenized lung tissue and subsequent titration of the cell homogenate. Another study in non-human primates demonstrated infectious virus in BAL macrophages by detecting double stranded RNA, non-structural protein expression and transcriptome changes after culturing them for 18 hours (PMID: 37919524). Thus, it seems that an initial propagation step and/or a targeted approach need to be applied to demonstrate the persistence of infectious virus in the SARS-CoV-2 model, and that negative results from studies using standard methods utilizing whole lung tissue have to be interpreted with caution. Therefore, we cannot completely exclude the presence of minimal amounts of replicating virus below the limits of detection beyond 28 dpi. Unfortunately, our experimental design did not include collection of BAL fluid and the lung samples have been stabilized in RNAlater and subsequently frozen, precluding their use for virus titration.

Interestingly, pulmonary viral persistence has been correlated with chronic weight loss in the hamster model (PMID: 34491552). Therefore, we took a closer look at the body weight dynamics in the animals of the chronic phase, and we found that the animals with detectable viral RdRp RNA in the lung were the ones with the most severe weight loss and/or the ones that showed the least recovery of initial body weight.

In view of this new evidence, we added the new results and modified and expanded the discussion section in the revised manuscript to include the hypothesis that persistence of viral RNA might have

contributed to chronic clinical and morphologic abnormalities in the hamster model. Furthermore, we included additional paragraphs critically discussing the data in the context of existing literature.

Based on these new results the **results**, **discussion** as well as **materials and methods**, and **supplementary material** have been revised accordingly in the manuscript as follows:

RESULTS

new line number 307-380

4. SARS-CoV-2 Delta infection induces transient inflammation and chronic lesions with persistence of viral RNA in the lungs of aged hamsters

Infection was confirmed by immunohistochemical staining and quantification of SARS-CoV-2 nucleoprotein (NP) antigen in the lung. As described previously¹⁸, viral antigen was present in the alveolar and airway epithelial cells as well as in alveolar macrophages (Supplementary Fig. 1). Positive cells were detected as early as 1 dpi with the highest number of positive cells at 3 dpi. At 14 dpi, no viral protein was detectable (Supplementary Fig. 1). Similar results were obtained by the immunohistochemical staining and quantification of SARS-CoV-2 spike protein (SP, (Supplementary Fig. 1). Mock-infected animals showed no viral protein at any time-point. Additionally, RNAseq revealed no SARS-CoV-2 spike (S) and nucleoprotein (N) transcripts at any time-point in the mock-infected animals. Both SARS-CoV-2 S and N transcripts peaked at 1 dpi, were marginally present until 6 dpi, and were not detected in samples from the following time-points in SARS-CoV-2-infected hamsters (Supplementary Fig. 1). There is growing evidence that either replication competent virus or viral products can persist in the lung of COVID-19 patients and SARS-CoV-2 animals beyond the seeming recovery from disease^{65,66}. Moreover, it has been reported in hamsters that a pulmonary viral persistence is associated with chronic weight loss⁶⁷. Since we observed that SARS-CoV-2 infected hamsters did not recover their initial body weight, we additionally performed PCR for viral RNA-dependent RNA polymerase (RdRp) and subgenomic viral RNA (Esub) in the lung of mock- and SARS-CoV-2 infected animals from 28 dpi onwards. Interestingly, RdRp RNA was detected in all animals at 28 dpi, 6/7 animals at 56 dpi and 6/8 animals at 112 dpi. Esub RNA was present in 6/8 animals at 28 dpi, but not detected at later time-points (Supplementary Fig. 1). Interestingly, the animals with detectable viral RdRp RNA in the lung during the chronic phase were the ones with the most severe weight loss and/or the ones that showed the least recovery of initial body weight (Supplementary Data 1).

DISCUSSION

new line number 1065-1120

It has been postulated that persistence of SARS-CoV-2 in the lung could be the key to explain lasting symptoms of respiratory long-COVID^{106,107}. Sporadic reports of viral RNA shedding for up to 83 days in the upper respiratory tract, 59 days in the lower respiratory tract, 126 days in stools, and 60 days in serum have been identified by a meta-analysis¹⁰⁸. In one study conducted in deceased patients SARS-CoV-2 Spike and NP immunohistochemistry revealed positive signal in bronchial cartilage chondrocytes and parabronchial glands of most patients, despite repeated negative results in nasopharyngeal swabs or bronchioalveolar lavage (BAL) for an average of 105 days⁶⁵. In addition, at least 60 cases with conclusive evidence for persistent SARS-CoV-2 infection and virus replication have been described¹⁰⁶. With bulk RNAseq and immunohistochemistry, we failed to detect any of the viral proteins and transcripts at time-points later than 6 dpi. However, with PCR we detected Esub RNA until 28 dpi and RdRp RNA until 112 dpi. The presence of Esub RNA until 28 dpi could point to a prolonged presence of replicating virus¹⁰⁹. However, the correlation between the presence of the Esub RNA and active virus

replication/transcription and thus active infection has been contested¹¹⁰. After 28 dpi, it seems that only viral genomic RNA persisted in the lungs of recovered hamsters in our study. In animal models, only few long-term studies have been performed so far and there is limited and partly inconsistent data on virus persistence. While most studies fail to demonstrate infectious virus, viral RNA and/or viral protein in lung beyond acute disease^{26,79}, others have reported minimal amounts of infectious virus particles at 42 dpi in the hamster model⁶⁷. Another study in macaques showed that replication competent SARS-CoV-2 was detectable in BAL macrophages beyond 6 months inducing IFN γ and NK cells dysregulation⁶⁶. The discrepancies could be related to differences in study design, choice of animals, virus strain and dose as well as to the very low amount of persistent virus and the choice of detection methods. For instance, in the hamster study demonstrating infectious virus persistence at 42 dpi, negative results were initially obtained by a standard CPE based titration method, and demonstration of infectious virus was only achieved following a 7-day culture of homogenized lung tissue and subsequent titration of the cell homogenate. The study in non-human primates demonstrated infectious virus in BAL macrophages by detecting double stranded RNA, non-structural protein expression and transcriptome changes after culturing them for 18 hours⁶⁶. Thus, it seems that an initial propagation step and/or a targeted approach need to be applied to demonstrate the persistence of infectious virus, and that the negative results from studies using standard methods utilizing whole lung tissue have to be interpreted with caution. Therefore, we cannot completely exclude the presence of minimal amounts of replicating virus below the limits of detection of immunohistochemistry.

Interestingly, pulmonary viral persistence has been associated with chronic weight loss in the hamster model by others⁶⁷, and we also found that the animals with detectable viral RNA in the lung were the ones with the most severe weight loss and/or the ones that showed the least recovery of the initial body weight. Therefore, even in the absence of replicating virus, a link between viral RNA persistence and clinical signs should be considered. A recent study conducted as part of a national infection survey reported that patients with detection of SARS-CoV-2 RNA at high titer persisting for at least 30 days had had more than 50% higher odds of self-reporting long COVID¹⁰⁶. However, one needs to be aware that the current evidence pointing to virus persistence in long-COVID-19 patients is mainly based on the detection of viral RNA, which is not necessary proof of replication-competent virus in all cases¹⁰⁶. Conclusive evidence of persisting infection, e.g. through serial isolation of infectious virus for extended periods, demonstration of within-host evolution, or visualization of viral antigen in tissues of deceased patients, is still rare^{65,107}. Chronic presence of antigen within the lung has only been demonstrated in cases with a fatal outcome and evidence of chronic lung inflammation with ongoing tissue damage⁶⁵. It remains open whether persistent infection of the lung can also be convincingly linked to milder cases of respiratory long-COVID.

MATERIALS AND METHODS

new line number 1462-1470

7. Immunohistochemistry

SARS-CoV-2 Spike protein has been added to the list of antibodies used.

new line number 1579-1598

12. Real-time PCR

RNA was isolated from lung tissue as described above. The RNA was then amplified using RT-qPCR (quantitative real-time reverse transcription PCR) targeting the RNA-dependent RNA polymerase (RdRp) and the subgenomic sequence of the envelope protein (subgenomic E) of SARS-CoV-2. In more

detail, isolated RNA was processed with the Luna® Universal One-Step RT-qPCR Kit (NEB #E3005, New England Biolabs GmbH) in a CFX96-Touch Real-Time PCR system (Bio-Rad). The following RdRp-targeting primers were used: SARS-2-IP4 forward (5'-GGTAACTGGTATGATTTCG-3'), reverse (5'-CTGGTCAAGGTTAATATAGG-3'), and probe (5'-TCATACAAACCACGCCAGG-3' [5' FAM, 3' BHQ-1]). Subgenomic E was amplified using the following primers: Subgenomic leader sequence sgLead-SARS-CoV-2 forward (5'-CGATCTCTGTAGATCTGTTCTC-3'), E_Sarbeco reverse (3'-ATATTGCAGCAGTACGCACACA-5'), and probe (5'-ACACTAGCCATCCTTACTGCGCTTCG-3' [5' FAM, 3' BBQ]). The PCR program includes the reverse transcription at 55 °C for 10 minutes with a subsequent denaturation at 95 °C for 1 minute. This step is followed by 44 cycles of denaturation (95°C, 20 seconds) and annealing and elongation (56°C, 30 seconds) of the samples. Relative fluorescence units (RFUs) were measured at the end of the elongation phase. A standard RNA transcript was used to convert the obtained Ct values into copy numbers per µl of total RNA. Ct values > 37 were evaluated as negative results.

New graphs in Supplementary Figure 1

Supplementary figure 1: Experimental design, exercise tolerance and virus quantification

a Schematic drawing of the experimental design. Male, 1-year old Syrian hamsters were infected with SARS-CoV-2 Delta variant and sacrificed at 1, 3, 6, 14, 28, 56 and 112 dpi. The study design allowed to distinguish three phases of the disease: acute phase (infection – 6 dpi), sub-acute phase (6 – 28 dpi), and chronic phase (28 dpi – 112 dpi). During the experiment, repeated lung function measurements were conducted using whole-body plethysmography (WBP) with respiratory gas analysis. Physical

exercise on a rodent treadmill was used to exacerbate possible latent respiratory impairment from 21 dpi onwards and was repeated weekly. **b** Scoring of exercise tolerance (running behavior on the treadmill, mean and SEM). N = 18 (Training 1 until 28 dpi), 12 (35-56 dpi), or 6 (63-112 dpi) animals/group. **c-d** Immunohistochemistry for SARS-CoV-2 nucleoprotein (NP, c), and spike protein (SP, d). For each staining, an overview, a high magnification and the quantification in the whole section of the left lung lobe is shown. Quantitative data is shown as box and whisker plots. The bounds of the box plot indicate the 25th and 75th percentiles, the bar indicates medians, and the whiskers indicate minima and maxima. Dots show individual values. Data was tested by two-tailed Mann–Whitney U test. A p value of ≤ 0.05 was chosen as the cutoff for statistical significance. N = 7-8 animals/group/time-point. **e** Beeswarm plots for SARS-CoV-2 genes. The bounds of the box plot indicate the 25th and 75th percentiles, the bar indicates medians, and the whiskers indicate minima and maxima. Dots show individual values. N = 4 animals/group/time-point. **f** PCR for SARS-CoV-2 RNA-dependent RNA polymerase (RdRp) and subgenomic viral RNA in the chronic phase. Graphs show mean, SEM, and individual values. N = 7-8 animals/group/time-point. Negative results were set to -1 to enhance readability. Source data is provided as a Source Data file.

4. Lines 358-386 - Technically, if its not statistically significant then its not really increased.

RESPONSE: We apologize for the statement. The sentence has been removed.